# On the Cryptographic Hardness
# of Learning Single Periodic Neurons

**Min Jae Song**[*]
Courant Institute
New York University
minjae.song@nyu.edu

**Ilias Zadik**[*]
Department of Mathematics
Massachusetts Institute of Technology
izadik@mit.edu

**Joan Bruna**
Courant Institute
Center for Data Science
New York University
bruna@cims.nyu.edu

## Abstract

We show a simple reduction which demonstrates the cryptographic hardness of learning a single periodic neuron over isotropic Gaussian distributions in the presence of noise. More precisely, our reduction shows that *any* polynomial-time algorithm (not necessarily gradient-based) for learning such functions under small noise implies a polynomial-time quantum algorithm for solving *worst-case* lattice problems, whose hardness form the foundation of lattice-based cryptography. Our core hard family of functions, which are well-approximated by one-layer neural networks, take the general form of a univariate periodic function applied to an affine projection of the data. These functions have appeared in previous seminal works which demonstrate their hardness against gradient-based (Shamir'18), and Statistical Query (SQ) algorithms (Song et al.'17). We show that if (polynomially) small noise is added to the labels, the intractability of learning these functions applies to *all polynomial-time algorithms*, beyond gradient-based and SQ algorithms, under the aforementioned cryptographic assumptions. Moreover, we demonstrate the *necessity of noise* in the hardness result by designing a polynomial-time algorithm for learning certain families of such functions under exponentially small adversarial noise. Our proposed algorithm is not a gradient-based or an SQ algorithm, but is rather based on the celebrated Lenstra-Lenstra-Lovász (LLL) lattice basis reduction algorithm. Furthermore, in the absence of noise, this algorithm can be directly applied to solve CLWE detection (Bruna et al.'21) and phase retrieval with an optimal sample complexity of $d + 1$ samples. In the former case, this improves upon the quadratic-in-$d$ sample complexity required in (Bruna et al.'21).

## 1  Introduction

The empirical success of Deep Learning has given an impetus to provide theoretical foundations explaining when and why it is possible to efficiently learn from high-dimensional data with neural networks. Currently, there are large gaps between positive and negative results for learning, even for the simplest neural network architectures [61, 29, 14, 27]. These gaps offer a large ground for debate, discussing the extent up to which improved learning algorithms can be designed, or whether a fundamental computational barrier has been reached.

One particular challenge in closing these gaps is establishing negative results for improper learning in the distribution-specific setting, in which the learner can exploit the peculiarities of a known input distribution, and is not limited to outputting hypotheses from the target function class. Over the last few years, authors have successfully developed distribution-specific hardness results in the context of learning neural networks, offering different flavors. On one hand, there have been several results

---

[*]Equal contribution

35th Conference on Neural Information Processing Systems (NeurIPS 2021).

proving the failure of a restricted class of algorithms, such as gradient-based algorithms [54, 52], or more generally Statistical Query (SQ) algorithms [33, 20, 57, 29, 17]. Notably, such results apply to the simplest cases, such as learning one-hidden-layer neural networks over the standard Gaussian input distribution [29, 17]. On the other hand, a different line of work has shown the hardness of learning *two*-hidden-layer neural networks for *any* polynomial-time algorithm by leveraging cryptographic assumptions, such as the existence of local pseudorandom generators (PRGs) with polynomial stretch [15]. Despite such significant advances, important open questions remain, such as whether the simpler case of learning one hidden-layer neural network over standard Gaussian input remains hard for algorithms not captured by the SQ framework. To make this question more precise, are non-SQ polynomial-time algorithms, which may inspect individual samples – such as stochastic gradient descent (SGD) [1] – able to learn one-hidden layer neural networks over Gaussian input? Understanding the answer to this question is a partial motivation of the present work.

A key technique for constructing hard-to-learn functions is leveraging "high-frequency" oscillations in high-dimensions. The simplest instance of such functions is given by pure cosines of the form $f(x) = \cos(2\pi\gamma\langle w, x\rangle)$, where we refer to $w \in S^{d-1}$ as its *hidden direction*, and $\gamma$ as its *frequency*. Such functions have already been investigated by previous works [57, 54, 52] in the context of lower bounds for learning neural networks. For these hard constructions, the frequency $\gamma$ is taken to scale polynomially with the dimension $d$. Note that as the univariate function $\cos(2\pi\gamma t)$ is $O(\gamma)$-Lipschitz, the function $f$ is well-approximated by one-hidden-layer ReLU network of poly($\gamma$)-width on any compact set (see e.g., Appendix G). Hence, understanding the hardness of learning such functions is an unavoidable step towards understanding the hardness of learning one-hidden-layer ReLU networks.

In this work, we pursue this line of inquiry, focusing on weakly learning the cosine neuron class over the standard Gaussian input distribution in the presence of noise. Our main result is a proof, via a reduction from a fundamental problem in lattice-based cryptography called the Shortest Vector Problem (SVP), that such learning task is hard for *any* polynomial-time algorithm, based on the widely-believed cryptographic assumption that (approximate) SVP is computationally intractable against quantum algorithms (See e.g., [49, 43, 19, 3] and references therein). Our result therefore extends the hardness of learning such functions from a restricted family of algorithms, such as gradient-based algorithms or SQ, to *all* polynomial-time algorithms by leveraging cryptographic assumptions. Note, however, that SQ lower bounds are unconditional because they are of an information-theoretic nature. Therefore, our result, which is conditional on a computational hardness assumption, albeit a well-founded one in the cryptographic community, and SQ lower bounds are not directly comparable.

The problem of learning cosine neurons with noise can be studied in the broader context of inferring hidden structures in noisy high-dimensional data, as a particular instance of the family of Generalized Linear Models (GLM) [45, 44]. Multiple inference settings, including, for example, the well-known planted clique model [30, 5], but also GLMs such as sparse regression [23] exhibit so-called *computational-to-statistical gaps*. These gaps refer to intervals of signal-to-noise ratio (SNR) values where inference of the hidden structure is possible by exponential-time estimators but appears out of reach for any polynomial-time estimator. Following this line of work, we define the SNR of our cosine neuron learning problem to be the inverse of the noise level, and analyze its hardness landscape. As it turns out, weakly learning the cosine neuron class provides a rich landscape, yielding a computational-to-statistical gap based on *a worst-case hardness* guarantee. We note that this is in contrast with the "usual" study of such gaps where such worst-case hardness guarantees are elusive and they are mostly based on the refutation of restricted computational classes, such as Sum-of-Squares [10], low-degree polynomials [37], Belief Propagation [9], or local search methods [24].

Finally, we establish an upper bound for the computational threshold, thanks to a polynomial-time algorithm based on the Lenstra-Lenstra-Lovász(LLL) lattice basis reduction algorithm (see details in Section 3.3). Our proposed algorithm is shown to be highly versatile, in the sense that it can be directly used to solve two seemingly very different GLMs: the CLWE detection problem from cryptography and the phase retrieval problem from high-dimensional statistics. Remarkably, this method bypasses the SQ and gradient-based hardness established by previous works [54, 57]. Our use of the LLL algorithm to bypass a previously considered "computationally-hard" region adjoins similar efforts to solve linear regression with discrete coefficients [60, 26], [37, Sec. 4.2.1], as well as the *correspondence retrieval* problem [6], which includes phase retrieval as a special case. We show in Section 3.3 and Appendix F how our algorithms obtain optimal sample complexity for recovery in both these problems in the noiseless setting. An interesting observation is that in the latter case, the resulting algorithm, and also the very similar LLL-based algorithm by [6], improves upon AMP-based

algorithms [11] in terms of sample complexity, often thought to be optimal among all polynomial-time algorithms [41]. While our LLL algorithm can be seen as an appropriate modification of [6], our analysis employs different tools, leading to improved guarantees. More precisely, our analysis easily extends to distributions that are both log-concave and sub-Gaussian, as opposed to solely Gaussian in [6]. In addition, our algorithm incorporates an explicit rounding step for LLL, which allows us to determine its precise noise-tolerance (see details in Appendix F).

## 1.1 Related work

**Hardness of learning from cryptographic assumptions.** Among several previous works [34, 35] which leverage cryptographic assumptions to establish hardness of improper learning, most relevant to our results is the seminal work of Klivans and Sherstov [36] whose hardness results are also based on SVP. To elaborate, they show that learning intersections of halfspaces, which can be seen as neural networks with the threshold activation, is hard based on the worst-case hardness of GapSVP, a decision version (approximate) SVP. Our work differs, though, in several important aspects from theirs. First, and perhaps most importantly, our result holds over the well-behaved Gaussian input distribution over $\mathbb{R}^d$, whereas their hardness utilizes a non-uniform distribution over the Boolean hypercube $\{0, 1\}^d$. Second, at a technical level and in agreement with our continuous input domain and their discrete input domain, we take a different reduction route from SVP. Their link to SVP is the Learning with Errors (LWE) Problem [49], whereas our link in the reduction is the recently developed Continuous Learning with Errors (CLWE) Problem [13]. On another front, very recently, [15] presented an abundance of novel hardness results in the context of improper learning by assuming the mere *existence* of Local Pseudorandom Generators (LPRGs) with polynomial stretch. While the LPRG and SVP assumptions are not directly comparable, we emphasize that we rely on the worst-case hardness of GapSVP, whereas LPRG assumes average-case hardness. A worst-case hardness assumption is arguably weaker as it requires only *one* instance to be hard, whereas an average-case hardness assumption requires instances to be hard on average.

**Lower bounds against restricted class of algorithms and upper bounds.** As mentioned previously, a widely adapted method for proving hardness of learning is through SQ lower bounds [33, 12, 58, 20]. Among previous work, most closely related to our work is [57] and [54], who consider learning linear-periodic function classes which contain cosine neurons. By constructing a different class of hard one-hidden-layer networks, stronger SQ lower bounds over the Gaussian distribution, in terms of both query complexity and noise rate, have been established [29, 17]. Yet, for technical reasons, the SQ model cannot rule out algorithms such as stochastic gradient descent (SGD), since these algorithms can in principle inspect each sample individually. In fact, [1] carry this advantage of SGD to the extreme and show that SGD is *poly-time universal*. [7] establishes sharp bounds using SGD for weakly learning a single planted neuron, and reveals a fundamental dependency between the regularity of their *dimension-independent* activation function, which they name the "information exponent", and the sample complexity. The regularity of the activation function has been leveraged in several works to yield positive learning results [31, 61, 27, 56, 4, 28, 21, 16]. Finally, statistical-to-computational gaps using the family of Approximate Message Passing (AMP) algorithms [18, 47] for the algorithmic frontier have been established in various high-dimensional inference settings, including proper learning of certain single-hidden layer neural networks [8], spiked matrix-tensor recovery [50] and also GLMs [11].

**The LLL algorithm and statistical inference problems.** For our algorithmic results, we employ the LLL algorithm. Specifically, our techniques are originally based on the breakthrough use of the LLL algorithm to solve a class of average-case subset sum problems in polynomial-time, as established first by Lagarias and Odlyzko [38] and later via a greatly simplified argument by Frieze [22]. While the power of LLL algorithm is very well established in the theoretical computer science [53, 39], integer programming [32], and computational number theory communities (see [55] for a survey), to the best of our knowledge, it has found only a handful of applications in the theory of statistical inference. Nevertheless, a few years ago, a strengthening of the original LLL-based arguments by Lagarias, Odlyzko and Frieze has been used to prove that linear regression with rational-valued hidden vector and continuous features can be solved in polynomial-time given access only to one sample [60]. This problem has been previously considered "computationally-hard" [23] and is proven to be impossible for the LASSO estimator [59, 25], greedy local-search methods [23] and the AMP algorithm [48]. In a subsequent work to [60], the suggested techniques have been generalized to the linear regression and phase retrieval settings under the more relaxed assumptions of discrete (and therefore potentially irrational)-valued hidden vector [26]. Our work is based on insights from

[60, 26], but is importantly generalizing the use of the LLL algorithm (a) for the recovery of an arbitrary unit *continuous-valued* hidden vector and (b) for multiple GLMs such as the cosine neuron, the phase retrieval problem, and the CLWE problem. However, for noiseless phase retrieval, we note that the optimal sample complexity of $d + 1$ has previously been achieved by [6] using an LLL-based algorithm very similar to ours.

## 1.2 Main Contributions: the Hardness Landscape of Learning Cosine Neurons

In this work, we thoroughly study the hardness of improperly learning single cosine neurons over isotropic $d$-dimensional Gaussian data. We study them under the existence of a small amount of adversarial noise per sample, call it $\beta \geq 0$, which we prove is necessary for the hardness to take place. Specifically we study improperly (weakly) learning in the squared loss sense, the function $f(x) = \cos(2\pi\gamma\langle w, x\rangle)$, for some hidden direction $w \in S^{d-1}$, from $m$ samples of the form $z_i = f(x_i) + \xi_i, i = 1, \ldots, m$ where $x_i \overset{\text{i.i.d.}}{\sim} N(0, I_d)$ and arbitrary $|\xi_i| \leq \beta$.

**Information-theoretic bounds under constant noise.** We first address the statistical, or also known as information-theoretic, question of understanding for which noise level $\beta$ one can hope to learn $f(x)$ from polynomially in $d$ many samples, by using computationally unconstrained estimators. Since the range of the functions $f = f_w$ is the interval $[-1, 1]$ it is a trivial observation that for any $\beta \geq 1$ learning is impossible. This follows because the (adversarial) noise could then produce always the uninformative case where $z_i = 0$ for all $i = 1, \ldots, m$.

Our first result (see Section 3.1 for details), is a design and analysis of an algorithm which runs in $O(\exp(d\log(\gamma/\beta)))$ time and satisfies the following property. For any $\beta$ smaller than a *sufficiently small constant*, the output hypothesis of the algorithm learns the function $f$ with access to $O(d\log(\gamma/\beta))$ samples, with high probability. To the best of our knowledge, such an information-theoretic result has not appeared before in the literature of learning a single cosine neuron. We consider this result essential and reassuring as it implies that the learning task is statistically achievable if $\beta$ is less than a small constant. Therefore, any hardness claim in terms of polynomial-time algorithms aiming to learn this function class is meaningful and implies a computational barrier.

**Cryptographic hardness under moderately small noise.** Our second and main result, presented in Section 3.2, is a reduction establishing that (weakly) learning this function class under any $\beta$ which scales at least inverse polynomially with $d$, i.e. $\beta \geq d^{-C}$ for some constant $C > 0$, is as hard as a worst-case lattice problem on which the security of lattice-based cryptography is based on.

**Theorem 1.1** (Informal). *Consider the function class $\mathcal{F}_\gamma = \{f_{\gamma,w}(x) = \cos(2\pi\gamma\langle x, w\rangle) \mid w \in \mathcal{S}^{d-1}\}$. Weakly learning $\mathcal{F}_\gamma$ over Gaussian inputs $x \sim N(0, I_d)$ under any inverse-polynomial adversarial noise when $\gamma \geq 2\sqrt{d}$ and $\beta = 1/\text{poly}(d)$, is hard, assuming worst-case lattice problems are secure against quantum attacks.*

The exact sense of cryptographic hardness used is that weakly learning the single cosine neuron under the described assumptions, reduces to solving a *worst-case* lattice problem, known as the Gap Shortest Vector Problem (GapSVP). The approximation factor of GapSVP obtained in our reduction, is not known to be NP-hard [2], but it is widely believed to be computationally hard against any polynomial-time algorithm, including quantum algorithms [43]. The reduction makes use of a recently developed average-case detection problem, called Continuous Learning with Errors (CLWE) [13] which has been established to be hard under the same hardness assumption on GapSVP. Our reduction shows that weakly learning the single cosine neuron in polynomial time, implies a polynomial-time algorithm for solving the CLWE problem (see Section 2 for the definition). The link here between the two settings comes from the periodicity of the cosine function, and the fact that the CLWE has an appropriate $\text{mod } 1$ structure, as well.

Interestingly, our reduction works for any class of function $g(x) = \phi(\gamma\langle w, x\rangle)$ where $\phi$ is a 1-periodic and $O(1)$-Lipschitz function and under $\gamma \geq 2\sqrt{d}$, generalizing the hardness claim much beyond the single cosine neuron. Moreover, our reduction shows that the computational hardness in fact applies to a certain position-dependent *random* noise model, instead of bounded *adversarial* noise (See Remark 3.5). Lastly, as mentioned above, we highlight that this is a (conditional) lower bound against *any* polynomial-time estimator, not just SQ or gradient-based methods.

**Polynomial-time algorithm under exponentially small noise.** We finally address the question of whether there is some polynomial-time algorithm that can weakly learn the single cosine neuron, in

the presence of potentially exponentially small noise. Notably, the current lower bound with respect to SQ [57] or gradient based methods [54] apply without any noise assumption *per-sample*, raising the suspicion that no "standard" learning method works even in the case $\beta = 0$.

We design and analyze an algorithm for the single cosine neuron which provably succeeds in learning the function $f = f_w$ when $\beta \leq \exp(-\tilde{O}(d^3))$, and with access to only $d + 1$ samples. Note that this sample complexity is perhaps surprising: one needs *only one more sample* than just receiving the samples in the "pure" linear system form $\langle w, x_i \rangle$ instead of $\cos(2\pi\gamma\langle w, x_i \rangle) + \xi_i$. The algorithm comes from reducing the problem to an integer relation detection question and then make a careful use of the powerful Lenstra-Lenstra-Lovász (LLL) lattice basis reduction algorithm [40] to solve it in polynomial time. The integer relation detection allows us to recover the (unknown) integer periods naturally occuring because of the periodicity of the cosine, which then allows us to provably "invert" the cosine, and then learn the hidden direction $w$ simply by solving a linear system.

The LLL algorithm is a celebrated algorithm in theoretical computer science and mathematics, which has rarely been used in the learning literature (with the notable recent exceptions [6, 60, 26]). We consider our connection between learning the single cosine neuron, integer relation detection and the LLL algorithm, a potentially interesting algorithmic novelty of the present work. We note that [13] likewise use the LLL algorithm to solve CLWE in the noiseless setting. When applied to CLWE, our algorithm, via a significantly more involved application of the LLL algorithm and careful analysis, improves upon their algorithm in terms of both sample complexity and noise-tolerance.

**Application to noiseless phase retrieval: $d + 1$ samples suffice.** Notice that the cosine activation function loses information in two distinct steps: first it "loses" the sign, since it is an even function, and then it "loses" localisation beyond its period (fixed at $[-1/2, 1/2)$). As a result, any algorithm learning the cosine neuron (such as our proposed LLL-based algorithm) can be immediately extended to solve the two separate cases, where one only loses the sign (which is known as the phase retrieval problem in high dimensional statistics) or only the localisation (which is known as the CLWE problem in cryptography). In particular, the noiseless cosine learning problem 'contains' the phase retrieval problem, where one is asked to recover an unknown vector $w$ from measurements $|\langle x_i, w \rangle|$, since $\cos(2\pi\gamma\langle w, x_i \rangle) = \cos(2\pi\gamma|\langle x_i, w \rangle|)$. Therefore, as an immediate consequence of our algorithmic results, we achieve the optimal sample complexity of noiseless[2] phase retrieval. As mentioned previously, this algorithmic result, while interesting and a consequence of our analysis for cosine learning, has already been established in the prior work [6] using a very similar LLL-based algorithm.

We note that achieving in polynomial-time the optimal sample complexity is perhaps of independent interest from a pure algorithm design point of view. While Gaussian elimination can trivially solve for $w$ given $d$ samples of the form $\langle x_i, w \rangle$ where $x_i \stackrel{\text{i.i.d.}}{\sim} N(0, I_d)$, our algorithm shows that by "losing" the sign of $\langle x_i, w \rangle$ one needs only one sample more to recover again $w$ in polynomial-time. However, we remark that the LLL algorithm has a running time of $O(d^6 \log^3 M)$ [46][3], where $d$ is the dimension and $M$ is the maximum $\ell_2$-norm of the given lattice basis vectors, which can relatively quickly become computationally challenging with increasing dimension despite its polynomial time complexity. We refer the reader to Appendix F for a formal statement of the phase retrieval problem and our results.

## 2 Definitions and Notations

**Distribution-specific PAC-learning.** We consider the problem of learning a sequence of real-valued function classes $\{\mathcal{F}_d\}_{d\in\mathbb{N}}$, each over the standard Gaussian input distribution on $\mathbb{R}^d$, an instance of what is called distribution-specific PAC learning [35, 54]. The input is a multiset of i.i.d. labeled examples $(x, y) \in \mathbb{R}^d \times \mathbb{R}$, where $x \sim N(0, I_d)$, $y = f(x) + \xi$, $f \in \mathcal{F}_d$, and $\xi \in \mathbb{R}$ is some type of observation noise. We denote by $D = D_f$ the resulting data distribution. The goal of the learner is to output an hypothesis $h : \mathbb{R}^d \to \mathbb{R}$ that is close to the target function $f$ in the squared loss sense over the Gaussian input distribution. We say a learning algorithm is *proper* if it outputs an hypothesis $h \in \mathcal{F}_d$. On the other hand, we say a learning algorithm is *improper* if $h$ is not necessarily in $\mathcal{F}_d$ [51]. We omit the index $d$, when the input dimension is clear from the context.

We denote by $\ell : \mathbb{R} \times \mathbb{R} \to \mathbb{R}_{\geq 0}$ the squared loss function defined by $\ell(y, z) = (y - z)^2$. For a given hypothesis $h$ and a data distribution $D$ on pairs $(x, z) \in \mathbb{R}^d \times \mathbb{R}$, we define its *population loss $L_D(h)$*

---

[2]Or exponentially small noise; see Corollary F.1 for the precise statement

[3]The $L^2$ algorithm by [46] speeds up LLL using floating-point arithmetic, but the running time still grows faster than $O(d^5)$.

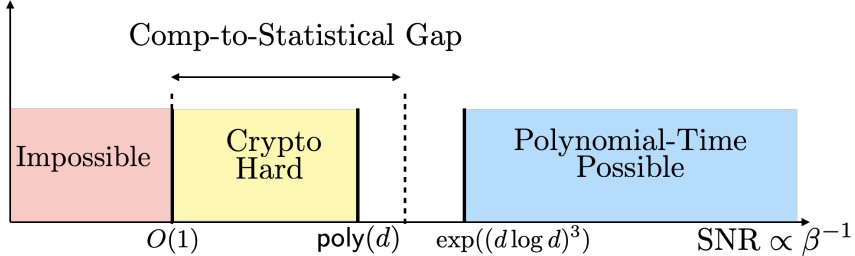

Figure 1: Our results at a glance for weakly learning the class $\mathcal{F}_\gamma$. Section 3.1 describes information-theoretical limits, Section 3.2 presents the reduction from CLWE, while Section 3.3 introduces an efficient algorithm based on LLL.

over a data distribution $D$ by

$$L_D(h) = \mathop{\mathbb{E}}_{(x,y)\sim D}[\ell(h(x), y)] . \tag{1}$$

**Definition 2.1** (Weak learning). *Let $\epsilon = \epsilon(d) > 0$ be a sequence of numbers, $\delta \in (0,1)$ a fixed constant, and let $\{\mathcal{F}_d\}_{d\in\mathbb{N}}$ be a sequence of function classes defined on input space $\mathbb{R}^d$. We say that a (randomized) learning algorithm $\mathcal{A}$ $\epsilon$-weakly learns $\{\mathcal{F}_d\}_{d\in\mathbb{N}}$ over the standard Gaussian distribution if for every $f \in \mathcal{F}_d$ the algorithm outputs a hypothesis $h_d$ such that for large values of $d$ with probability at least $1 - \delta$*

$$L_{D_f}(h_d) \leq L_{D_f}(\mathbb{E}[f(x)]) - \epsilon .$$

*Note that $\mathbb{E}[f(x)]$ is the best constant predictor for the data distribution $D = D_f$. Hence, we refer to $L_D(\mathbb{E}[f(x)]) = \mathrm{Var}_{Z\sim N(0,I_d)}(f(Z))$, as the trivial loss, and $\epsilon$ as the edge of the learning algorithm.*

From simplicity, we refer to an hypothesis as *weakly learning* a function class if it can achieve edge $\epsilon$ which is depending inverse polynomially in $d$.

**Periodic Neurons.** Let $\gamma = \gamma(d) > 1$ be a sequence of numbers indexed by the input dimension $d \in \mathbb{N}$, and let $\phi : \mathbb{R} \to [-1, 1]$ be an 1-periodic function. We denote by $\mathcal{F}_\gamma^\phi$ the function class

$$\mathcal{F}_\gamma^\phi = \{f : \mathbb{R}^d \to [-1, 1] \mid f(x) = \phi(\gamma\langle w, x\rangle), w \in S^{d-1}\} \tag{2}$$

Note that each member of the function class $\mathcal{F}_\gamma^\phi$ is fully characterized by a unit vector $w \in S^{d-1}$. We refer such function classes as *periodic neurons*.

**Cosine Learning.** We define the *cosine distribution* on dimension $d$ with frequency $\gamma = \gamma(d)$, adversarial noise rate $\beta = \beta(d)$, and hidden direction $w \in S^{d-1}$ to be the distribution of samples of the form $(x, z) \in \mathbb{R}^d \times \mathbb{R}$, where $x \overset{\text{i.i.d.}}{\sim} N(0, I_d)$, some bounded adversarial noise $|\xi| \leq \beta$, and

$$z = \cos(2\pi\gamma\langle w, x\rangle) + \xi. \tag{3}$$

The cosine learning problem consists of weakly learning the cosine distribution, per Definition 2.1. This learning problem is the central subject of our analysis. Hence, we slightly abuse notation and denote the corresponding cosine function class by

$$\mathcal{F}_\gamma = \{\cos(2\pi\gamma\langle w, x\rangle) \mid w \in S^{d-1}\}. \tag{4}$$

**Continuous Learning with Errors (CLWE) [13].** We define the CLWE distribution $A_{w,\beta,\gamma}$ on dimension $d$ with frequency $\gamma = \gamma(d) \geq 0$, and noise rate $\beta = \beta(d) \geq 0$ to be the distribution of i.i.d. samples of the form $(x, z) \in \mathbb{R}^d \times [-1/2, 1/2)$ where $x \sim N(0, I_d), \xi \sim N(0, \beta)$ and

$$z = \gamma\langle x, w\rangle + \xi \mod 1 . \tag{5}$$

Note that for the $\mod 1$ operation, we take the representatives in $[-1/2, 1/2)$. The CLWE problem consists of detecting between i.i.d. samples from the CLWE distribution or an appropriate null distribution. In the context of CLWE, we refer to the distribution $N(0, I_d) \times U([-1/2, 1/2))$ as the

null distribution and denote it by $A_0$. Given $\gamma = \gamma(d)$ and $\beta = \beta(d)$, we consider a sequence of decision problems $\{\text{CLWE}_{\beta,\gamma}\}_{d\in\mathbb{N}}$, indexed by the input dimension $d$, in which the learner is given samples from an unknown distribution $D$ such that either $D \in \{A_{w,\beta,\gamma} \mid w \in S^{d-1}\}$, and $D = A_0$. The algorithm is asked to decide whether $D \in \{A_{w,\beta,\gamma} \mid w \in S^{d-1}\}$ or $D = A_0$ in polynomial-time. Under this setup, we define the *advantage* of an algorithm as the difference between the probability that it correctly detects samples from $D \in \{A_{w,\beta,\gamma} \mid w \in S^{d-1}\}$, and the probability that errs (decides "$D \neq A_0$") given samples from $D = A_0$. We call the advantage *negligible* if it decays superpolynomially. For a more detailed setup of this problem, we refer the reader to Appendix B.

Bruna et al. [13] showed worst-case evidence that the CLWE problem is computationally hard even with inverse polynomial noise rate $\beta$ if $\gamma \geq 2\sqrt{d}$, despite its seemingly mild requirement of non-negligible advantage. In fact, their evidence of computational intractability is based on *worst-case* lattice problems called the Gap Shortest Vector Problem (GapSVP) [42]. In particular, they showed that distinguishing a *typical* CLWE distribution, where the randomness is over the uniform choice of hidden direction $w \in S^{d-1}$, from the null distribution is as hard as solving the *worst* instance of GapSVP. For a formal definition of the GapSVP, we refer the reader to Appendix B, but note that the (quantum) worst-case hardness of this lattice problems is widely-believed by the cryptography community [43] (See Conjecture 2.3).

**Theorem 2.2** ([13, Corollary 3.2]). *Let $\beta = \beta(d) \in (0,1)$ and $\gamma = \gamma(d) \geq 2\sqrt{d}$ such that $\gamma/\beta$ is polynomially bounded. Then, there is a polynomial-time quantum reduction from $O(d/\beta)$-GapSVP to $\text{CLWE}_{\beta,\gamma}$.*

**Conjecture 2.3** ([43, Conjecture 1.2]). *There is no polynomial-time quantum algorithm that solves GapSVP to within polynomial factors.*

**Weak learning and parameter recovery.** Recall that every element of the function class $\mathcal{F}_\gamma$ is fully characterized by the hidden unit vector $w \in S^{d-1}$. Hence, one possible strategy towards achieving weak learning of the cosine distribution, could be to recover the vector $w$ from samples of the form (3). The following lemma (proven in Appendix I) shows that given any $w'$ sufficiently close to $w$ one can construct an hypothesis that weakly learns the function $f(x) = \cos(2\pi\gamma\langle w, x\rangle)$.

**Proposition 2.4.** *Suppose $\gamma = \omega(1)$. For any $w' \in S^{d-1}$ with $\min\{\|w - w'\|_2^2, \|w + w'\|_2^2\} \leq 1/(16\pi^2\gamma^2)$, the functions $h_A(x) = \cos(2\pi\gamma\langle A, x\rangle)$, $A \in \{w', w\}$ satisfy for large values of $d$ that*

$$\mathbb{E}_{x\sim N(0,I_d)}[\ell((h_w(x), h_{w'}(x))] \leq \text{Var}_{x\sim N(0,I_d)}[(h_w(x))^2] - 1/12.$$

**The LLL algorithm and integer relation detection.** In our algorithmic result, we make use of an appropriate integer relation detection application of the celebrated lattice basis reduction LLL algorithm [40]. We say that for some $b \in \mathbb{R}^n$ the vector $m \in \mathbb{Z}^n \setminus \{0\}$ is an *integer relation* for $b$ if $\langle m, b\rangle = 0$. We make use of the following theorem, and we refer the interested reader to the Appendix E for a complete proof and intuition behind the result.

**Theorem 2.5.** *Let $n, N \in \mathbb{N}$. Suppose $b \in (2^{-N}\mathbb{Z})^n$ with $b_1 = 1$. Let also $m \in \mathbb{Z}^n$ be an integer relation of $b$. Then an appropriate application of the LLL algorithm with input $b$ outputs an integer relation $m' \in \mathbb{Z}^n$ of $b$ with $\|m'\|_2 = O(2^{n/2}\|m\|_2\|b\|_2)$ in time polynomial in $n, N$ and $\log(\|m\|_\infty\|b\|_\infty)$.*

**Notation.** Let $\mathbb{Z}$ denote the set of integers and $\mathbb{R}$ denote the set of real numbers. For $a \in \mathbb{R}$, We use $\mathbb{Z}_{\geq a}$ and $\mathbb{R}_{\geq a}$ for the set of integers at least equal to $a$, and for the set of real numbers at least equal to $a$, respectively. We denote by $\mathbb{N} = \mathbb{Z}_{\geq 1}$ the set of natural numbers. For $k \in \mathbb{N}$ we set $[k] := \{1, 2, \ldots, k\}$. For $d \in \mathbb{N}$, $1 \leq p < \infty$ and any $x \in \mathbb{R}^d$, $\|x\|_p$ denotes the $p$-norm $(\sum_{i=1}^d |x_i|^p)^{1/p}$ of $x$, and $\|x\|_\infty$ denotes $\max_{1\leq i\leq d} |x_i|$. Given two vectors $x, y \in \mathbb{R}^d$ the Euclidean inner product is $\langle x, y\rangle := \sum_{i=1}^d x_i y_i$. By $\log : \mathbb{R}^+ \to \mathbb{R}$ we refer the natural logarithm with base $e$. For $x \in \mathbb{Z}$ and $N \in \mathbb{N}$ we denote by $(x)_N := \text{sgn}(x)\lfloor 2^N x\rfloor/2^N$. Throughout the paper we use the standard asymptotic notation, $o, \omega, O, \Theta, \Omega$ for comparing the growth of two positive sequences $(a_d)_{d\in\mathbb{N}}$ and $(b_d)_{d\in\mathbb{N}}$: we say $a_d = \Theta(b_d)$ if there is an absolute constant $c > 0$ such that $1/c \leq a_d/b_d \leq c$; $a_d = \Omega(b_d)$ or $b_d = O(a_d)$ if there exists an absolute constant $c > 0$ such that $a_d/b_d \geq c$; and $a_d = \omega(b_d)$ or $b_d = o(a_d)$ if $\lim_d a_d/b_d = 0$. We say $x = \text{poly}(d)$ if for some $0 \leq q < r$ it holds $\Omega(d^q) = x = O(d^r)$.

# 3 Main Results

In this section we present our main results towards understanding the fundamental hardness of (weakly) learning the single cosine neuron class given by (4). We present our results in terms of signal to noise ratio (SNR) equal to $1/\beta$, where recall that $\beta > 0$ is an upper bound on the level of adversarial noise $\xi$ one may introduce at the samples given by (3). All proofs of the statements are deferred to the appendices of each subsection.

**A key correspondence.** At the heart of our main results are the following simple elementary equalities that hold for all $v \in \mathbb{R}$, and may help the intuition of the reader.

$$\cos(2\pi(v \mod 1)) = \cos(2\pi v) \tag{6}$$

$$\arccos(\cos(2\pi v)) = 2\pi |v \mod 1| , \tag{7}$$

where in Eq (7), we recall that our $\mod 1$ operation takes representatives in $[-1/2, 1/2)$.

An immediate outcome of these equalities, is a key correspondence between the labels of cosine samples and "phaseless" CLWE samples, where we reminder the reader that the notion of a CLWE sample is defined in (5). By (6), applying the cosine function to CLWE labels results in the cosine distribution with the same frequency, and hidden direction. Conversely, by (7), applying $\arccos$ to cosine labels results in an arguably harder variant of CLWE, in which the $(\mod 1)$-signs of the labels are dropped, with again the same frequency and hidden direction. We say this "phaseless" variant of CLWE is harder than CLWE as we can trivially take the absolute value of CLWE labels to obtain these phaseless CLWE samples, and so an algorithm for solving phaseless CLWE automatically implies an algorithm for CLWE.

We have ignored the issue of additive noise for the sake of simplicity in the above discussion. Indeed, the amount of noise in the samples is a key quantity for characterizing the difficulty of these learning problems and the main technical challenge in carrying the reduction between learning single cosine neurons and CLWE. In subsequent sections, we carefully analyze the interplay between the noise level and the computational difficulty of these learning problems.

## 3.1 The Information-Theoretically Possible Regime: Small Constant Noise

Before discussing the topic of computational hardness, we address the important first question of identifying the noise levels $\beta$ under which *some estimator*, running in potentially exponential time, can weakly learn the class of interest from polynomially many samples. Note that any constant level of noise above 1, that is $\beta \geq 1$, would make learning impossible for trivial reasons. Indeed, as the cosine takes values in $[-1, 1]$ if $\beta \geq 1$ all the labels $z_i$ can be transformed to the uninformative 0 value because of the adversarial noise. One can naturally wonder whether any estimator can succeed at the presence of some constant noise level $\beta \in (0, 1)$. In this section, we establish that for sufficiently small but constant $\beta > 0$ weak learning is indeed possible with polynomially many samples by running an appropriate exponential-time estimator.

Towards establishing this result, we leverage Proposition 2.4, according to which to achieve weak lernability it suffices to construct an estimator that achieves $\ell_2$ recovery of $w$ or $-w$ with an $\ell_2$ error $O(1/\gamma)$. For this reason, we build an exponential-time algorithm that achieves this $\ell_2$ guarantee.

**Theorem 3.1** (Information-theoretic upper bound). *For some constants $c_0, C_0 > 0$ (e.g. $c_0 = 1 - \cos(\frac{\pi}{200}), C_0 = 40000$) the following holds. Let $d \in \mathbb{N}$ and let $\gamma = \gamma(d) > 1$, $\beta(d) \leq c_0$, and $\tau = \arccos(1 - \beta)/(2\pi)$. Moreover, let $P$ be data distribution given by (3) with frequency $\gamma$, hidden direction $w$, and noise level $\beta$. Then, there exists an $\exp(O(d \log(\gamma/\tau)))$-time algorithm using $O(d \log(\gamma/\tau))$ i.i.d. samples from $P$ that outputs a direction $\hat{w} \in S^{d-1}$ satisfying $\min\{\|\hat{w} - w\|_2^2, \|\hat{w} + w\|_2^2\} \leq C_0 \tau^2/\gamma^2$ with probability $1 - \exp(-\Omega(d))$.*

The following corollary follows immediately from Theorem 3.1, Proposition 2.4 and the elementary identity that $\arccos(1 - \beta) = \Theta(\sqrt{\beta})$ for small $\beta$.

**Corollary 3.2.** *Under the assumptions of Theorem 3.1 there exists some sufficiently small $c_1 > 0$, such that if $\beta \leq c_1$ there exist a $\exp(O(d \log(\gamma/\beta)))$-time algorithm using $O(d \log(\gamma/\beta))$ i.i.d. samples from $P$ that weakly learns the function class $\mathcal{F}_\gamma$.*

The proof of both Theorem 3.1 and Corollary 3.2 can be found in Appendix C.

## 3.2 The Cryptographically Hard Regime: Polynomially Small Noise

Given the results in the previous subsection, we discuss now whether a polynomial-time algorithm can achieve weak learnability of the class $\mathcal{F}_\gamma$ for some noise level $\beta$ smaller than an inverse polynomial quantity in $d$, which we call an inverse-polynomial edge, per Definition 2.1. We answer this in the negative by showing a reduction from CLWE to the problem of weakly learning $\mathcal{F}_\gamma$ to any inverse-polynomial edge. This implies that a polynomial-time algorithm for weakly learning $\mathcal{F}_\gamma$ would yield polynomial-time quantum attacks against worst-case lattice problems, which are widely believed to be hard against quantum computers. As mentioned in the introduction, our reduction applies with any 1-periodic and $O(1)$-Lipschitz activation $\phi$. We defer the proofs to Appendix D.

**Theorem 3.3.** *Let $d \in \mathbb{N}$, $\gamma = \omega(\sqrt{\log d}), \beta = \beta(d) \in (0, 1)$. Moreover, let $L > 0$, let $\phi : \mathbb{R} \to [-1, 1]$ be an $L$-Lipschitz 1-periodic univariate function, and $\tau = \tau(d)$ be such that $\beta/(L\tau) = \omega(\sqrt{\log d})$. Then, a polynomial-time (improper) algorithm that weakly learns the function class $\mathcal{F}_\gamma^\phi = \{f_{\gamma,w}(x) = \phi(\gamma\langle w, x\rangle) \mid w \in \mathcal{S}^{d-1}\}$ over Gaussian inputs $x \overset{i.i.d.}{\sim} N(0, I_d)$ under $\beta$-bounded adversarial noise implies a polynomial-time algorithm for $\mathrm{CLWE}_{\tau,\gamma}$.*

By the hardness of CLWE (Theorem 2.2) and our Theorem 3.3, we can immediately deduce the cryptographic hardness of learning the single cosine neuron under inverse polynomial noise.

**Corollary 3.4.** *Let $d \in \mathbb{N}$, $\gamma = \gamma(d) \geq 2\sqrt{d}$ and $\tau = \tau(d) \in (0, 1)$ be such that $\gamma/\tau = \mathsf{poly}(d)$, and $\beta = \beta(d)$ be such that $\beta/\tau = \omega(\sqrt{\log d})$. Then, a polynomial-time algorithm that weakly learns the cosine neuron class $\mathcal{F}_\gamma$ under $\beta$-bounded adversarial noise implies a polynomial-time quantum algorithm for $O(d/\tau)$-GapSVP.*

**Remark 3.5** (Robust learning under position-dependent random noise is hard)**.** *Robustness against advesarial noise in Theorem 3.3 is not necessary for computational hardness. In fact, the reduction only requires robustness against a certain position-dependent random noise. More precisely, for a fixed hidden direction $w \in S^{d-1}$, the random noise $\tilde{\xi}$ is given by $\tilde{\xi} = \phi(\gamma\langle w, x\rangle + \xi) - \phi(\gamma\langle w, x\rangle)$, where $x \sim N(0, I_n)$ and $\xi \sim N(0, \beta)$.*

## 3.3 The Polynomial-Time Possible Regime: Exponentially Small Noise

In this section, in sharp contrast with the previous section, we design and analyze a novel polynomial-time algorithm which provably weakly learns the single cosine neuron with only $d + 1$ samples, when the noise is exponentially small. The algorithm is based on the celebrated lattice basis reduction LLL algorithm and its specific application obtaining the integer relation detection guarantee described in Theorem 2.5. Let us also recall from notation that for a real number $x$ and $N \in \mathbb{Z}_{>1}$, we denote by $(x)_N := \mathrm{sgn}(x)\lfloor 2^N x\rfloor/2^N$. We establish the following result, proved in Appendix E.

**Theorem 3.6.** *Suppose that $1 \leq \gamma \leq d^Q$ for some fixed $Q > 0$, and $\beta \leq \exp(-(d\log d)^3)$. Then Algorithm 1 with input $(x_i, z_i)_{i=1,...,d+1}$ i.i.d. samples from (3) with frequency $\gamma$, hidden direction $w$ and noise level $\beta$, outputs $w' \in S^{d-1}$ with*

$$\min\{\|w' - w\|_2, \|w' + w\|_2\} = O\left(\frac{\beta}{\gamma}\right) = \frac{1}{\gamma}\exp(-\Omega((d\log d)^3)),$$

*and terminates in $\mathsf{poly}(d)$ steps, with probability $1 - \exp(-\Omega(d))$. Moreover, if the algorithm skips the last normalization step, the output $w' \in \mathbb{R}^d$ satisfies $\min\{\|w' - \gamma w\|_2, \|w' + \gamma w\|_2\} = O(\beta)$.*

In particular, by combining our result with Proposition 2.4, one concludes the following result.

**Corollary 3.7.** *Suppose that $\omega(1) = \gamma = \mathsf{poly}(d)$ and $\beta \leq \exp(-(d\log d)^3)$. Then there exists a polynomial-in-$d$ time algorithm using $d + 1$ samples from a single cosine neuron distribution (3), with frequency $\gamma$ and noise level $\beta$, that weakly learns the function class $\mathcal{F}_\gamma$.*

*Proof sketch of Theorem 3.6.* For the purposes of the sketch let us focus on the noiseless case, explaining at the end how an exponentially small tolerance is possible. In this setting, we receive $m$ samples of the form $z_i = \cos(2\pi\langle w, x_i\rangle), i \in [m]$. The algorithm then uses the arcosine and obtains the "phaseless" CLWE values $\tilde{z}_i$ which according to (7) satisfy for some *unknown* $\epsilon_i \in \{-1, 1\}, K_i \in \mathbb{Z}$ $\langle w, x_i\rangle = \epsilon_i\tilde{z}_i + K_i$. Notice that if we knew the integer values of $\epsilon_i, K_i$, since we know $\tilde{z}_i$, the problems becomes simply solving a linear system for $w$. The algorithm then leverages the application of the powerful LLL algorithm to perform integer relation detection and identify the values of $\epsilon_i, K_i$, as stated in Theorem 2.5. The way it does it is as follows. It first finds coefficients $\lambda_i, i = 1, 2, \ldots, d+1$

**Algorithm 1:** LLL-based algorithm for learning the single cosine neuron.

---

**Input:** i.i.d. noisy $\gamma$-single cosine neuron samples $\{(x_i, z_i)\}_{i=1}^{d+1}$.
**Output:** Unit vector $\hat{w} \in S^{d-1}$ such that $\min(\|\hat{w} - w\|, \|\hat{w} + w\|) = \exp(-\Omega((d\log d)^3))$.

---

**for** $i = 1$ **to** $d + 1$ **do**
    $z_i \leftarrow \mathrm{sgn}(z_i) \cdot \min(|z_i|, 1)$
    $\tilde{z}_i = \arccos(z_i)/(2\pi) \mod 1$

Construct a $d \times d$ matrix $X$ with columns $x_2, \ldots, x_{d+1}$, and let $N = d^3(\log d)^2$.
**if** $\det(X) = 0$ **then**
    **return** $\hat{w} = 0$ and output FAIL

Compute $\lambda_1 = 1$ and $\lambda_i = \lambda_i(x_1, \ldots, x_{d+1})$ given by $(\lambda_2, \ldots, \lambda_{d+1})^\top = X^{-1}x_1$.
Set $M = 2^{3d}$ and $\tilde{v} = ((\lambda_2)_N, \ldots, (\lambda_{d+1})_N, (\lambda_1 z_1)_N, \ldots, (\lambda_{d+1} z_{d+1})_N, 2^{-N}) \in \mathbb{R}^{2d+2}$
Output $(t_1, t_2, t) \in \mathbb{Z}^{d+1} \times \mathbb{Z}^{d+1} \times \mathbb{Z}$ from running the LLL basis reduction algorithm on the
  lattice generated by the columns of the following $(2d+3) \times (2d+3)$ integer-valued matrix,

$$\left( \begin{array}{c|c} M2^N(\lambda_1)_N & M2^N\tilde{v} \\ \hline 0_{(2d+2)\times 1} & I_{(2d+2)\times(2d+2)} \end{array} \right)$$

Compute $g = \gcd(t_2)$, by running Euclid's algorithm.
**if** $g = 0 \vee (t_2/g) \notin \{-1, 1\}^{d+1}$ **then**
    **return** $\hat{w} = 0$ and output FAIL
$\hat{w} \leftarrow \mathrm{SolveLinearEquation}(w', X^\top w' = (t_2/g)z + (t_1/g))$
**return** $\hat{w}/\|\hat{w}\|$ and output SUCCESS.

---

such that $\sum_{i=1}^{d+1} \lambda_i x_i = 0$ which can be easily computed because we have $d + 1$ vectors in $\mathbb{R}^d$. Then using the definition of $\tilde{z}_i$, the relation between the coefficient implies the identity

$$\sum_{i=1}^{d+1} \epsilon_i \lambda_i \tilde{z}_i + \sum_{i=1}^{d+1} K_i \lambda_i = \sum_{i=1}^{d+1} \lambda_i \langle x_i, w \rangle = 0. \tag{8}$$

In particular, the $\epsilon_i, K_i$ are coefficients in an integer relation connecting the *known* numbers $\lambda_i z_i, \lambda_i, i = 1, 2, \ldots, d + 1$. Now, an issue is that as one cannot enter the real numbers as input for the lattice-based LLL, the algorithm truncates the numbers to the first $N$ bits and then hope that post-truncation *all the near-minimal integer relations* between these truncated numbers remain a (small multiple of) $\epsilon_i, K_i$, a sufficient condition so that LLL can identify them based on Theorem 2.5. We establish that indeed this the case and this is the most challenging part of the argument. The argument is based on some careful application of the anticoncentration properties of low-degree polynomials (notice that the $\lambda_i$ are rational functions of $x_i$ by Cramer's rule), to deduce that the numbers $\lambda_i, \lambda_i z_i$ are in "sufficient general position", in terms of rational independence, for the argument to work. We remark that this is a potentially important technical advancement over the prior applications of the LLL algorithm towards performing such inference tasks, such as for average-case subset sum problems [38, 22] or regression with discrete coefficients [60, 26] where the corresponding $\lambda_i, \lambda_i z_i$ coefficients are (truncated) i.i.d. continuous random variables in which case anticoncentration is immediate (see e.g. [60, Theorem 2.1]). The final step is to prove that the algorithm is able to tolerate some noise level. We establish that indeed if $N = \tilde{\Theta}(d^3)$ then indeed the argument can still work and tolerate $\exp(-\tilde{\Theta}(d^3))$-noise by showing that the near-minimal integer relations remain unchanged under this level of exponentially small noise. $\qquad\square$

**Remark 3.8** (CLWE with exponentially small noise). *Notice that the detection problem in CLWE (5) reduces to the cosine learning problem (3). Indeed, if $\check{z} = \gamma\langle x, w \rangle + \check{\xi} \mod 1 \in [-1/2, 1/2)$ is a CLWE sample, then $z = \cos(\check{z})$ satisfies*

$$z = \cos(2\pi\gamma\langle x, w \rangle) + \xi,$$

*with $|\xi| \le 2\pi\gamma|\check{\xi}|$. Algorithm 1 and the associated analysis Theorem 3.6 thus improve upon the exact CLWE recovery of [13, Section 6] in two aspects: (i) it requires $d + 1$ samples as opposed to $d^2$; and (ii) it tolerates exponentially small noise.*

## Acknowledgments and Disclosure of Funding

We thank Oded Regev, Ohad Shamir, Lenka Zdeborová, Antoine Maillard, and Afonso Bandeira for providing helpful comments. We also thank Daniel Hsu for pointing out the relevant prior work [6]. MS and JB are partially supported by the Alfred P. Sloan Foundation, NSF RI-1816753, NSF CAREER CIF-1845360, and NSF CCF-1814524. IZ is supported by the CDS Moore-Sloan Postdoctoral Fellowship.

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
