# Appendix

## Table of Contents

# A Future Directions

Our results heavily rely on the specific nature of the periodic activation function, so a natural question is to which extent our results can be extended beyond the single periodic neuron class.

- For lower bounds, a challenging but very interesting generalization would be to establish the cryptographic-hardness of learning certain family of GLMs whose activation function does not need to be periodic. A potentially easier route forward on this direction, would be to consider the Hermite decomposition of the activation function, similar to [A3], and establish lower bounds on the performance of low-degree methods [A23], of SGD [A3], or of local search methods methods [A15], for activation functions whose low-degree Hermite coefficients are exponentially small.

- For upper bounds, we believe that our proposed LLL-based algorithm may be extended beyond learning even periodic activation functions, such as the cosine activation, by appropriately post-processing the measurements, but leave this for future work. Furthermore, it would be interesting to better understand (empirically or analytically) the noise tolerance of our LLL-based algorithm for "low-frequency" activation functions, such as the absolute value underlying the phase retrieval problem which has "zero" frequency.

# B Formal Setup

In this section, we present the formal definitions of all problems required to state our hardness result (Theorem 2.2). We begin with a description of average-case decision problems, of which the CLWE decision problem is a special instance [A6].

## B.1 Average-Case Decision Problems

We introduce the notion of average-case decision problems (or simply binary hypothesis testing problems), based on [A17], where we refer the interested reader for more details. In such average-case decision problems the statistician receives $m$ samples from either a distribution $D$ or another distribution $D'$, and needs to decide based on the produced samples whether the generating distribution is $D$ or $D'$. We assume that the statistician may use any, potentially randomized, algorithm $\mathcal{A}$ which is a measurable function of the $m$ samples and outputs the Boolean decision $\{\text{YES}, \text{NO}\}$ corresponding to their prediction of whether $D$ or $D'$ respectively generated the observed samples. Now, for any Boolean-valued algorithm $\mathcal{A}$ examining the samples, we define the *advantage* of $\mathcal{A}$ solving the decision problem, as the sequence of positive numbers

$$\left| \mathbb{P}_{x \sim D^{\otimes m}}[\mathcal{A}(x) = \text{YES}] - \mathbb{P}_{x \sim D'^{\otimes m}}[\mathcal{A}(x) = \text{YES}] \right|.$$

As mentioned above, we assume that the algorithm $\mathcal{A}$ outputs two values "YES" or "NO". Furthermore, the output "YES" means that algorithm $\mathcal{A}$ has decided that the given samples $x$ comes from the distribution $D$, and "NO" means that $\mathcal{A}$ decided that $x$ comes from the alternate distribution $D'$. Therefore, naturally the advantage quantifies by how much the algorithm is performing better than just deciding with probability $1/2$ between the two possibilities.

Our setup requires two standard adjustments to the setting described above. First, in our setup we consider a sequence of distinguishing problems, indexed by a growing (dimension) $d \in \mathbb{N}$, and for every $d$ we receive $m = m(d)$ samples and seek to distinguish between two distributions $D_d$ and $D'_d$. Now, for any sequence of Boolean-valued algorithms $\mathcal{A} = \mathcal{A}_d$ examining the samples, we naturally define the *advantage* of $\mathcal{A}$ solving the sequence of decision problems, as the sequence of positive numbers

$$\left| \mathbb{P}_{x \sim D_d^{\otimes m}}[\mathcal{A}(x) = \text{YES}] - \mathbb{P}_{x \sim D'^{\otimes m}_d}[\mathcal{A}(x) = \text{YES}] \right|.$$

As a remark, notice that any such distinguishing algorithm $\mathcal{A}$ required to terminate in at most time $T = T(d)$, is naturally implying that the algorithm has access to at most $m \leq T$ samples.

Now, as mentioned above, we require another adjustment. We assume that the distributions $D_d, D'_d$ are each generating $m$ samples in two stages: first by drawing a common structure for all samples, unknown to the statistician (also usually called in the statistics literature as a latent variable), which

we call $s$, and second by drawing some additional and independent-per-sample randomness. In CLWE, $s$ corresponds to the hidden vector $w$ chosen uniformly at random from the unit sphere and the additional randomness per sample comes from the Gaussian random variables $x_i$. Now, to appropriately take into account this adjustment, we define the *advantage* of a sequence of algorithms $\mathcal{A} = \{\mathcal{A}_d\}_{d \in \mathbb{N}}$ solving the *average-case* decision problem of distinguishing two distributions $D_{d,s}$ and $D'_{d,s}$ parametrized by $d$ and some latent variable $s$ chosen from some distribution $\mathcal{S}_d$, as

$$\left| \mathbb{P}_{s \sim \mathcal{S}_d, x \sim D_{d,s}^{\otimes m}}[\mathcal{A}(x) = \text{YES}] - \mathbb{P}_{s \sim \mathcal{S}_d, x \sim D'^{\otimes m}_{d,s}}[\mathcal{A}(x) = \text{YES}] \right| .$$

Finally, we say that algorithm $\mathcal{A} = \{\mathcal{A}_d\}_{d \in \mathbb{N}}$ has *non-negligible advantage* if its advantage is at least an inverse polynomial function of $d$, i.e., a function behaving as $\Omega(d^{-c})$ for some constant $c > 0$.

## B.2 Decision and Phaseless CLWE

We now give a formal definition of the decision CLWE problem, continuing the discussion from Section 2. We also introduce the phaseless-CLWE distribution, which can be seen as the CLWE distribution $A_{w,\beta,\gamma}$ defined in (5), with the absolute value function applied to the labels (recall that we take representatives in $[-1/2, 1/2]$ for the mod 1 operation). The Phaseless-CLWE distribution is, at an intuitive level, useful for stating and proving guarantees of our LLL algorithm in the exponentially small noise regime for learning the cosine neuron (See Section 3.3 and Appendix E).

**Definition B.1** (Decision-CLWE). *For parameters $\beta, \gamma > 0$, the average-case decision problem* $\text{CLWE}_{\beta,\gamma}$ *is to distinguish from i.i.d. samples the following two distributions over $\mathbb{R}^d \times [-1/2, 1/2]$ with non-negligible advantage: (1) the CLWE distribution $A_{w,\beta,\gamma}$, per (5), for some uniformly random unit vector $w \in S^{d-1}$ (which is fixed for all samples), and (2) $N(0, I_d) \times U([-1/2, 1/2])$.*

**Phaseless-CLWE.** We define the Phaseless-CLWE distribution on dimension $d$ with frequency $\gamma$, $\beta$-bounded adversarial noise, hidden direction $w$ to be the distribution of the pair $(x, z) \in \mathbb{R}^d \times [0, 1/2]$ where $x \overset{\text{i.i.d.}}{\sim} N(0, I_d)$ and

$$z = \epsilon(\gamma \langle x, w \rangle + \xi) \mod 1 \tag{9}$$

for some $\epsilon \in \{-1, 1\}$ such that $z \geq 0$, and bounded noise $|\xi| \leq \beta$.

## B.3 Worst-Case Lattice Problems

We begin with a definition of a lattice. A *lattice* is a discrete additive subgroup of $\mathbb{R}^d$. In this work, we assume all lattices are full rank, i.e., their linear span is $\mathbb{R}^d$. For a $d$-dimensional lattice $\Lambda$, a set of linearly independent vectors $\{b_1, \ldots, b_d\}$ is called a *basis* of $\Lambda$ if $\Lambda$ is generated by the set, i.e., $\Lambda = B\mathbb{Z}^d$ where $B = [b_1, \ldots, b_d]$. Formally,

**Definition B.2.** *Given linearly independent $b_1, \ldots, b_d \in \mathbb{R}^d$, let*

$$\Lambda = \Lambda(b_1, \ldots, b_d) = \left\{ \sum_{i=1}^{d} \lambda_i b_i : \lambda_i \in \mathbb{Z}, i = 1, \ldots, d \right\}, \tag{10}$$

*which we refer to as the lattice generated by $b_1, \ldots, b_d$.*

We now present a worst-case *decision* problem on lattices called GapSVP. In GapSVP, we are given an instance of the form $(\Lambda, t)$, where $\Lambda$ is a $d$-dimensional lattice and $t \in \mathbb{R}$, the goal is to distinguish between the case where $\lambda_1(\Lambda)$, the $\ell_2$-norm of the shortest non-zero vector in $\Lambda$, satisfies $\lambda_1(\Lambda) < t$ from the case where $\lambda_1(\Lambda) \geq \alpha(d) \cdot t$ for some "gap" $\alpha(d) \geq 1$. Given a decision problem, it is straightforward to conceive of its search variant. That is, given a $d$-dimensional lattice $\Lambda$, approximate $\lambda_1(\Lambda)$ up to factor $\alpha(d)$. Note that the search version, which we call $\alpha$-approximate SVP in the main text, is *harder* than its decision variant, since an algorithm for the search variant immediately yields an algorithm for the decision problem. Hence, the worst-case hardness of decision problems implies the hardness of their search counterparts. We note that GapSVP is known to be NP-hard for "almost" polynomial approximation factors, that is, $2^{(\log d)^{1-\epsilon}}$ for any constant $\epsilon > 0$, assuming problems in NP cannot be solved in quasi-polynomial time [A22, A20]. As mentioned in the introduction of the

---

**Algorithm 2:** Information-theoretic recovery algorithm for learning cosine neurons

---

**Input:** Real numbers $\gamma = \gamma(d) > 1$, $\beta = \beta(d)$, and a sampling oracle for the cosine distribution (3) with frequency $\gamma$, $\beta$-bounded noise, and hidden direction $w$.

**Output:** Unit vector $\hat{w} \in S^{d-1}$ s.t. $\min\{\|\hat{w} - w\|_2, \|\hat{w} + w\|_2\} = O(\arccos(1-\beta)/\gamma)$.

---

Let $\tau = \arccos(1-\beta)/(2\pi)$, $\epsilon = 2\tau/\gamma$, $m = 64d\log(1/\epsilon)$, and let $\mathcal{C}$ be an $\epsilon$-cover of the unit sphere $S^{d-1}$. Draw $m$ samples $\{(x_i, y_i)\}_{i=1}^m$ from the cosine distribution (3).

**for** $i = 1$ **to** $m$ **do**
  $\lfloor$   $z_i = \arccos(y_i)/(2\pi)$

**for** $v \in \mathcal{C}$ **do**
  Compute
  $\lfloor$   $T_v = \frac{1}{m}\sum_{i=1}^m \mathbb{1}\left[|\gamma\langle v, x_i\rangle - z_i \mod 1| \leq 3\tau\right] + \mathbb{1}\left[|\gamma\langle v, x_i\rangle + z_i \mod 1| \leq 3\tau\right]$

**return** $\hat{w} = \arg\max_{v \in \mathcal{C}} T_v$.

---

paper, the problem is strongly believed to be computationally hard (even with quantum computation), for *any* polynomial approximation factor $\alpha(d)$ [A32].

Below we present formal definitions of two of the most fundamental lattice problems, GapSVP and the Shortest Independent Vectors Problem (SIVP). The SIVP problem, similar to GapSVP, is also believed to be computationally hard (even with quantum computation) for *any* polynomial approximation factor $\alpha(d)$. Interestingly, the hardness of CLWE can also be based on the worst-case hardness of SIVP [A6].

**Definition B.3** (GapSVP). *For an approximation factor $\alpha = \alpha(d)$, an instance of $\alpha$-GapSVP is given by an $d$-dimensional lattice $\Lambda$ and a number $t > 0$. In* YES *instances, $\lambda_1(\Lambda) \leq t$, whereas in* NO *instances, $\lambda_1(\Lambda) > \alpha \cdot t$.*

**Definition B.4** (SIVP). *For an approximation factor $\alpha = \alpha(d)$, an instance of $\text{SIVP}_\alpha$ is given by an $d$-dimensional lattice $\Lambda$. The goal is to output a set of $d$ linearly independent lattice vectors of length at most $\alpha \cdot \lambda_d(\Lambda)$.*

## C  Exponential-Time Algorithm: Constant Noise

We provide full details of the proof of Theorem 3.1, restated as Corollary C.5 at the end of this section. The goal of Algorithm 2 is to use $m = \text{poly}(d)$ samples to recover in polynomial-time the hidden direction $w \in S^{d-1}$, in the $\ell_2$ sense. More concretely, the goal is to compute an estimator $\hat{w} = \hat{w}((x_i, z_i)_{i=1,\dots,m})$ for which it holds $\min\{\|\hat{w} - w\|_2^2, \|\hat{w} + w\|_2^2\} = o(1/\gamma^2)$, with probability $1 - \exp(-\Omega(d))$.

We first start with Lemma C.1, which reduces the recovery problem under the cosine distribution (See Eq. (3)) to the recovery problem under the phaseless CLWE distribution (See Appendix B.2). Then, we prove Lemma C.4, which states that there is an exponential-time algorithm for recovering the hidden direction $w \in S^{d-1}$ in Phaseless-CLWE under sufficiently small adversarial noise. Theorem 3.1 follows from Lemmas C.1 and C.4.

**Lemma C.1.** *Assume $\beta \in [0, 1]$. Suppose that one receives a sample $(x, \tilde{z})$ from the cosine distribution on dimension $d$ with frequency $\gamma$ under $\beta$-bounded adversarial noise. Let $\bar{z} := \text{sgn}(\tilde{z})\min(1, |\tilde{z}|)$. Then, the pair $(x, \arccos(\bar{z})/(2\pi) \mod 1)$ is a sample from the Phaseless-CLWE distribution on dimension $d$ with frequency $\gamma$ under $\frac{1}{2\pi}\arccos(1-\beta)$-bounded adversarial noise.*

*Proof.* Recall $\tilde{z} = \cos(2\pi(\gamma\langle w, x\rangle)) + \xi$, for $x \sim N(0, I_d)$ and $|\xi| \leq \beta$. It suffices to show that

$$\frac{1}{2\pi}\arccos(\bar{z}) = \epsilon\gamma\langle w, x\rangle + \xi' \mod 1 \tag{11}$$

for some $\epsilon \in \{-1, 1\}$ and $\xi' \in \mathbb{R}$ with $|\xi'| \leq \frac{1}{2\pi}\arccos(1-\beta)$.

First, notice that we may assume that without loss of generality $\bar{z} = \tilde{z}$. Indeed, assume for now $\tilde{z} > 1$. The case $\tilde{z} < -1$ can be shown with almost identical reasoning. From the definition of $\tilde{z}$, it must hold that $\xi > 0$ and $\tilde{z} \leq 1 + \xi$. Hence

$$\bar{z} = 1 = \cos(2\pi(\gamma\langle w, x\rangle)) + \tilde{\xi}.$$

**Algorithm 3:** Information-theoretic recovery algorithm for learning the Phaseless-CLWE

---

**Input:** Real numbers $\gamma = \gamma(d) > 1$, $\beta = \beta(d)$, and a sampling oracle for the phaseless-CLWE distribution (9) with frequency $\gamma$, $\beta$-bounded noise, and hidden direction $w$.

**Output:** Unit vector $\hat{w} \in S^{d-1}$ s.t. $\min\{\|\hat{w} - w\|_2, \|\hat{w} + w\|_2\} = O(\beta/\gamma)$.

---

Let $\epsilon = 2\tau/\beta$, $m = 64d \log(1/\epsilon)$, and let $\mathcal{C}$ be an $\epsilon$-cover of the unit sphere $S^{d-1}$. Draw $m$ samples $\{(x_i, z_i)\}_{i=1}^m$ from the phaseless CLWE distribution (9).

**for** $v \in \mathcal{C}$ **do**

$\quad$ Compute
$\quad T_v = \frac{1}{m} \sum_{i=1}^m \mathbb{1}\left[|\gamma\langle v, x_i\rangle - z_i \mod 1| \leq 3\beta\right] + \mathbb{1}\left[|\gamma\langle v, x_i\rangle + z_i \mod 1| \leq 3\beta\right]$

**return** $\hat{w} = \arg\max_{v \in \mathcal{C}} T_v$.

---

for $\tilde{\xi} := \xi + 1 - \tilde{z} \in (0, \xi) \subseteq (0, \beta)$. Hence, $(x, \bar{z})$ is a sample from the cosine distribution in dimension $d$ with frequency $\gamma$ under $\beta$-bounded adversarial noise.

Now, given the above observation, to establish (11), it suffices to show that for some $\epsilon \in \{-1, 1\}$, and $K \in \mathbb{Z}$,

$$\left|\frac{1}{2\pi} \arccos(\tilde{z}) - \epsilon\gamma\langle w, x\rangle - K\right| \leq \frac{1}{2\pi} \arccos(1 - \beta) ,$$

or equivalently using that the cosine function is $2\pi$ periodic and even, it suffices to show that

$$|\arccos(\tilde{z}) - \arccos(\cos(2\pi\gamma\langle w, x\rangle))| \leq \arccos(1 - \beta) .$$

The result then follows from the definition of $\tilde{z}$ and the simple calculus Lemma K.7. $\qquad\square$

We will use the following covering number bound for the running time analysis of Algorithm 2, and the proof of Lemma C.4.

**Lemma C.2** ([A42, Corollary 4.2.13]). *The covering number $\mathcal{N}$ of the unit sphere $S^{d-1}$ satisfies the following upper and lower bound for any $\epsilon > 0$*

$$\left(\frac{1}{\epsilon}\right)^d \leq \mathcal{N}(S^{d-1}, \epsilon) \leq \left(\frac{2}{\epsilon} + 1\right)^d . \tag{12}$$

**Remark C.3.** *An $\epsilon$-cover for the unit sphere $S^{d-1}$ can be constructed in time $O(\exp(d \log(1/\epsilon)))$ by sampling $O(N \log N)$ unit vectors uniformly at random from $S^{d-1}$, where we denote by $N = \mathcal{N}(S^{d-1}, \epsilon)$. The termination time gurantee follows from Lemma C.2 and the property holds with probability $1 - \exp(-\Omega(d))$. We direct the reader for a complete proof of this fact in Appendix H.*

Now we prove our main lemma, which states that recovery of the hidden direction in Phaseless-CLWE under adversarial noise is possible in exponential time, when the noise level $\beta$ is smaller than a small constant.

**Lemma C.4** (Information-theoretic upper bound for recovery of Phaseless-CLWE). *Let $d \in \mathbb{N}$ and let $\gamma = \gamma(d) > 1$, and $\beta = \beta(d) \in (0, 1/400)$. Moreover, let $P$ be the Phaseless-CLWE distribution with frequency $\gamma$, $\beta$-bounded adversarial noise, and hidden direction $w$. Then, there exists an $\exp(O(d \log(\gamma/\beta)))$-time algorithm, described in Algorithm 3, using $O(d \log(\gamma/\beta))$ samples from $P$ that outputs a direction $\hat{w} \in S^{d-1}$ satisfying*

$$\min(\|\hat{w} - w\|_2^2, \|\hat{w} + w\|_2^2) \leq 40000\beta^2/\gamma^2 \tag{13}$$

*with probability $1 - \exp(-\Omega(d))$.*

*Proof.* Let $P$ be the Phaseless-CLWE distribution and $w$ be the hidden direction of $P$. We describe first the steps of the Algorithm 3 we use and then prove its correctness.

Let $\epsilon = \beta/\gamma$, and $\mathcal{C}$ be an $\epsilon$-cover of the unit sphere. By Remark C.3, we can construct such an $\epsilon$-cover $\mathcal{C}$ in $O(\exp(d \log(\gamma/\beta)))$ time such that $|\mathcal{C}| \leq \exp(O(d \log(\gamma/\beta)))$. We now draw $m = 36d \log(\gamma/\beta)$ samples $\{(x_i, z_i)\}_{i=1}^m$ from $P$. Now, given these samples and the threshold value

$t = 3\beta$, we compute for each of the $|\mathcal{C}| \leq \exp(O(d\log(\gamma/\beta)))$ directions $v \in \mathcal{C}$ the following counting statistic,

$$T_v := \frac{1}{m}\sum_{i=1}^{m}\left(\mathbb{1}\left[|\gamma\langle v, x_i\rangle - z_i \mod 1| \leq 3\beta\right] + \mathbb{1}\left[|\gamma\langle v, x_i\rangle + z_i \mod 1| \leq 3\beta\right]\right).$$

$T_v$ is simply measuring the fraction of the $z_i$'s falling in a mod 1-width $3\beta$ interval around $\gamma\langle v, x_i\rangle$ or $-\gamma\langle v, x_i\rangle$, accounting for the uncertainty over the sign $\epsilon \in \{-1, 1\}$ in the definition of Phaseless-CLWE. We then suggest our estimator to be $\hat{w} = \arg\max_{v \in \mathcal{C}} T_v$. The algorithm can be clearly implemented in $|\mathcal{C}| \leq \exp(O(d\log(\gamma/\beta)))$ time.

We prove the correctness of our algorithm by establishing (13) with probability $1 - \exp(-\Omega(d))$. We first show that some direction $v \in \mathcal{C}$ which is sufficiently close to $w$ satisfies $T_v \geq \frac{2}{3}$ with probability $1 - \exp(-\Omega(d))$. Indeed, let us consider $v \in \mathcal{C}$ be a direction such that $\|w - v\|_2 \leq \epsilon = \beta/\gamma$. The existence of such a $v$ follows from our definition of $\mathcal{C}$. We denote for every $i = 1, \ldots, m$ by $\epsilon_i \in \{-1, 1\}$ the sign chosen by the $i$-th sample, and

$$\xi_i = z_i - \epsilon_i\gamma\langle w, x_i\rangle \tag{14}$$

the adversarial noise added to the sample per (9). Now notice that the following trivially holds almost surely for $v$,

$$T_v \geq \frac{1}{m}\sum_{i=1}^{m}\mathbb{1}\left[|\gamma\langle v, x_i\rangle - \epsilon_i z_i \mod 1| \leq 3\beta\right].$$

By elementary algebra and using (14) we have $\epsilon_i z_i - \gamma\langle v, x_i\rangle \mod 1 = \gamma\langle w - v, x_i\rangle + \xi_i \mod 1$. Combining the above it suffices to show that

$$\frac{1}{m}\sum_{i=1}^{m}\mathbb{1}\left[|\gamma\langle w - v, x_i\rangle + \xi_i \mod 1| \leq 3\beta\right] \geq \frac{2}{3}. \tag{15}$$

with probability $1 - \exp(-\Omega(d))$.

Now we have

$$\mathbb{P}[|\gamma\langle w - v, x_i\rangle + \xi_i \mod 1| \leq 3\beta] \geq \mathbb{P}[|\gamma\langle w - v, x_i\rangle \mod 1| \leq 2\beta]$$
$$\geq \mathbb{P}[|\gamma\langle w - v, x_i\rangle| \leq 2\beta]$$

using for the first inequality that $\beta$-bounded adversarial noise cannot move points within distance $2\beta$ to the origin to locations with distance larger than $3\beta$ from the origin and for the second the trivial inequality $|a| \geq |a \mod 1|$. Now, notice that $\gamma\langle w - v, x_i\rangle$ is distributed as a sample from a Gaussian (see Definition K.1) with mean 0 and standard deviation at most $\gamma\|v - w\|_2 \leq \gamma\epsilon = \beta$. Hence, we can immediately conclude $\mathbb{P}[|\gamma\langle w - v, x_i\rangle| \leq 2\beta] \geq 3/4$ since the probability of a Gaussian vector falling within 2 standard deviations of the mean is at least 0.95. By a standard application of Hoeffding's inequality, we can then conclude that (15) holds with probability $1 - \exp(-\Omega(m)) = 1 - \exp(-\Omega(d))$.

We now show that with probability $1 - \exp(-\Omega(d))$ for any $v \in \mathcal{C}$ which satisfies $\min(\|v - w\|_2, \|v + w\|_2) \geq 200\beta/\gamma$, it holds $T_v \leq 1/2$. Notice that given the established existence of a $v$ which is $\beta/\gamma$-close to $w$ and satisfies $T_v \geq 2/3$, with probability $1 - \exp(-\Omega(d))$, the result follows. Let $v \in \mathcal{C}$ be a direction satisfying $\|v - w\|_2 \geq 200\beta/\gamma$. Without loss of generality, assume that $\|v - w\|_2 \leq \|v + w\|_2$. Then, using (14) we have $\gamma\langle v, x_i\rangle - z_i = \gamma\langle v - \epsilon_i w, x_i\rangle - \epsilon_i\xi_i \mod 1$ and $\gamma\langle v, x_i\rangle + z_i = \gamma\langle v + \epsilon_i w, x_i\rangle + \epsilon_i\xi_i \mod 1$. Hence, since $\epsilon \in \{-1, 1\}, |\xi_i| \leq \beta$ for all $i = 1, \ldots, m$ we have by a triangle inequality

$$T_v \leq \frac{1}{m}\sum_{i=1}^{m}\left(\mathbb{1}\left[|\gamma\langle v - w, x_i\rangle \mod 1| \leq 4\beta\right] + \mathbb{1}\left[|\gamma\langle v + w, x_i\rangle \mod 1| \leq 4\beta\right]\right).$$

Now by our assumption on $v$ both $\gamma\langle v - w, x_i\rangle$ and $\gamma\langle v + w, x_i\rangle$ are distributed as mean-zero Gaussians with standard deviation at least $\gamma\|w - v\|_2 \geq 200\beta$. Hence, both $\gamma\langle v - w, x_i\rangle \mod 1$ and $\gamma\langle v + w, x_i\rangle \mod 1$ are distributed as periodic Gaussians with width at least $200\beta$ (see Definition K.1). By Claim K.6 and the fact that $\beta < 1/400$,

$$\mathbb{P}[|\gamma\langle v - w, x_i\rangle \mod 1| \leq 4\beta] \leq 16\beta/(400\beta\sqrt{2\pi}) \cdot (1 + 2(1 + (400\beta)^2)e^{-1/(160000\beta^2)}$$

$$\leq 4/(25\sqrt{2\pi}) < \frac{1}{12}.$$

By symmetry the same upper bound holds for $\mathbb{P}[|\gamma\langle v+w, x_i\rangle \mod 1| \le 4\beta]$. Hence,

$$\mathbb{P}_{(x_i, z_i) \sim P}\left[\{|\gamma\langle v-w, x_i\rangle \mod 1| \le 3\beta\} \cup \{|\gamma\langle v+w, x_i\rangle \mod 1 \mod 1| \le 3\beta\}\right] < 1/6.$$

By a standard application of Hoeffding's inequality, we have

$$\mathbb{P}[T_v > 1/2] \le \exp(-m/18) \le \exp(-2d \log(1/\epsilon)),$$

and by the union bound over all $v \in \mathcal{C}$ satisfying $\|v - w\| \ge 200\beta/\gamma$,

$$\mathbb{P}\left[\bigcup_{\|v-w\| \ge 200\beta/\gamma} \{T_v > 1/2\}\right] < |\mathcal{C}| \cdot \exp(-2d\log(1/\epsilon)) = \exp(-\Omega(d)).$$

This completes the proof. □

Finally, we discuss the recovery in terms of samples from the cosine distribution.

**Corollary C.5** (Restated Theorem 3.1). *For some constants $c_0, C_0 > 0$ (e.g., $c_0 = 1 - \cos(\pi/200), C_0 = 40000$) the following holds. Let $d \in \mathbb{N}$ and let $\gamma = \gamma(d) > 1$, $\beta = \beta(d) \le c_0$, and $\tau = \frac{1}{2\pi}\arccos(1-\beta)$. Moreover, let $P$ be the cosine distribution with frequency $\gamma$, hidden direction $w$, and noise level $\beta$. Then, there exists an $\exp(O(d\log(\gamma/\tau)))$-time algorithm, described in Algorithm 2, using $O(d\log(\gamma/\tau))$ i.i.d. samples from $P$ that outputs a direction $\hat{w} \in S^{d-1}$ satisfying $\min\{\|\hat{w}-w\|_2^2, \|\hat{w}+w\|_2^2\} \le C_0\tau^2/\gamma^2$ with probability $1 - \exp(-\Omega(d))$.*

*Proof.* We first define $m = O(d\log(\gamma/\beta))$ reflecting the sample size needed for the algorithm analyzed in Lemma C.4 to work. We then draw $m$ samples $\{(x_i, \tilde{z}_i)\}_{i=1}^m$ from the cosine distribution. From this point Algorithm 2 simply combines the reduction step of Lemma C.1 and then the algorithm described in the proof of Lemma C.4.

Specifically, using Lemma C.1, we can transform our i.i.d. samples to i.i.d. samples from the Phaseless CLWE distribution on dimension $d$ with frequency $\gamma$ under $\frac{1}{2\pi}\arccos(1-\beta)$-bounded adversarial noise. The transformation simply happens by applying the arccosine function to every projected $\tilde{z}_i$, so it takes $O(1)$ time per sample, a total of $O(m)$ steps. We then use the last step of Algorithm 2 and employ Lemma C.4 which analyzes Algorithm 2 to conclude that the output $\hat{w} \in S^{d-1}$ satisfies $\min(\|\hat{w}-w\|^2, \|\hat{w}+w\|^2) \le 40000\tau^2/\gamma^2$ with probability $1 - \exp(-\Omega(d))$. □

# D Cryptographically-Hard Regime: Polynomially-Small Noise

We give a full proof of Theorem 3.3, restated as Theorem D.1 here. Given Theorem 3.3, Corollary 3.4, also restated below as Corollary D.2, follows from the hardness of CLWE [A6].

**Theorem D.1** (Restated Theorem 3.3). *Let $d \in \mathbb{N}$, $\gamma = \omega(\sqrt{\log d})$, $\beta = \beta(d) \in (0, 1)$. Moreover, let $L > 0$, let $\phi : \mathbb{R} \to [-1, 1]$ be an $L$-Lipschitz 1-periodic univariate function, and $\tau = \tau(d)$ be such that $\beta/(L\tau) = \omega(\sqrt{\log d})$. Then, a polynomial-time (improper) algorithm that weakly learns the function class $\mathcal{F}_\gamma^\phi = \{f_{\gamma,w}(x) = \phi(\gamma\langle w, x\rangle) \mid w \in \mathcal{S}^{d-1}\}$ over Gaussian inputs $x \overset{i.i.d.}{\sim} N(0, I_d)$ under $\beta$-bounded adversarial noise implies a polynomial-time algorithm for $\text{CLWE}_{\tau,\gamma}$.*

*Proof.* Recall that a polynomial-time algorithm for $\text{CLWE}_{\tau,\gamma}$ refers to distinguishing between $m$ samples $(x_i, z_i = \gamma\langle w, x_i\rangle + \xi_i \mod 1)_{i=1,2,\ldots,m}$, where $x_i \sim N(0, I_d)$, $\xi_i \sim N(0, \tau)$ and $w \sim U(S^{d-1})$, from $m$ random samples $(x_i, z_i)_{i=1,2,\ldots,m}$, where $y_i \sim U([0, 1])$ with non-negligible advantage over the trivial random guess (See Appendix B.1 and B.2). We refer to the former sampling process as drawing $m$ i.i.d. samples from the CLWE distribution, where from now on we call $P$ for the CLWE distribution, and to the latter sampling process as drawing $m$ i.i.d. samples from the null distribution, which we denote by $Q$. Here, and everywhere in this proof, the number of samples $m$ denotes a quantity which depends polynomially on the dimension $d$.

Let $\epsilon = \epsilon(d) \in (0, 1)$ be an inverse polynomial, and let $\mathcal{A}$ be a polynomial-time learning algorithm that takes as input $m$ samples from $P$, and with probability $2/3$ outputs a hypothesis $h : \mathbb{R} \to \mathbb{R}$ such that $L_P(h) \le L_P(\mathbb{E}[\phi(z)]) - \epsilon$. Since we are using the squared loss, we can assume without loss of generality that $h : \mathbb{R} \to [-1, 1]$ because clipping the output of the hypothesis $h$, i.e.,

$\tilde{h}(x) = \text{sgn}(h) \cdot \max(|h(x)|, 1)$ is always an improvement over $h$ pointwise because the labels are always inside the range $[-1, 1]$.

Let $D$ be an unknown distribution on $2m$ i.i.d. samples, that is equal to either $P$ or $Q$. Our reduction consists of a statistical test that distinguishes between $D = P$ and $D = Q$. Our test is using the (successful in weakly learning $f_{\gamma,w}$ if $D = P$) predictor $h$ returned by $\mathcal{A}$ on (some appropriate function of the first) $m$ out of the $2m$ samples drawn from $D$. Then, we compute the empirical loss of $h$ on the remaining $m$ samples from $D$, and $m$ samples drawn from $Q$, respectively, and test

$$\hat{L}_D(h) \leq \hat{L}_Q(h) - \epsilon/4 . \tag{16}$$

We conclude $D = P$ if $h$ passes the test and $D = Q$ otherwise. The way we prove that this test succeeds with probability $2/3 - o(1)$, is by using the fact that $\mathcal{A}$ outputs a hypothesis $h$ with $\epsilon$-edge with probability $2/3$ when given $m$ samples from $P$ as input. In the following, we now formally prove the correctness of this test.

We first assume $D = P$, and consider the first $m$ samples $(x_i, z_i)_{i=1,\ldots,m}$ drawn from $P$. Now observe the elementary equality that for all $v \in \mathbb{R}$ it holds $\phi(v \mod 1) = \phi(v)$. Hence,

$$\phi(\gamma\langle w, x_i\rangle + \xi_i) = \phi(z_i).$$

Furthermore, notice that by the fact that the $\phi$ is an $L$-Lipschitz function we have

$$\phi(\gamma\langle w, x_i\rangle) + \tilde{\xi}_i = \phi(z_i) \tag{17}$$

for some $\tilde{\xi}_i \in [-L|\xi_i|, L|\xi_i|]$. By Mill's inequality, for all $i = 1, 2, \ldots, m$ we have $\mathbb{P}[|\xi_i| > \beta/L] \leq \sqrt{2/\pi} \exp(-\beta^2/(2L^2\tau^2))$. Since $\beta/(L\tau) = \omega(\sqrt{\log d})$, we conclude that

$$\mathbb{P}[\bigcup_{i=1}^{m} \{|\xi_i| > \beta/L\}] \leq \sqrt{2/\pi} \cdot m \exp(-\beta^2/(8\pi^2\tau^2)) = md^{-\omega(1)} = o(1) ,$$

where the last equality holds because $m$ depends polynomially on $d$. Hence, it holds that

$$|\xi_i'| \leq L|\xi_i| \leq \beta ,$$

for all $i = 1, \ldots, m$ with probability $1 - o(1)$ over the randomsess of $\xi_i, i = 1, 2, \ldots, m$. Combining the above with (17), we conclude that with probability $1 - o(1)$ over $\xi_i$, using our knowledge of $(x_i, z_i)$, we have at our disposal samples from the function $f_{\gamma,w}(x) = \phi(\gamma\langle w, x\rangle)$ corrupted by adversarial noise of magnitude at most $\beta$. Let us write by $\phi(P)$ the data distribution obtained by applying $\phi$ to labels of the samples from $P$, and similarly write $\phi(Q)$ for the null distribution $Q$.

By assumption and the above, given these samples $(x_i, \phi(z_i))_{i=1,2,\ldots,m}$ we have that $\mathcal{A}$ outputs an hypothesis $h : \mathbb{R}^d \to [-1, 1]$ such that for $m$ large enough, with probability at least $2/3$,

$$L_{\phi(P)}(h) \leq L_{\phi(P)}\left(\underset{(x,z)\sim P}{\mathbb{E}}[\phi(z)]\right) - \epsilon,$$

for some $\epsilon = 1/\text{poly}(d) > 0$.

Now, note that by Claim K.6, the marginal distribution of $\phi(\gamma\langle w, x\rangle)$ is $2\exp(-2\pi^2\gamma^2)$-close in total variation distance to the distribution of $\phi(y)$, where $y \sim U([0, 1])$. Moreover, notice that since the loss $\ell$ is continuous, and $h(x), x \in \mathbb{R}^d$ and of course $\phi(z), y \in \mathbb{R}$ both take values in $[-1, 1]$,

$$\underset{(x,y)\in\mathbb{R}^d\times\mathbb{R}}{\sup} \ell(h(x), \phi(y)) \leq \underset{(a,b)\in[-1,1]^d\times[-1,1]}{\sup} \ell(a, b) \leq 4; . \tag{18}$$

Let us denote $c = \mathbb{E}_{(x,y)\sim Q}[\phi(y)]$ for simplicity. Clearly $|c|, |\phi(y)| \leq 1$. Also,

$$|L_{\phi(P)}(c) - L_{\phi(Q)}(c))| = \left|\underset{(x,y)\sim P}{\mathbb{E}}[(\phi(y) - c)^2] - \underset{(x,y)\sim Q}{\mathbb{E}}[(\phi(y) - c)^2]\right|$$

$$\leq \int_{-1}^{1} \phi(y)^2 |P(y) - Q(y)|dy + 2c\int_{-1}^{1} |\phi(y)||P(y) - Q(y)|dy$$

$$\leq (1 + 2|c|)\int_{-1}^{1} |P(y) - Q(y)|dy$$

$$\leq 6 \cdot TV(P_y, Q_y)$$

$$\leq 12\exp(-2\pi^2\gamma^2) .$$

From the above, since $\mathbb{E}_{z\sim P}[\phi(z)]$ is the optimal predictor for $P$ under the squared loss, we deduce

$$L_{\phi(P)}\left(\mathbb{E}_{(x,z)\sim P}[\phi(z)]\right) \leq L_{\phi(P)}\left(\mathbb{E}_{y\sim Q}[\phi(y)]\right) \leq L_{\phi(Q)}\left(\mathbb{E}_{y\sim Q}[\phi(y)]\right) + 12\exp(-2\pi^2\gamma^2)\ .$$

Now since $\mathbb{E}_{y\sim Q}[\phi(y)]$ is the optimal predictor for $Q$ under the squared loss, $L_{\phi(Q)}(\mathbb{E}[\phi(y)]) \leq L_{\phi(Q)}(h)$ for any predictor $h$. In addition, $\exp(-2\pi^2\gamma^2) = o(\epsilon)$ since $\gamma = \omega(\sqrt{\log d})$ and $\epsilon$ is an inverse polynomial in $d$. Hence, for $d$ large enough, with probability at least $2/3$

$$\begin{aligned} L_{\phi(P)}(h) &\leq L_{\phi(P)}(\mathbb{E}[\phi(\gamma\langle w, x\rangle)]) - \epsilon \\ &\leq L_{\phi(Q)}(h) + 12\exp(-2\pi^2\gamma^2) - \epsilon \\ &\leq L_{\phi(Q)}(h) - \epsilon/2\ . \end{aligned} \tag{19}$$

Using the remaining $m$ samples from $P$, we now compute the empirical losses $\hat{L}_{\phi(P)}(h) = \frac{1}{m}\sum_{i=1}^{m}\ell(h(x_i),\phi(z_i))$, and $\hat{L}_{\phi(Q)}(h) = \frac{1}{m}\sum_{i=1}^{m}\ell(h(x_i),\phi(y_i))$, where $(x_i, z_i)$ are drawn from $P$ and $(x_i, y_i)$ are drawn from $Q$. By a standard use of Hoeffding's inequality, and the fact that the loss is bounded based on (18), it follows that

$$|\hat{L}_{\phi(P)}(h) - L_{\phi(P)}(h)| \leq \frac{\epsilon}{8}\ ,$$

with probability $1 - \exp(-\Omega(m))$ and respectively

$$|\hat{L}_{\phi(Q)}(h) - L_{\phi(Q)}(h)| \leq \frac{\epsilon}{8}\ ,$$

with probability $1 - \exp(-\Omega(m))$ for sufficiently large, but still polynomial in $d$, $m$. Combining the last two displayed equations with (19), we have that, for $m$ large enough, with probability at least $2/3 - o(1)$,

$$\hat{L}_{\phi(P)}(h) \leq L_{\phi(P)}(h) + \frac{\epsilon}{8} \leq \hat{L}_{\phi(Q)}(h) - \frac{\epsilon}{4}.$$

Hence, for $m$ large enough, with probability at least $2/3 - o(1)$, the test correctly concludes $D = P$ or $D = Q$ by using the empirical loss $\hat{L}_{\phi(D)}(h)$, and comparing it with the value $\hat{L}_{\phi(Q)}(h) - \epsilon/4$. $\square$

**Corollary D.2** (Restated Corollary 3.4). *Let $d \in \mathbb{N}$, $\gamma = \gamma(d) \geq 2\sqrt{d}$ and $\tau = \tau(d) \in (0,1)$ be such that $\gamma/\tau = \mathrm{poly}(d)$, and $\beta = \beta(d)$ be such that $\beta/\tau = \omega(\sqrt{\log d})$. Then, a polynomial-time algorithm that weakly learns the cosine neuron class $\mathcal{F}_\gamma$ under $\beta$-bounded adversarial noise implies a polynomial-time quantum algorithm for $O(d/\tau)$-GapSVP.*

*Proof.* The cosine function $\phi(z) = \cos(2\pi z)$ is $2\pi$-Lipschitz and 1-periodic. Hence, the result follows from Theorem D.1 with $L = 2\pi$. $\square$

**Remark D.3** (CLWE with subexponentially small noise). *The intermediate regime of subexponentially small noise, which corresponds to the uncharted region between "Crypto-Hard" and "Polynomial-Time Possible" in Figure 1 where $\beta = \exp(-\Theta(d^c))$ for some $c \in (0,1)$, has not been explored in our work. However, we conjecture that this regime is still hard for polynomial-time algorithms. While [A6] did not consider this noise regime for the CLWE problem, given the problem's analogy to the LWE problem [A36], it is plausible that the quantum reduction from CLWE to GapSVP also applies for subexponentially small noise, since the quantum reduction for LWE extends to subexponentially small noise. That is, it is possible that the requirement $\gamma/\beta = \mathrm{poly}(d)$ in Theorem 2.2 can be relaxed, given the high degree of similarity between CLWE and LWE. If this is true, then a polynomial-time algorithm for CLWE with $\gamma \geq 2\sqrt{d}$ and $\beta \in (0,1)$ implies a polynomial-time quantum algorithm for $O(d/\beta)$-GapSVP. Hence, by Theorem 3.3, a polynomial-time algorithm for our setting with subexponentially small noise would yield a "breakthrough" quantum algorithm for GapSVP, since no polynomial-time algorithms are known to achieve subexponential approximation factors of the form $2^{O(d^c)}$ for any constant $c < 1$. In more detail, the best known algorithms for GapSVP are lattice block reductions, such as the Block Korkin-Zolotarev (BKZ) algorithm and its variants [A39, A38, A33], or slide reductions [A14, A1], which actually solve the* harder *search problem. These block reduction algorithms, which can be seen as generalizations of the LLL algorithm, trade-off running time for better SVP approximation factors. However, none is known to achieve approximation factors of the form $2^{O(d^c)}$ for any constant $c < 1$ in polynomial time.*

# E    LLL-based Algorithm: Exponentially Small Noise

In this section we offer the required missing proofs from the Section 3.3.

## E.1    The LLL Algorithm: Background and the Proof of Theorem 2.5

The most crucial component of the algorithm analyzed in this section is an appropriate use of the LLL lattice basis reduction algorithm. The LLL algorithm receives as input $n$ linearly independent vectors $v_1, \ldots, v_n \in \mathbb{Z}^n$ and outputs an integer combination of them with "small" $\ell_2$ norm. Specifically, let us (re)-define the lattice generated by $n$ *integer* vectors as simply the set of integer linear combination of these vectors.

**Definition E.1.** *Given linearly independent $v_1, \ldots, v_n \in \mathbb{Z}^n$, let*

$$\Lambda = \Lambda(v_1, \ldots, v_n) = \left\{ \sum_{i=1}^n \lambda_i v_i : \lambda_i \in \mathbb{Z}, i = 1, \ldots, n \right\} , \tag{20}$$

*which we refer to as the lattice generated by integer-valued $v_1, \ldots, v_n$. We also refer to $(v_1, \ldots, v_n)$ as an (ordered) basis for the lattice $\Lambda$.*

The LLL algorithm is defined to approximately solve the search version of the Shortest Vector Problem (SVP) on a lattice $\Lambda$, given a basis of it. We have already defined decision-SVP in Appendix B.3. We define the search version below for completeness.

**Definition E.2.** *An instance of the algorithmic $\Delta$-approximate SVP for a lattice $\Lambda \subseteq \mathbb{Z}^n$ is as follows. Given a lattice basis $v_1, \ldots, v_n \in \mathbb{Z}^n$ for the lattice, $\Lambda$; find a vector $\widehat{x} \in \Lambda$, such that*

$$\|\widehat{x}\| \leq \Delta \min_{x \in \Lambda, x \neq 0} \|x\| .$$

The following theorem holds for the performance of the LLL algorithm, whose details can be found in [A26].

**Theorem E.3** ([A26]). *There is an algorithm (namely the LLL lattice basis reduction algorithm), which receives as input a basis for a lattice $\Lambda$ given by $v_1, \ldots, v_n \in \mathbb{Z}^n$ which*

*(1)  solves the $2^{\frac{n}{2}}$-approximate SVP for $\Lambda$ and,*

*(2)  terminates in time polynomial in $n$ and $\log \left( \max_{i=1}^n \|v_i\|_\infty \right)$.*

In this work, we use the LLL algorithm for an integer relation detection application.

**Definition E.4.** *An instance of the integer relation detection problem is as follows. Given a vector $b = (b_1, \ldots, b_n) \in \mathbb{R}^n$, find an $m \in \mathbb{Z}^n \setminus \{\mathbf{0}\}$, such that $\langle b, m \rangle = \sum_{i=1}^n b_i m_i = 0$. In this case, $m$ is said to be an integer relation for the vector $b$.*

We now establish Theorem 2.5, by proving following more general result. In particular, Theorem 2.5 follows from the theorem below by choosing $M = 2^{n+1} \|m'\|_2$ and using notation $m$ (used in Theorem 2.5) instead of $m'$ (used in Theorem E.5), and $m'$ (used in Theorem 2.5) instead of $t$ (used in Theorem E.5).

The following theorem, is rigorously showing how the LLL algorithm can be used for integer relation detection. The proof of the theorem, is based upon some key ideas of the breakthrough use of the LLL algorithm to solve the average-case subset sum problem by Lagarias and Odlyzko [A24], and Frieze [A13], and its recent extensions in the context of regression [A44, A16].

**Theorem E.5.** *Let $n, N \in \mathbb{Z}_{>0}$. Suppose $b \in (2^{-N}\mathbb{Z})^n$ with $b_1 = 1$. Let also $m' \in \mathbb{Z}^n$ be an integer relation of $b$, an integer $M \geq 2^{\frac{n+1}{2}} \|m'\|_2$ and set $b_{-1} = (b_2, \ldots, b_n) \in (2^{-N}\mathbb{Z})^{n-1}$. Then running the LLL basis reduction algorithm on the lattice generated by the columns of the following $n \times n$ integer-valued matrix,*

$$B = \left( \begin{array}{c|c} M2^N b_1 & M2^N b_{-1} \\ \hline 0_{(n-1)\times 1} & I_{(n-1)\times(n-1)} \end{array} \right) \tag{21}$$

*outputs $t \in \mathbb{Z}^n$ which*

*(1) is an integer relation for b with $\|t\|_2 \le 2^{\frac{n+1}{2}} \|m'\|_2 \|b\|_2$ and,*

*(2) terminates in time polynomial in $n, N, \log M$ and $\log(\|b\|_\infty)$.*

*Proof.* It is immediate that $B$ is integer-valued and that the determinant of $B$ is $M2^N \ne 0$, and therefore the columns of $B$ are linearly independent. Hence, from Theorem E.3, we have that the LLL algorithm outputs a vector $z = Bt$ with $t \in \mathbb{Z}^n$ such that it holds

$$\|z\|_2 \le 2^{\frac{n}{2}} \min_{x \in \mathbb{Z}^n \setminus \{0\}} \|Bx\|_2. \tag{22}$$

Moreover, it terminates in time polynomial in $n$ and $\log(M2^N \|b_\infty\|_\infty)$ and therefore in time polynomial in $n, N, \log M$ and $\log(\|b\|_\infty)$.

Since $m'$ is an integer relation for $b$ it holds, $Bm' = (0, m'_2, \dots, m'_n)^t$ and therefore

$$\min_{x \in \mathbb{Z}^n \setminus \{0\}} \|Bx\|_2 \le \|Bm'\|_2 \le \|m'\|_2.$$

Hence, combining with (22) we conclude

$$\|z\|_2 \le 2^{\frac{n}{2}} \|m'\|_2. \tag{23}$$

or equivalently

$$\sqrt{(M\langle 2^N b, t\rangle)^2 + \|t_{-1}\|_2^2} \le 2^{\frac{n}{2}} \|m'\|_2, \tag{24}$$

where $t_{-1} := (t_2, \dots, t_n) \in \mathbb{Z}^{n-1}$.

Now notice that since $2^N \langle b, t\rangle = \langle 2^N b, t\rangle \in \mathbb{Z}$ either $2^N \langle b, t\rangle \ne 0$ and the left hand side of (24) is at least $M$, or $2^N \langle b, t\rangle = 0$. Since the former case is impossible given the right hand side of inequality described in (24) and that $M \ge 2^{\frac{n+1}{2}} \|m'\|_2 > 2^{\frac{n}{2}} \|m'\|_2$ we conclude that $2^N \langle b, t\rangle = 0$ or equivalently $\langle b, t\rangle = 0$. Therefore, $t$ is an integer relation for $b$.

To conclude the proof it suffices to show that $\|t\|_2 \le 2^{\frac{n}{2}+1} \|m'\|_2 \|b\|_2$. Now again from (24) and the fact that $t$ is an integer relation for $b$, we conclude that

$$\|t_{-1}\|_2 \le 2^{\frac{n}{2}} \|m'\|_2. \tag{25}$$

But since $\langle b, t\rangle = 0$ and $b_1 = 1$ we have by Cauchy-Schwartz and (24)

$$|t_1| = |\langle t_{-1}, b_{-1}\rangle| \le \|t_{-1}\|_2 \|b_{-1}\|_2 \le 2^{\frac{n}{2}} \|m'\|_2 \|b\|_2.$$

Hence,

$$\|t\|_2 \le \sqrt{2} \max\{2^{\frac{n}{2}} \|m'\|_2 \|b\|_2, 2^{\frac{n}{2}} \|m'\|_2\} \le 2^{\frac{n+1}{2}} \|m'\|_2 \|b\|_2,$$

since $\|b\|_2 \ge |b_1| = 1$. □

## E.2 Towards proving Theorem 3.6: Auxiliary Lemmas

We first repeat the algorithm we analyze here for convenience, see Algorithm 4. Next, we present here three crucial lemmas towards proving the Theorem 3.6. The proofs of them are deferred to later sections, for the convenience of the reader.

**Remark E.6.** *While the main recovery guarantee in Theorem 3.6 is stated in terms of the hidden direction $w \in S^{d-1}$, Algorithm 4 in fact also recovers the vector $\gamma w$ (up to global sign), if one skips the last line of the algorithm, which normalises the output to the unit sphere. Such recovery is shown as a crucial step towards establishing the main result. This stronger recovery will be used for exact phase retrieval (See Appendix F).*

The first lemma establishes that given a small, in $\ell_2$ norm, "approximate" integer relation between real numbers, one can appropriately truncate each number to some sufficiently large number of bits, so that the truncated numbers satisfy a small in $\ell_2$-norm integer relation between them. This lemma is important for the appropriate application of the LLL algorithm, which needs to receive integer-valued input. Recall that for real number $x$ we denote by $(x)_N$ its truncation to its first $N$ bits after zero, i.e. $(x)_N := 2^{-N} \lfloor 2^N x \rfloor$.

**Algorithm 4:** LLL-based algorithm for learning the single cosine neuron (Restated)

---

**Input:** i.i.d. noisy $\gamma$-single cosine neuron samples $\{(x_i, z_i)\}_{i=1}^{d+1}$.
**Output:** Unit vector $\hat{w} \in S^{d-1}$ such that $\min(\|\hat{w} - w\|, \|\hat{w} + w\|) = \exp(-\Omega((d\log d)^3))$.

---

**for** $i = 1$ **to** $d + 1$ **do**
$\quad z_i \leftarrow \text{sgn}(z_i) \cdot \min(|z_i|, 1)$
$\quad \tilde{z}_i = \arccos(z_i)/(2\pi) \mod 1$

Construct a $d \times d$ matrix $X$ with columns $x_2, \ldots, x_{d+1}$, and let $N = d^3(\log d)^2$.
**if** $\det(X) = 0$ **then**
$\quad$ **return** $\hat{w} = 0$ and output FAIL

Compute $\lambda_1 = 1$ and $\lambda_i = \lambda_i(x_1, \ldots, x_{d+1})$ given by $(\lambda_2, \ldots, \lambda_{d+1})^\top = X^{-1}x_1$.
Set $M = 2^{3d}$ and $\tilde{v} = \left((\lambda_2)_N, \ldots, (\lambda_{d+1})_N, (\lambda_1 z_1)_N, \ldots, (\lambda_{d+1}z_{d+1})_N, 2^{-N}\right) \in \mathbb{R}^{2d+2}$
Output $(t_1, t_2, t) \in \mathbb{Z}^{d+1} \times \mathbb{Z}^{d+1} \times \mathbb{Z}$ from running the LLL basis reduction algorithm on the
lattice generated by the columns of the following $(2d + 3) \times (2d + 3)$ integer-valued matrix,

$$\left(\begin{array}{c|c} M2^N(\lambda_1)_N & M2^N\tilde{v} \\ \hline 0_{(2d+2)\times 1} & I_{(2d+2)\times(2d+2)} \end{array}\right)$$

Compute $g = \gcd(t_2)$, by running Euclid's algorithm.
**if** $g = 0 \vee (t_2/g) \notin \{-1, 1\}^{d+1}$ **then**
$\quad$ **return** $\hat{w} = 0$ and output FAIL

$\hat{w} \leftarrow \text{SolveLinearEquation}(w', X^\top w' = (t_2/g)z + (t_1/g))$
**return** $\hat{w}/\|\hat{w}\|$ and output SUCCESS.

---

**Lemma E.7.** *Suppose $n \leq C_0 d$ for some constant $C_0 > 0$ and $s \in \mathbb{R}^n$ satisfies for some $m \in \mathbb{Z}^n$ that $|\langle m, s \rangle| = \exp(-\Omega((d\log d)^3))$. Then for some sufficiently large constant $C > 0$, if $N = \lceil d^3(\log d)^2 \rceil$ there is an $m' \in \mathbb{Z}^{n+1}$ which is equal with $m$ in the first $n$ coordinates, which satisfies that $\|m'\|_2 \leq Cd^{\frac{1}{2}}\|m\|_2$ and is an integer relation for the numbers $(s_1)_N, \ldots, (s_n)_N, 2^{-N}$.*

The proof of Lemma E.7 is in Section K.3.

The following lemma establishes multiple structural properties surrounding $d + 1$ samples from the cosine neuron, of the form $(x_i, z_i), i = 1, \ldots, d+1$ given by (3).

**Lemma E.8.** *Suppose that $\gamma \leq d^Q$ for some constant $Q > 0$. For some hidden direction $w \in S^{d-1}$ we observe $d + 1$ samples of the form $(x_i, z_i), i = 1, \ldots, d+1$ where for each $i$, $x_i$ is a sample from the distribution $N(0, I_d)$, and*

$$z_i = \cos(2\pi(\gamma\langle w, x_i\rangle)) + \xi_i,$$

*for some unknown and arbitrary $\xi_i \in \mathbb{R}$ satisfying $|\xi_i| \leq \exp(-(d\log d)^3)$. Denote by $X \in \mathbb{R}^{d\times d}$ the random matrix with columns given by the $d$ vectors $x_2, \ldots, x_{d+1}$. With probability $1 - \exp(-\Omega(d))$ the following properties hold.*

*(1) $\max_{i=1,\ldots,d+1} \|x_i\|_2 \leq 10\sqrt{d}$.*

*(2) $\min_{i=1,\ldots,d+1} |\sin(2\pi\gamma\langle x_i, w\rangle)| \geq 2^{-d}$.*

*(3) For all $i = 1, \ldots, d+1$ it holds $z_i \in [-1, 1]$ and $z_i = \cos(2\pi(\gamma\langle x_i, w\rangle + \xi_i'))$, for some $\xi_i' \in \mathbb{R}$ with $|\xi_i'| = \exp(-\Omega((d\log d)^3))$.*

*(4) The matrix $X$ is invertible. Furthermore, $\|X^{-1}x_1\|_\infty = O(2^{\frac{d}{2}}\sqrt{d})$.*

*(5) $0 < |\det(X)| = O(\exp(d\log d))$.*

The proof of Lemma E.8 is in Section K.3.

As explained in the description of our main results in Section 3.3, a step of crucial importance is to show that all "near-minimal" integer relations, such as (8), for the (truncated versions of) $\lambda_i, \lambda_i \tilde{z}_i, i = 1, \ldots, d+1$ are "informative". In what follows, we show that the integer relation with

appropriately "small" norm are indeed informative in terms of recovering the unknown $\epsilon_i, K_i$ of (8) and therefore the hidden vector $w$. The following technical lemma is of instrumental importance for the analysis of the algorithm.

**Lemma E.9.** *Suppose that $\gamma \leq d^Q$ for some constant $Q > 0$, and $N = \lceil d^3 (\log d)^2 \rceil$. Let $\xi' \in \mathbb{R}^{d+1}$ be such that $\|\xi'\|_\infty \leq \exp(-(d \log d)^3)$ and $w \in S^{d-1}$. Suppose that for all $(x_i)_{i=1,\dots,d+1}$ are i.i.d. $N(0, I_d)$ and that for each $i = 1, \dots, d+1$ for some $\tilde{z}_i \in [-1/2, 1/2]$ there exist $\epsilon_i \in \{-1, 1\}, K_i \in \mathbb{Z}$ with $|K_i| \leq d^Q$ such that*

$$\gamma \langle w, x_i \rangle = \epsilon_i \tilde{z}_i + K_i - \xi'_i. \tag{26}$$

*Define also $X \in \mathbb{R}^{d \times d}$ the matrix with columns the $x_2, \dots, x_{d+1}$ and set $\lambda_1 = 1$ and $(\lambda_2, \dots, \lambda_{d+1})^t = X^{-1} x_1$. Then with probability $1 - \exp(-\Omega(d))$, any integer relation $t \in \mathbb{Z}^{2d+3}$ between the numbers $(\lambda_1)_N, \dots, (\lambda_{d+1})_N, (\lambda_1 \tilde{z}_1)_N, \dots, (\lambda_{d+1} \tilde{z}_{d+1})_N, 2^{-N}$ with $\|t\|_2 \leq 2^{2d}$ satisfies in the first $2d + 2$ coordinates it is equal to a non-zero integer multiple of $(K_1, \dots, K_{d+1}, \epsilon_1, \dots, \epsilon_{d+1})$.*

The proof of Lemma E.9 is in Section E.4.

## E.3 Proof of Theorem 3.6

We now proceed with the proof of the Theorem 3.6 using the lemmas from the previous sections.

*Proof.* We analyze the algorithm by first analyze it's correctness step by step as it proceeds and then conclude with the polynomial-in-$d$ bound on its termination time.

We start with using part 3 of Lemma E.8 which gives us that $z_i \in [-1, 1]$ with probability $1 - \exp(-\Omega(d))$ for all $i = 1, 2, \dots, d+1$. Therefore the $z_i$'s remain invariant under the operation $z_i \leftarrow \text{sgn}(z_i) \min(|z_i|, 1)$, with probability $1 - \exp(-\Omega(d))$. Furthermore, using again the part 3 of Lemma E.8 the $\tilde{z}_i$'s computed in the second step satisfy

$$\cos(2\pi \tilde{z}_i) = \cos(2\pi (\gamma \langle w, x_i \rangle + \xi'_i))$$

for some $\xi'_i \in \mathbb{R}$ with $|\xi'_i| \leq \exp(-\Omega((d \log d)^3))$. Using the $2\pi$- periodicity of the cosine as well as that it is an even function we conclude that for all for $i = 1, \dots, d+1$ there exists $\epsilon_i \in \{-1, 1\}, K_i \in \mathbb{Z}$ for which it holds for every $i = 1, \dots, d+1$

$$\gamma \langle w, x_i \rangle = \epsilon_i \tilde{z}_i + K_i - \xi'_i. \tag{27}$$

Notice that if we knew the exact values of $\epsilon_i, K_i$, since we already know $x_i, \tilde{z}_i$ the problem would reduce to inverting a (noisy) linear system of $d + 1$ equations and $d$ unknowns. The rest of the algorithm uses an appropriate application of the LLL to learn the values of $\epsilon_i, K_i$ and solve the (noisy) linear system.

Now, notice that using the part 5 of Lemma E.8 with probability $1 - \exp(-\Omega(d))$ the matrix $X$ is invertible and the algorithm is not going to terminate in the second step.

In the following step, the $\lambda_i, i = 1, 2, \dots, d+1$ are given by $\lambda_1 = 1$ and the unique $\lambda_i = \lambda_i(x_1, \dots, x_{d+1}) \in \mathbb{R}, i = 2, \dots, d+1$ satisfying

$$\sum_{i=1}^{d+1} \lambda_i x_i = x_1 + X(\lambda_2, \dots, \lambda_{d+1})^\top = 0.$$

Hence, we conclude that for the unknown direction $w$ it holds

$$\sum_{i=1}^{d+1} \lambda_i \gamma \langle w, x_i \rangle = \gamma \langle w, \sum_{i=1}^{d+1} \lambda_i x_i \rangle = 0.$$

Using now (27) and rearranging the noise terms we conclude

$$\sum_{i=1}^{d+1} \lambda_i \tilde{z}_i \epsilon_i + \sum_{i=1}^{d+1} \lambda_i K_i = \sum_{i=1}^{d+1} \lambda_i \xi'_i. \tag{28}$$

Now using the fourth part of Lemma E.8 and the upper bound on $\|\xi'\|_\infty$ we have with probability $1 - \exp(-\Omega(d))$ that

$$\left|\sum_{i=1}^{d+1}\lambda_i\xi_i'\right| = O(d\|\lambda\|_\infty\|\xi'\|_\infty) = O(d2^{\frac{d}{2}}\sqrt{d}\exp(-\Omega((d\log d)^3))) = \exp(-\Omega((d\log d)^3)).$$

Hence, using (28) we conclude that with probability $1 - \exp(-\Omega(d))$ it holds

$$\left|\sum_{i=1}^{d+1}\lambda_i\bar{z}_i\epsilon_i + \sum_{i=1}^{d+1}\lambda_iK_i\right| = \exp(-\Omega((d\log d)^3)). \tag{29}$$

Define $s \in \mathbb{R}^{2d+2}$ given by $s_i = \lambda_i, i = 1,\ldots,d+1$ and $s_i = \lambda_{i-d-1}\tilde{z}_{i-d-1}, i = d+2,\ldots,2d+2$. Define also $m \in \mathbb{Z}^{2d+2}$ given by $m_i = K_i, i = 1,\ldots,d+1$ and $m_i = \epsilon_{i-d-1}, i = d+1,\ldots,2d+2$. For these vectors, given the above, it holds with probability $1 - \exp(-\Omega(d))$ that $|\langle s, m\rangle| = \exp(-\Omega((d\log d)^3))$ based on (29). Now notice that

$$\max_{i=1,\ldots,d+1}|K_i| = O(\gamma\sqrt{d}) \tag{30}$$

with probability $1 - \exp(-\Omega(d))$. Indeed, from the definition of $K_i$ we have for large enough values of $d$ that $|K_i| \leq \gamma|\langle w, x_i\rangle| + 1 + |\xi_i| \leq \gamma\|x_i\|_2 + 2$. Recall that using part 1 of Lemma E.8 for all $i = 1,\ldots,d+1$ it holds $\|x_i\|_2 = O(\sqrt{d})$ with probability $1 - \exp(-\Omega(d))$. Hence, for all $i$, $|K_i| = O(\gamma\sqrt{d})$, with probability $1 - \exp(-\Omega(d))$. Therefore, since $|\epsilon_i| = 1$ for all $i = 1,\ldots,d+1$ it also holds with probability $1 - \exp(-\Omega(d))$ that $\|m\|_2 = O(d\|K\|_\infty) = O(\gamma d^{\frac{3}{2}})$.

We now employ Lemma E.7 for our choice of $s$ and $m$ to conclude that for the $N$ chosen by the algorithm there exists an integer $m'_{2d+3}$ so that $m' = (m, m'_{2d+3}) \in \mathbb{Z}^{2d+3}$ is an integer relation for $(\lambda_1)_N,\ldots,(\lambda_{d+1})_N,(\lambda_1z_1)_N,\ldots,(\lambda_{d+1}z_{d+1})_N,2^{-N}$ with $\|m'\|_2 = O(d^2\gamma)$.

Now we set $b \in (2^{-N}\mathbb{Z})^{2d+3}$ given by $b_i = (\lambda_i)_N$ for $i = 1,\ldots,d+1$, $b_i = (\lambda_{i-d-1}\tilde{z}_{i-d-1})_N$ for $i = d+2,\ldots,2d+2$, and $b_{2d+3} = 2^{-N}$. Notice that $b_1 = (1)_N = 1$ and furthermore that the $\tilde{v}$ defined by the algorithm satisfies $\tilde{v} = (b_2,\ldots,b_{2d+3})$. On top of this, we have that the $m'$ defined in previous paragraph is an integer relation for $b$ with $\|m'\|_2 = O(d^2\gamma)$. Since $\gamma$ is polynomial in $d$ we have that $2^{\frac{2d+3+1}{2}}\|m'\|_2 \leq 2^{3d}$ for large values of $d$. Hence, to analyze the LLL step of our algorithm we use Theorem E.5 for $n = 2d + 3$, to conclude that the output of the LLL basis reduction step is a $t = (t_1, t_2, t') \in \mathbb{Z}^{d+1} \times \mathbb{Z}^{d+1} \times \mathbb{Z}$ which is an integer relation for $b$ and it satisfies that

$$\|t\|_2 \leq 2^{d+2}\|m'\|_2\|b\|_2,$$

with probability $1 - \exp(-\Omega(d))$.

Now we use part 4 of Lemma E.8 to conclude that $\|\lambda\|_2 \leq d\|\lambda\|_\infty = O(2^{\frac{d}{2}}d^{\frac{3}{2}})$, with probability $1 - \exp(-\Omega(d))$. Since for any real number $x$ it holds $|(x)_N| \leq |x| + 1$ and $\tilde{z}_i \in [-1/2, 1/2]$ for all $i = 1, 2,\ldots,d+1$ we conclude that $\|b\|_2 = O(\|\lambda\|_2) = O(2^{\frac{d}{2}}d^{\frac{3}{2}})$, with probability $1 - \exp(-\Omega(d))$. Furthermore, since $\|m'\| = O(d^2\gamma)$ we conclude that since $\gamma$ is polynomial in $d$, for large values of $d$ it holds,

$$\|t\|_2 = O(2^{\frac{3d}{2}}) \leq 2^{2d}, \tag{31}$$

with probability $1 - \exp(-\Omega(d))$.

We now use the above and (30) to crucially apply Lemma E.9 and conclude that for some non-zero integer multiple $c$ it necessarily holds $(t_1)_i = cK_i$ and $(t_2)_i = c\epsilon_i$, with probability $1 - \exp(-\Omega(d))$. Note that the assumptions of the Lemma can be checked to be satisfied in straightforward manner. Now, the greatest common divisor between the elements of $t_2$ equals either $c$ or $-c$, since the elements of $t_2$ are just $c$-multiples of $\epsilon_i$ which themselves are taking values either $-1$ or $1$. Hence the step of the algorithm using Euclid's algorithm outputs $g$ such that $g = \epsilon c$ for some $\epsilon \in \{-1, 1\}$. In particular, $t_2/g = \epsilon(\epsilon_1,\ldots,\epsilon_{d+1}) \neq 0$ implying that the algorithm does not enter the if-condition branch on the next step.

Finally, since $c = \epsilon g$ it also holds $t_1/g = \epsilon(K_1,\ldots,K_{d+1})$ and therefore the last step of the algorithm is solving the linear equations for $i = 2,\ldots,d+1$ given by

$$\langle x_i, \hat{w}\rangle = \epsilon\left(\epsilon_i\tilde{z}_i + \epsilon K_i\right) = \epsilon\gamma\langle x_i, w\rangle + \epsilon\xi_i',$$

where we have used (27). Hence if $\xi' = (\xi'_2, \ldots, \xi'_{d+1})^t$ we have
$$\hat{w} = \epsilon\gamma w + \epsilon X^{-1}\xi .$$
Hence,
$$\|\hat{w} - \epsilon\gamma w\|_2 \leq \|X^{-1}\xi\|_2 .$$
Now, using standard results on the extreme singular values of $X$, such as [A37, Equation (3.2)], we have that $\sigma_{\max}(X^{-1}) = 1/\sigma_{\min}(X) \leq 2^d$, with probability $1 - \exp(-\Omega(d))$. Hence, with probability $1 - \exp(-\Omega(d))$ it holds
$$\|\hat{w} - \epsilon\gamma w\|_2 \leq O\left(2^{\frac{d}{2}}\|\xi\|_2\right).$$
Now since almost surely $\|\xi\|_2 \leq d\beta$ and $\beta \leq \exp(-(d\log d)^3)$ we have $2^{\frac{d}{2}}\|\xi\|_2 = O(\beta) = \exp(-\Omega((d\log d)^3))$ and therefore, with probability $1 - \exp(-\Omega(d))$ it holds
$$\|\hat{w} - \epsilon\gamma w\|_2 \leq O(\beta) = \exp(-\Omega((d\log d)^3)). \tag{32}$$
Finally, since $|\|x\|_2 - \|x'\|_2| \leq \|x - x'\|_2$ we also have $|\|\hat{w}\|_2 - \gamma| \leq O(\beta) = \exp(-\Omega((d\log d)^3))$ and therefore
$$
\begin{aligned}
\left\|\frac{\hat{w}}{\|\hat{w}\|} - \epsilon w\right\|_2 &= \gamma^{-1}\left\|\frac{\gamma}{\|\hat{w}\|_2}\hat{w} - \epsilon w\gamma\right\|_2 \leq \gamma^{-1}\left(\|\hat{w} - \epsilon\gamma w\|_2 + \frac{\|\hat{w} - \gamma\|_2}{\gamma - |\gamma - \|\hat{w}\|_2|}\right) \\
&\leq \gamma^{-1}\left(\|\hat{w} - \epsilon\gamma w\|_2 + O(\beta)\right) \\
&\leq O\left(\frac{\beta}{\gamma}\right) = \exp(-\Omega((d\log d)^3)),
\end{aligned}
$$
since $\gamma = \omega(\beta)$. Since $\epsilon \in \{-1, 1\}$ the proof of correctness is complete.

For the termination time, it suffices to establish that the step using the LLL basis reduction algorithm and the step using the Euclid's algorithm can be performed in polynomial-in-$d$ time. For the LLL step we use Theorem E.5 to conclude that it runs in polynomial-time in $d, N, \log M$ and $\log\|\lambda\|_\infty$. Now clearly $N, \log M$ are polynomial in $d$. Furthermore, by part 4 of Lemma E.8 also $\log\|\lambda\|_\infty$ is polynomial in $d$ with probability $1 - \exp(-\Omega(d))$. The Euclid's algorithm takes time which is polynomial in $d$ and in $\log\|t_2\|_\infty$. But we have established in (31) that $\|t_2\|_2 \leq \|t\|_2 \leq 2^{2d}$, with probability $1 - \exp(-\Omega(d))$ and therefore the Euclid's algorithm step also indeed requires time which is polynomial-in-$d$. $\qquad\square$

## E.4 Proof of Lemma E.9

We focus this section on proving the crucial Lemma E.9. As mentioned above, the proof of the lemma is quite involved, and, potentially interestingly, it requires the use of anticoncentration properties of the coefficients $\lambda_i$ which are rational function of the coordinates of $x_i$. In particular, the following result is a crucial component of establishing Lemma E.9.

**Lemma E.10.** *Suppose $w \in S^{d-1}$ is an arbitrary vector on the unit sphere and $\gamma \geq 1$. For two sequences of integer numbers $C = (C_i)_{i=1,2,\ldots,d+1}, C' = (C'_i)_{i=1,2,\ldots,d+1}$ we define the polynomial $P_{C,C'}(x_1, \ldots, x_{d+1})$ in $d(d+1)$ variables which equals*
$$\det(x_2, \ldots, x_{d+1})\left(\langle\gamma w, x_1\rangle C_1 + (C')_1\right) \tag{33}$$
$$+ \sum_{i=2}^{d+1}\det(x_2, \ldots, x_{i-1}, -x_1, x_{i+1}, \ldots, x_{d+1})\left(\langle\gamma w, x_i\rangle C_i + (C')_i\right),$$
*where each $x_1, \ldots, x_{d+1}$ is assumed to have a $d$-dimensional vector form.*

*We now draw $x_i$'s in an i.i.d. fashion from the standard Gaussian measure on $d$ dimensions. For any two sequences $C, C'$ it holds*
$$\mathrm{Var}(P_{C,C'}(x_1, \ldots, x_{d+1})) = (d-1)!\gamma^2\sum_{1\leq i<j\leq d+1}(C_i - C_j)^2 + d!\sum_{i=1}^{d+1}(C')_i^2.$$

*Furthermore, for some universal constant $B > 0$ the following holds. If $C_i, C'_i$ are such that either the $C_i$'s are not all equal to each other or the $C'_i$'s are not all equal to zero, then for any $\epsilon > 0$,*
$$\mathbb{P}(|P_{C,C'}(x_1, \ldots, x_{d+1})| \leq \epsilon) \leq B(d+1)\epsilon^{\frac{1}{d+1}}. \tag{34}$$

*Proof.* The second part follows from the first one combined with the fact that under the assumptions on $C, C'$ in holds that for some $i = 1, \ldots, d + 1$ either $(C_i - C_i')^2 \geq 1$ or $(C_i')^2 \geq 1$. In particular, in both cases since $\gamma \geq 1$,

$$\mathrm{Var}(P_{C,C'}(x_1, \ldots, x_{d+1})) \geq (d-1)! \geq 1.$$

Now we employ [A30, Theorem 1.4] (originally proved in [A7]) which implies that for some universal constant $B > 0$, since our polynomial is multilinear and has degree $d + 1$ it holds for any $\epsilon > 0$

$$\mathbb{P}\left(|P_{C,C'}(x_1, \ldots, x_{d+1})| \leq \epsilon\sqrt{\mathrm{Var}(P_{C,C'}(x_1, \ldots, x_{d+1}))}\right) \leq B(d+1)\epsilon^{\frac{1}{d+1}}.$$

Using our lower bound on the variance we conclude the result.

Now we proceed with the variance calculation. First we denote

$$\mu(x_{-1}) := \det(x_2, \ldots, x_{d+1}),$$

and for each $i > 2$

$$\mu(x_{-i}) := \det(x_2, \ldots, x_{i-1}, -x_1, x_{i+1}, \ldots, x_{d+1}).$$

As all coordinates of the $x_i$'s are i.i.d. standard Gaussian, for each $i = 1, \ldots, d + 1$ the random variable $\mu(x_{-i})$ has mean zero and variance $d!$. Furthermore, let us denote $\ell(x_i) := \langle \gamma w, x_i \rangle$, which is a random variable with mean zero and variance $\gamma^2$. In particular $\mu(x_{-i})\ell(x_i)$ has also mean zero as $\mu(x_{-i})$ is independent with $x_i$. Now notice that under this notation,

$$P_{C,C'}(x_1, \ldots, x_{d+1}) = \sum_{i=1}^{d} C_i \mu(x_{-i})\ell(x_i) + \sum_{i=1}^{d} C_i' \mu(x_{-i}).$$

Hence, we conclude

$$\mathbb{E}[P_{C,C'}(x_1, \ldots, x_{d+1})] = 0.$$

Now we calculate the second moment of the polynomial. We have

$$\mathbb{E}[P_{C,C'}^2(x_1, \ldots, x_{d+1})] = \sum_{i=1}^{d+1} C_i^2 d! \gamma^2 + \sum_{1 \leq i \neq j \leq d} C_i C_j \mathbb{E}[\mu(x_{-i})\ell(x_i)\mu(x_{-j})\ell(x_j)] + \sum_{i=1}^{d+1} C_i'^2 d!.$$

Now for all $i \neq j$,

$$\begin{aligned}
&\mathbb{E}[\mu(x_{-i})\ell(x_i)\mu(x_{-j})\ell(x_j)] \\
&= \mathbb{E}[\det(\ldots, x_{i-1}, -x_1, x_{i+1}, \ldots)\det(\ldots, x_{j-1}, -x_1, x_{j+1}, \ldots)\langle \gamma w, x_i \rangle \langle \gamma w, x_j \rangle] \\
&= \sum_{p,q=1}^{d} \gamma^2 w_p w_q \mathbb{E}[\det(\ldots, x_{i-1}, -x_1, x_{i+1}, \ldots)\det(\ldots, x_{j-1}, -x_1, x_{j+1}, \ldots)(x_i)_p (x_j)_q]
\end{aligned}$$

Now observe that the monomials of the product

$$\det(\ldots, x_{i-1}, -x_1, x_{i+1}, \ldots)\det(\ldots, x_{j-1}, -x_1, x_{j+1}, \ldots)(x_i)_p (x_j)_q$$

have the property that each coordinate of the various $x_i's$ appears at most twice; in other words the degree per variable is at most 2. Hence, the monomials that could potentially have not zero mean with respect to the standard Gaussian measure are the ones where all coordinates of every $x_i, i = 1, \ldots, d + 1$ appear exactly twice or none at all, in which case the monomial has mean equal to the coefficient of the monomial. By expansion of the determinants, we have that the studied product of polynomials equals to the sum over all $\sigma, \tau$ permutations on $d$ variables of the terms

$$(-1)^{\mathrm{sgn}(\sigma\tau^{-1})}(\ldots x_{i-1,\sigma(i-1)}(-x_1)_{\sigma(i)}x_{i+1,\sigma i+1} \ldots)(\ldots x_{j-1,\tau(j-1)}(-x_1)_{\tau(j)}x_{j+1,\tau(j+1)} \ldots)(x_i)_p(x_j)_q.$$

Hence, a straightforward inspection allows us to conclude that for every coordinate to appear exactly twice, we need the corresponding permutations $\sigma, \tau$ to satisfy $\tau(i) = p, \sigma(j) = q$ (from the coordinates $(x_i)_p, (x_j)_q$), $\sigma(i) = \tau(j)$ (from the coordinate of $x_1$) and finally $\sigma(x) = \tau(x)$ for all

$x \in [d] \setminus \{i, j\}$ (the rest coordinates). Furthermore, the value of the mean of this monomial would then be given simply by $(-1)^{\mathrm{sgn}(\sigma\tau^{-1})}$.

Now we investigate more which permutations $\sigma, \tau$ can satisfy the above conditions. The last two conditions imply in straightforward manner that $\tau^{-1}\sigma$ is the transposition $(i, j)$. Hence, $\tau^{-1}\sigma(j) = i$. But we have $\sigma(j) = q$ and therefore $i = \tau^{-1}\sigma(j) = \tau^{-1}(q)$ which gives $\tau(i) = q$. We have though as our condition that $\tau(i) = p$ which implies that for such a pair of permutations $\sigma, \tau$ to exist it must hold $p = q$. Furthermore, for any $\sigma$ with $\sigma(j) = p$ there exist a unique $\tau$ satisfying the above given by $\tau = \sigma \circ (i, j)$, where $\circ$ corresponds to the multiplication in the symmetric group $S_d$. Hence, if $p \neq q$ no such pair of permutations exist and the mean of the product is zero. If $p = q$ there are exactly $(d - 1)!$ such pairs (all permutations $\sigma$ sending $j$ to $p$ and $\tau$ given uniquely given $\sigma$) which correspond to $(d - 1)!$ monomials with mean $(-1)^{\mathrm{sgn}(\sigma)+\mathrm{sgn}(\tau)} = (-1)^{\mathrm{sgn}(\sigma^{-1}\tau)} = -1$, where we used that the sign of a transposition is $-1$. Combining the above we conclude that

$$\mathbb{E}[\det(\ldots, x_{i-1}, -x_1, x_{i+1}, \ldots)\det(\ldots, x_{j-1}, -x_1, x_{j+1}, \ldots)(x_i)_p(x_j)_q] = -(d - 1)!\mathbb{1}(p = q).$$

Hence, since $\|w\|_2 = 1$,

$$\mathbb{E}[\mu(x_{-i})\ell(x_i)\mu(x_{-j})\ell(x_j)] = \sum_{p=1}^{d} -\gamma^2 w_p^2 = -\gamma^2.$$

Therefore,

$$\mathbb{E}[P_{C,C'}^2(x_1, \ldots, x_{d+1})] = \sum_{i=1}^{d+1} C_i^2 d! \gamma^2 - (d-1)!\gamma^2 \sum_{1 \leq i \neq j \leq d+1} C_i C_j + \sum_{i=1}^{d+1} C_i'^2 d!$$

$$= (d-1)!\gamma^2 \sum_{1 \leq i < j \leq d+1} (C_i - C_j)^2 + d! \sum_{i=1}^{d+1} (C')_i^2.$$

The proof is complete. $\qquad\square$

We now proceed with the proof of Lemma E.9.

*Proof of Lemma E.9.* Let $t_1, t_2 \in \mathbb{Z}^{d+1}, t' \in \mathbb{Z}$ with $\|(t_1, t_2, t')\|_2 \leq 2^{2d}$ which is an integer relation;

$$\sum_{i=1}^{d+1} (\lambda_i)_N (t_1)_i + \sum_{i=1}^{d+1} (\lambda_i \tilde{z}_i)_N (t_2)_i + t' 2^{-N} = 0.$$

First note that it cannot be the case that $t_1 = t_2 = 0$ as from the integer relation it should be also that $t' = 0$ and therefore $t = 0$ but an integer relation needs to be non-zero. Hence, from now on we restrict ourselves only to the case where $t_1, t_2$ are not both zero. Now, as clearly $|t'| \leq 2^{2d}$ it also holds

$$\left|\sum_{i=1}^{d+1} (\lambda_i)_N (t_1)_i + \sum_{i=1}^{d+1} (\lambda_i \tilde{z}_i)_N (t_2)_i\right| \leq 2^{2d} 2^{-N}.$$

Consider $\mathcal{T}$ the set of all pairs $t = (t_1, t_2) \in (\mathbb{Z}^{d+1} \times \mathbb{Z}^{d+1}) \setminus \{0\}$ for which there does not exist a $c \in \mathbb{Z} \setminus \{0\}$ such that for $i = 1, \ldots, d+1$ $(t_1)_i = cK_i$ and $(t_2)_i = c\epsilon_i$.

To prove our result it suffices therefore to prove that

$$\mathbb{P}\left(\bigcup_{t \in \mathcal{T}, \|t\|_2 \leq 2^{2d}} \left\{\left|\sum_{i=1}^{d+1} (\lambda_i)_N (t_1)_i + \sum_{i=1}^{d+1} (\lambda_i \tilde{z}_i)_N (t_2)_i\right| \leq 2^{2d}/2^N\right\}\right) \leq \exp(-\Omega(d))$$

for which, since for any $x$ it holds $|x - (x)_N| \leq 2^{-N}$ and $\|(t_1, t_2)\|_1 \leq \sqrt{2(d+1)}\|(t_1, t_2)\|_2 \leq 2^{3d}$ for large values of $d$, it suffices to prove that for large enough values of $d$,

$$\mathbb{P}\left(\bigcup_{t \in \mathcal{T}, \|t\|_2 \leq 2^{2d}} \left\{\left|\sum_{i=1}^{d+1} \lambda_i (t_1)_i + \sum_{i=1}^{d+1} \lambda_i \tilde{z}_i (t_2)_i\right| \leq 2^{4d}/2^N\right\}\right) \leq \exp(-\Omega(d)).$$

Notice that by using the equations (26) it holds

$$\sum_{i=1}^{d+1} \lambda_i (t_1)_i + \sum_{i=1}^{d+1} \lambda_i \tilde{z}_i (t_2)_i$$

$$= \sum_{i=1}^{d+1} \lambda_i (t_1)_i + \sum_{i=1}^{d+1} \lambda_i (\epsilon_i \gamma \langle w, x_i \rangle - \epsilon_i K_i + \epsilon_i \xi_i')(t_2)_i$$

$$= \sum_{i=1}^{d+1} \lambda_i \left( \epsilon_i \langle \gamma w, x_i \rangle (t_2)_i - \epsilon_i K_i (t_2)_i + \epsilon_i \xi_i (t_2)_i + (t_1)_i \right)$$

$$= \sum_{i=1}^{d+1} \lambda_i \left( \langle \gamma w, x_i \rangle C_i + C_i' \right) + \sum_{i=1}^{d} \lambda_i \xi_i' C_i,$$

for the integers $C_i = \epsilon_i (t_2)_i$ and $C_i' = -\epsilon_i K_i (t_2)_i + (t_1)_i$. Since $t \in \mathcal{T}$ some elementary algebra considerations imply that either not all $(C_i)_{i=1,\dots,d+1}$ are equal to each other or one of the $(C_i')_{i=1,2,\dots,d+1}$ is not equal to zero. Let us call this region of permissible pairs $(C, C')$ as $\mathcal{C}$. Furthermore, given that all $t$ satisfy $\|t\|_2 \le 2^{2d}$, and that for all $K_i$ satisfy $|K_i| \le d^Q$ it holds that any $(C, C')$ defined through the above equations with respect to $t_1, t_2, \epsilon_i, K_i$ satisfies the crude bound that

$$\|(C, C')\|_2^2 \le \|t_2\|_2^2 + 2(d^{2Q} \|t_2\|_2^2 + \|t_1\|_2^2) \le 2^{6d}.$$

Hence, using this refined notation it suffices to show

$$\mathbb{P} \left( \bigcup_{(C,C') \in \mathcal{C}, \|(C,C')\|_2 \le 2^{3d}} \left\{ \left| \sum_{i=1}^{d+1} \lambda_i (\langle \gamma w, x_i \rangle C_i + C_i') + \sum_{i=1}^{d} \lambda_i \xi_i C_i \right| \le 2^{4d}/2^N \right\} \right) \le \exp(-\Omega(d)).$$

Now notice that from our exponential-in-$d$ norm upper bound assumptions on $C$, the part 4 of Lemma E.8, and since $N = o((d \log d)^3)$, the following holds with probability $1 - \exp(-\Omega(d))$

$$\sum_{i=1}^{d} |\lambda_i \xi_i C_i| = O(2^{4d} \|\xi\|_\infty) = O(\exp(-(d \log d)^3)) = O(2^{-N}).$$

Hence it suffices to show that for large enough values of $d$,

$$\mathbb{P} \left( \bigcup_{(C,C') \in \mathcal{C}, \|(C,C')\|_2 \le 2^{3d}} \left\{ \left| \sum_{i=1}^{d+1} \lambda_i (\langle \gamma w, x_i \rangle C_i + C_i') \right| \le 2^{5d}/2^N \right\} \right) \le \exp(-\Omega(d)).$$

Using the polynomial notation of Lemma E.10 and specifically notation (33), as well as the fact that by Cramer's rule $\lambda_i$ are rational functions of the coordinates of $x_i$ satisfying $\lambda_i \det(x_2, \dots, x_{d+1}) = \det(\dots, x_{i-1}, -x_1, x_{i+1}, \dots)$ it suffices to show

$$\mathbb{P} \left( \bigcup_{(C,C') \in \mathcal{C}, \|(C,C')\|_2 \le 2^{3d}} \{|P_{C,C'}(x_1, \dots, x_{d+1})| \le |\det(x_2, \dots, x_{d+1})| 2^{5d}/2^N \} \right) \le \exp(-\Omega(d)).$$

Using the fifth part of the Lemma E.8 there exists some constant $D > 0$ for which it suffices to show

$$\mathbb{P} \left( \bigcup_{(C,C') \in \mathcal{C}, \|(C,C')\|_2 \le 2^{3d}} \{|P_{C,C'}(x_1, \dots, x_{d+1})| \le 2^{Dd \log d}/2^N \} \right) \le \exp(-\Omega(d)).$$

Now since $N = \Theta(d^3 (\log d)^2)$ we have $N = \omega(d \log d)$. Hence, for sufficiently large $d$ it suffices to show

$$\mathbb{P} \left( \bigcup_{(C,C') \in \mathcal{C}, \|(C,C')\|_2 \le 2^{3d}} \{|P_{C,C'}(x_1, \dots, x_{d+1})| \le 2^{-\frac{N}{2}} \} \right) \le \exp(-\Omega(d)).$$

By a union bound, it suffices

$$\sum_{(C,C')\in\mathcal{C},\|(C,C')\|_2\leq 2^{3d}} \mathbb{P}\left(|P_{C,C'}(x_1,\ldots,x_{d+1})| \leq 2^{-\frac{N}{2}}\right) \leq 2^{-\Omega(d)}. \qquad (35)$$

Now the integer points $(C,C')$ with $\ell_2$ norm at most $2^{3d}$ are at most $2^{3d^2+d}$ as they have at most $2^{3d+1}$ choices per coordinate. Furthermore, using the anticoncentration inequality (34) of Lemma E.10, we have for any $(C,C')\in\mathcal{C}$ that it holds for some universal constant $B > 0$,

$$\mathbb{P}\left(|P_{C,C'}(x_1,\ldots,x_{d+1})| \leq 2^{-\frac{N}{2}}\right) \leq B(d+1)2^{-\frac{N}{2(d+1)}}.$$

Combining the above with the left hand side of (35), the right hand side is at most

$$B(d+1)2^{3d^2+d}2^{-\frac{N}{2(d+1)}} = \exp(O(d^2) - \Omega(N/d)) = \exp(-\Omega(d)),$$

where we used that $N/d = \Omega(d^2 \log d)$. This completes the proof. $\qquad\square$

# F  Exact Recovery for Phase Retrieval with Optimal Sample Complexity

Phase retrieval is a classic inverse problem [A12] with important applications in computational physics and signal processing, and which has been thoroughly studied in the high-dimensional statistics and non-convex optimization literature [A4, A21, A18, A35, A5, A9, A28, A29, A34]. In the noiseless setting, the phase retrieval problem asks one to exactly recover a hidden signal $w \in \mathbb{R}^d$, up to global symmetry $\pm w$, given sign-less measurements of the form

$$y = |\langle x, w\rangle|\,.$$

As mentioned in Section 1.2, our cosine learning problem can be seen as "containing" the phase retrieval problem since the even-ness of the cosine function immediately "erases" the sign of the inner product $\langle x, w\rangle$. More precisely, the phase retrieval problem can be *reduced* to the cosine learning problem by simply applying the cosine function to the measurements and noticing that

$$\cos(2\pi|\langle x, w\rangle|) = \cos(2\pi\langle x, w\rangle)\,.$$

Hence, Algorithm 1, without the last normalization step (see Remark E.6), can be immediately used to exactly solve phase retrieval under exponentially small noise. Formally, Theorem 3.6 (for $\gamma = \|w\|_2$) certifies near exact recovery for (Gaussian) phase retrieval using only $d + 1$ samples:

**Corollary F.1** (Recovery of Phase Retrieval under exponentially small noise). *Let us consider noise level $\beta \leq (2\pi)^{-1}\exp(-(d\log d)^3)$, and arbitrary $w \in \mathbb{R}^d$ such that $1 \leq \|w\|_2 = \mathsf{poly}(d)$. Suppose $\{(x_i, y_i)\}_{i=1,\ldots d+1}$ are i.i.d. samples of the form $x_i \sim N(0, I_d)$ and $y_i = |\langle x_i, w\rangle| + \check{\xi}_i$, with arbitrary $|\check{\xi}_i| \leq \beta$. Then Algorithm 1 with input $\{(x_i, z_i = \cos(2\pi y_i))\}_{i=1,\ldots d+1}$ returns an un-normalized output $w'$ satisfying $\min\{\|w' - w\|_2, \|w' + w\|_2\} = O(\beta)$ and terminates in $\mathsf{poly}(d)$ steps, with probability $1 - \exp(-\Omega(d))$.*

Remarkably, our lattice-based algorithm improves upon the AMP-based algorithm analysed in [A5], which requires $m \approx 1.128d$ in the high-dimensional regime for exact recovery, and therefore shows that AMP is not optimal amongst polynomial-time algorithms in the regime of exponentially small adversarial noise. Hence, this adds phase retrieval to a list of problems, including for example linear regression with discrete coefficients, where in the exponentially-small noise regime no computational-statistical gap is present [A44] [A23, Section 4.2.1]. We note that the possibility that LLL might be efficient for exponentially-small noise phase retrieval was already suggested in [A44] and later established for discrete-valued $w$ in [A16]. In fact, previous results by [A2] have already shown that exact (i.e., noiseless) phase retrieval is possible with optimal sample complexity using an LLL-based algorithm very similar to ours. We also remark that our result is stated under the Gaussian distribution, as opposed to generic *i.i.d.* entries as in [A5]. The reason is that we rely crucially on anti-concentration properties of random low-degree polynomials, which are satisfied in the Gaussian case [A7, A30]. However, these anti-concentration properties can be extended to log-concave random variables [A7, Theorem 8], and as a result our analysis easily extends to $x_i$ following a product distribution of a density which is both log-concave and sub-Gaussian. In this respect, we strengthen previous results by [A2], whose analysis is tailored to the Gaussian case.

An interesting question is whether the sample size $d + 1$ is *information-theoretically optimal* to recover $w$ up to error $\beta$ from the studied phase retrieval setting. In other words, whether the recovery is possible with $d$ samples by any estimator, and irrespective of any computational constraints. For simplicity, we focus on the noiseless case $\beta = 0$, in which case the goal is *exact recovery*. We note that the answer depends on the prior knowledge on $w$, or, assuming throughout a rotationally invariant prior for $w$, on the prior distribution of $\|w\|$. Indeed, in the extreme setting where the hidden vector $w \in \mathbb{R}^d$ is unconstrained, we immediately observe that there are $2^d$ possible vectors $w'$ satisfying $|\langle x_i, w' \rangle| = |\langle x_i, w \rangle|$. As a consequence, by taking into consideration the global sign flip symmetry, exact recovery is possible only with probability at most $2^{-d+1}$. On the other extreme, if one knew that $\|w\| = 1$, then generically only two ($w$ and $-w$) of these $2^d$ possibilities will satisfy the exact norm constraint, making exact recovery (up to global sign flip) possible with only $d$ samples in that case. The following theorem addresses the general case between these two extremes, and establishes that exact recovery using only $d$ samples cannot be generally certified with high probability, in stark contrast with Corollary F.1.

**Theorem F.2.** *Assume a uniform prior on the direction $w/\|w\|_2 \in S^{d-1}$, and assume that $\gamma = \|w\|_2 > 0$ is distributed independently of $w$ according to a probability density $q_\gamma$ which satisfies the following assumption: For some $B > \sqrt{2}$ and $C > 0$, the function $q_\gamma : \mathbb{R} \to [0, +\infty)$ satisfies*

$$q_\gamma(t) t^{-d+1} \text{ is non-increasing in } t \in [1, B], \text{ and } \int_{\sqrt{2}}^{B} q_\gamma(t) dt \geq C. \tag{36}$$

*Consider $d \geq 2$ i.i.d. samples $\{x_i, y_i = |\langle x_i, w \rangle|\}_{i=1...d}$, where $x_i$ are i.i.d. $N(0, I_d)$ and $w$ is drawn from two independent variables: $w/\|w\|$ uniformly distributed in $S^{d-1}$ and $\|w\|$ is distributed with density $q_\gamma$ satisfying (36). Let $\mathcal{A}$ be any estimation procedure (deterministic or randomized) that takes as input $\{(x_i, y_i)\}_{i=1,...,d}$ and outputs $w' \in \mathbb{R}^d$. Then with probability $\omega(d^{-2})$ it holds $w' \notin \{-w, w\}$.*

This theorem is proved in Appendix J. The main idea of the proof is to show that, with non-neglibile probability ($\omega(d^{-2})$), some of the 'spurious' solutions $w'$ satisfying $|\langle x_i, w' \rangle| = |\langle x_i, w \rangle|$ are such that $\|w'\| \leq \|w\|$. Combined with our assumption on the prior $q_\gamma$ and the optimality of MAP estimators in terms of error probability, the result follows. We also note that Assumption (36) is very mild, and is satisfied e.g. when $\gamma$ is uniformly distributed in $[1, B]$, or when $w$ is either uniformly distributed in a circular ring, or follows a Gaussian distribution. Therefore, our proposed algorithm, as well as the algorithm used in [A2], obtains a *sharp* optimal sample complexity in this phase-retrieval setup, in the sense that even one less sample than the sample complexity of our algorithm is not sufficient for exact recovery with high probability.

Finally, we would like to highlight that our result and the described lower bound should be also understood in contrast with the recently established *weak recovery* threshold that $d/2(1 + o(1))$ measurements actually suffice for achieving some non-trivial (constant) error with $w$ [A35].

## G   Approximation with One-Hidden-Layer ReLU Networks

The members of the cosine function class $\mathcal{F}_\gamma = \{\cos(2\pi\gamma\langle w, x \rangle) \mid w \in S^{d-1}\}$ consist of a composition of the univariate $2\pi$-Lipschitz, 1-periodic function $\phi(z) = \cos(2\pi z)$, and an one-dimensional linear projection $z = \gamma\langle w, x \rangle$. Notice that since $x \sim N(0, I_d)$, $z$ lies within the interval $[-R, R]$, where $R = \gamma\sqrt{2\log(1/\delta)}$, with probability at least $1 - \delta$ due to Mill's inequality (Lemma K.3). Hence, to achieve $\epsilon$-squared loss over the Gaussian input distribution, it suffices for the ReLU network to uniformly approximate the univariate function $\phi(z) = \cos(2\pi z)$ on some compact interval $[-R(\gamma, \epsilon), R(\gamma, \epsilon)]$, and output 0 for all $z \in \mathbb{R}$ outside the compact interval.

The uniform approximability of univariate Lipschitz functions by the family of one-hidden-layer ReLU networks on compact intervals is well-known. To establish our results, we will use the quantitative result from [A11], which we reproduce here as Lemma G.1. We present our ReLU approximation result for the cosine function class right after, in Theorem G.2.

**Lemma G.1** ([A11, Lemma 19]). *Let $\sigma(z) = \max\{0, z\}$ be the ReLU activation function, and fix $L, \eta, R > 0$. Let $f : \mathbb{R} \to \mathbb{R}$ be an $L$-Lipschitz function which is constant outside an interval $[-R, R]$.*

*There exist scalars $a, \{\alpha_i, \beta_i\}_{i=1}^w$, where $w \leq 3\frac{RL}{\eta}$, such that the function*

$$h(x) = a + \sum_{i=1}^w \alpha_i \sigma(x - \beta_i)$$

*is L-Lipschitz and satisfies*

$$\sup_{x \in \mathbb{R}} |f(x) - h(x)| \leq \eta.$$

*Moreover, one has $|\alpha_i| \leq 2L$.*

**Theorem G.2.** *Let $d \in \mathbb{N}$, $\gamma \geq 1$, and $\epsilon \in (0, 1)$ be a real number. Then, the cosine function class $\mathcal{F}_\gamma = \{\cos(2\pi\gamma\langle w, x\rangle) \mid w \in S^{d-1}\}$ can be $\epsilon$-approximated (in the squared loss sense) over the Gaussian input distribution $x \sim N(0, I_d)$ by one-hidden-layer ReLU networks of width at most $O\left(\gamma\sqrt{\frac{\log(1/\epsilon)}{\epsilon}}\right)$.*

*Proof.* Let $R = \lceil \gamma\sqrt{2\log(8/\epsilon)} \rceil + 1/2$, and $z = \gamma\langle w, x\rangle$. Then, by Mill's inequality (Lemma K.3) and the fact that $R > \gamma$,

$$\mathbb{P}(|z| \geq R) \leq \sqrt{\frac{2}{\pi}} \exp\left(-\frac{R^2}{2\gamma^2}\right) \leq \frac{\epsilon}{8}. \tag{37}$$

Let $f : \mathbb{R} \to \mathbb{R}$ be a function which is equal to $\cos(2\pi z)$ on $[-R, R]$ and 0 outside the compact interval. We claim that $f$ is still $2\pi$-Lipschitz. First, note that $\cos(2\pi R) = \cos(-2\pi R) = 0$. Moreover, $f$ is $2\pi$-Lipschitz within the interval $[-R, R]$ and 0-Lipschitz in the region $|z| > R$. Hence, it suffices to consider the case when one point $z$ falls inside $[-R, R]$ and another point $z'$ falls outside the interval. Without loss of generality, assume that $z \in [-R, R]$ and $z' > R$. The same argument applies for $z' < -R$. Then,

$$|f(z') - f(z)| = |f(R) - f(z)| \leq 2\pi|R - z| \leq 2\pi|z' - z|.$$

Now set $L = 2\pi, \eta = \sqrt{\epsilon/2}, R = \lceil \gamma\sqrt{2\log(8/\epsilon)} \rceil + 1/2$ in the statement of Lemma G.1, and approximate $f$ with a one-hidden-layer ReLU network $g(z)$ of width at most $O\left(\gamma\sqrt{\frac{\log(1/\epsilon)}{\epsilon}}\right)$. Then,

$$
\begin{aligned}
\mathbb{E}_{x \sim N(0, I_d)}[(\cos(2\pi\gamma\langle w, x\rangle) - g(\gamma\langle w, x\rangle))^2] &= \mathbb{E}_{z \sim N(0,\gamma)}[(\cos(2\pi z) - g(z))^2] \\
&= \frac{1}{\gamma\sqrt{2\pi}} \int (\cos(2\pi z) - g(z))^2 \exp(-z^2/(2\gamma^2))dz \\
&= \frac{1}{\gamma\sqrt{2\pi}} \int_{|z| \leq R} (\cos(2\pi z) - g(z))^2 \exp(-z^2/(2\gamma^2))dz \\
&\quad + \frac{1}{\gamma\sqrt{2\pi}} \int_{|z| > R} (\cos(2\pi z) - g(z))^2 \exp(-z^2/(2\gamma^2))dz \\
&\leq \eta^2 + \frac{4}{\gamma\sqrt{2\pi}} \int_{|z| > R} \exp(-z^2/(2\gamma^2))dz \\
&\leq \eta^2 + 4(\epsilon/8) \\
&< \epsilon,
\end{aligned}
$$

where the first inequality follows from the fact that the squared loss is bounded by 4 for all $z \notin [-R, R]$ since $\cos(2\pi z) \in [-1, 1]$ and $g(z) \in [-\eta, \eta] \subset [-1, 1]$ and the second inequality uses (37). This completes the proof. $\square$

# H    Covering Algorithm for the Unit Sphere

The (very simple) randomized exponential-time algorithm for constructing an $\epsilon$-cover of the $d$-dimensional unit sphere $S^{d-1}$ is presented in Algorithm 5. We prove the algorithm's correctness in the following claim, which is essentially an appropriate application of the coupon collector problem.

**Algorithm 5:** Exponential-time algorithm for constructing an $\epsilon$-cover of the unit sphere

---

**Input:** A real number $\epsilon \in (0,1)$, and natural number $d \in \mathbb{N}$.
**Output:** An $\epsilon$-cover of the unit sphere $S^{d-1}$ containing $2N \log N$ points, where
    $N = (1 + 4/\epsilon)^d$ with probability $1 - \exp(-\Omega(d))$.

---

Initialize the cover $\mathcal{C} = \emptyset$, and set $m = 2N \log N$.
**for** $i = 1$ **to** $m$ **do**
  Sample $x \sim N(0,1)$
  $v \leftarrow x/\|x\|_2$
  Add $v \in S^{d-1}$ to $\mathcal{C}$
**return** $\mathcal{C}$.

---

**Claim H.1.** *Let $d \in \mathbb{N}$ be a number, let $\epsilon \in (0,1)$ be a real number, and let $N = \lceil (1 + 4/\epsilon)^d \rceil$. Then, $\lceil 2N \log N \rceil$ vectors sampled from $S^{d-1}$ uniformly at random forms an $\epsilon$-cover of $S^{d-1}$ with probability at least $1 - \exp(-\Omega(d))$.*

*Proof.* By Lemma C.2, we know that there exists an $\epsilon/2$-cover of $S^{d-1}$ with size less than $N = \lceil (1 + 4/\epsilon)^d \rceil$. Let us assume for simplicity and without loss of generality, that it's size equals to $N$, by adding additional arbitrary points on the sphere to the cover if necessary. We denote this $\epsilon/2$-cover by $\mathcal{K}$. Of course, $\mathcal{K} \subseteq S^{d-1}$ by the definition of an $\epsilon$-cover in [A42, Section 4.2].

Now, observe that any family $W$ of $M$ vectors on the sphere, say $W = \{w_1, \ldots, w_M\}$, with the property that for any $v \in \mathcal{K}$ there exist $i \in [M]$ such that $\|v - w_i\|_2 \le \epsilon/2$ is an $\epsilon$-cover of $S^{d-1}$. Indeed, let $x \in S^{d-1}$. Since $\mathcal{K}$ is an $\epsilon/2$-cover, there exist $v \in \mathcal{K}$ with $\|x - v\|_2 \le \epsilon/2$. Moreover, using the property of the family $W$, there exists some $i \in [M]$ for which $\|v - w_i\|_2 \le \epsilon/2$. By triangle inequality we have $\|w_i - x\|_2 \le \epsilon$.

Now, by definition of the $\epsilon/2$-cover it holds

$$\bigcup_{v \in \mathcal{K}} \left( B(v, \epsilon/2) \cap S^{d-1} \right) = S^{d-1},$$

where by $B(x,r)$ we denote the Euclidean ball in $\mathbb{R}^d$ with center $x \in \mathbb{R}^d$ and radius $r$. Hence, denoting by $\mu$ the uniform probability measure on the sphere, by a simple union bound we conclude that for all $v \in \mathcal{K}$, $N\mu(B(v, \epsilon/2) \cap S^{d-1}) \ge 1$ or

$$\mu(B(v, \epsilon/2) \cap S^{d-1}) \ge \frac{1}{N}. \tag{38}$$

In other words, if we fix some $v \in K$ and sample a uniform point $w$ on the sphere, it holds that with probability at least $1/N$ we have $\|w - v\|_2 \le \epsilon/2$.

Hence, the probability that $M$ random i.i.d. unit vectors $w_1, \ldots, w_M$ are all at distance more than $\epsilon/2$ from a fixed $v \in \mathcal{K}$ is upper bounded by

$$\mathbb{P}\left( \bigcap_{i=1}^{M} \{\|u_i - v\|_2 > \epsilon/2\} \right) \le (1 - 1/N)^m \le \exp(-m/N) .$$

Now let $M = 2N \log N$. By the union bound, the probability that there exists some $v \in \mathcal{K}$ not covered by $M$ random unit vectors $w_1, \ldots, w_M$ is upper bounded by

$$\mathbb{P}\left( \bigcup_{v \in \mathcal{K}} \{\|u_i - v\|_2 > \epsilon/2 \text{ for all } i = 1, \ldots, M\} \right) \le |\mathcal{K}| \cdot \exp(-M/N) \le 1/N .$$

Since $N = \exp(\Omega(d))$, we conclude that $M = 2N \log N$ random unit vectors form an $\epsilon$-cover of $S^{d-1}$ with probability $1 - \exp(-\Omega(d))$. The proof is complete. $\qquad\square$

# I The Population Loss and Parameter Estimation

Let $f(x) = \cos(2\pi\gamma\langle w, x\rangle)$ be the target function defined on Gaussian inputs $x \sim N(0, I_d)$. In this section, we consider the proper learning setup, where we wish to learn a unit vector $w'$ such that the hypothesis $g_{w'}(x) = \cos(2\pi\gamma\langle w', x\rangle)$ achieves small squared loss with respect to the target function $f$. Towards this goal, we define the squared loss associated with a unit vector $w' \in S^{d-1}$.

**Definition I.1.** *Let $d \in \mathbb{N}, \gamma \geq 1$, and $w \in S^{d-1}$ be some fixed hidden direction. For any $w' \in S^{d-1}$, we define the population loss $L(w')$ of the hypothesis $g_{w'}(x) = \cos(2\pi\gamma\langle w', x\rangle)$ with respect to $w$ by*

$$L(w') = \mathbb{E}_{x \sim N(0, I_d)}[(\cos(2\pi\gamma\langle w, x\rangle) - \cos(2\pi\gamma\langle w', x\rangle))^2] . \tag{39}$$

Notice that because the cosine function is even, the population loss inherits the sign symmetry and satisfies that $L(w') = L(-w')$ for all $w' \in S^{d-1}$. Reflecting that symmetry, we obtain a Lipschitz relation between the population loss and the squared $\ell_2$ difference between $w$ and $w'$ (or $-w'$ if $\|w + w'\|_2 \leq \|w - w'\|_2$). In particular, when $\gamma$ is diverging, we can rigorously show that recovery of $w$ with $o(1/\gamma)$ $\ell_2$-error is sufficient for (properly) learning the associated cosine function with constant edge. This is formally stated in Corollary I.3. We start with the following useful proposition.

**Proposition I.2.** *For every $w' \in S^{d-1}$ it holds*

$$L(w') = 2 \sum_{k \in 2\mathbb{Z}_{\geq 0}} \frac{(2\pi\gamma)^{2k}}{k!} \exp(-4\pi^2\gamma^2) \left(1 - \langle w, w'\rangle^k\right) . \tag{40}$$

*In particular,*

$$L(w') \leq 4\pi^2\gamma^2 \min\{\|w - w'\|_2^2, \|w + w'\|_2^2\}. \tag{41}$$

*Proof.* Let $\{h_k\}_{k \in \mathbb{Z}_{\geq 0}}$ be the (probabilist's) normalized Hermite polynomials. We have that the pair $Z = \langle w, x\rangle, Z_\rho = \langle w', x\rangle$ is a bivariate pair of standard Gaussian random variables with correlation $\rho = \langle w, w'\rangle$. Using the fact that $h_k$'s form an orthonormal basis in Gaussian space (See item (1) of Lemma K.10), we have by Parseval's identity that

$$L(w') = 2(\mathbb{E}[\cos(2\pi\gamma Z)^2] - \mathbb{E}[\cos(2\pi\gamma Z)\cos(2\pi\gamma Z_\rho)])$$
$$= 2\sum_{k \in \mathbb{Z}} \left(\mathbb{E}[\cos(2\pi\gamma Z)h_k(Z)]^2 - \mathbb{E}[\cos(2\pi\gamma Z)h_k(Z)]\mathbb{E}[\cos(2\pi\gamma Z_\rho)h_k(Z)]\right) .$$

Using now item (2) of Lemma K.10 for $\rho = 1$ and for $\rho = \langle w, w'\rangle$, we have

$$L(w') = 2\sum_{k \in \mathbb{Z}} \left(\frac{(2\pi\gamma)^{2k}}{k!} \exp(-4\pi^2\gamma^2) - \langle w, w'\rangle^k \frac{(2\pi\gamma)^{2k}}{k!} \exp(-4\pi^2\gamma^2)\right)$$
$$= 2\sum_{k \in 2\mathbb{Z}_{\geq 0}} \frac{(2\pi\gamma)^{2k}}{k!} \exp(-4\pi^2\gamma^2) \left(1 - \langle w, w'\rangle^k\right) ,$$

as we wanted for the first part.

For the second part, notice that since the summation on the right hand from Eq. (40) is only containing an even power of $\langle w, w'\rangle$ it suffices to establish the upper bound in terms of $\|w - w'\|_2^2$. The exact same argument can be used to obtain the upper bound in terms of $\|w + w'\|_2^2$, due to the observed sign symmetry of the population loss with respect to $w'$.

Now notice that using the elementary inequality that for $\alpha \in (0, 1), x \geq 1$ we have $(1-a)^x \geq 1-ax$, we conclude that for all $k \geq 0$ (the case $k = 0$ is trivial) it holds

$$1 - \langle w, w'\rangle^k = 1 - (1 - \frac{1}{2}\|w - w'\|_2^2)^k \leq \frac{k}{2}\|w - w'\|_2^2 .$$

Hence, combining with the first part, we have

$$L(w') \leq \sum_{k \in 2\mathbb{Z}_{\geq 0}} k\frac{(2\pi\gamma)^{2k}}{k!} \exp(-4\pi^2\gamma^2)\|w - w'\|_2^2$$

$$\leq \sum_{k \in \mathbb{Z}_{\geq 0}} k\frac{(2\pi\gamma)^{2k}}{k!} \exp(-4\pi^2\gamma^2)\|w - w'\|_2^2 .$$

Now notice that $\sum_{k \in \mathbb{Z}_{\geq 0}} k \frac{(2\pi\gamma)^{2k}}{k!} \exp(-4\pi^2\gamma^2)$ is just the mean of a Poisson random variable with parameter (and mean) equal to $4\pi^2\gamma^2$. Hence, the proof of the second part of the proposition is complete. $\qquad \square$

The following Corollary is immediate given the above result and the item (3) of Lemma K.10.

**Corollary I.3.** *Let $d \in \mathbb{N}$ and $\gamma = \gamma(d) = \omega(1)$. For any $w' \in S^{d-1}$ which satisfies $\min\{\|w - w'\|_2^2, \|w + w'\|_2^2\} \leq \frac{1}{16\pi^2\gamma^2}$ and sufficiently large $d$,*

$$L(w') \leq \mathrm{Var}(\cos(2\pi\gamma\langle w, x\rangle)) - 1/12 .$$

*Proof.* Using our condition and $w'$ and the second part of the Proposition I.2 we conclude

$$L(w') \leq \frac{1}{4} .$$

Now using item (3) of Lemma K.10 we have that for large values of $d$ (since $\gamma = \omega(1)$), it holds

$$\frac{1}{3} \leq \mathrm{Var}(\cos(2\pi\gamma\langle w, x\rangle)) .$$

The result follows from combining the last two displayed inequalities. $\qquad \square$

## J   Optimality of $d + 1$ samples for exact recovery under norm priors

In this appendix, we argue that $d + 1$ samples are necessary in order to obtain exact recovery with probability $1 - \exp(-\Omega(d))$, irrespective of any estimation procedure. Since our upper bound holds for arbitrary $w/\|w\|_2 \in S^{d-1}$, and arbitrary $1 \leq \gamma = \|w\|_2 = \mathrm{poly}(d)$, it suffices to prove a lower bound for *some* distributional assumption on $\gamma$ and $w/\|w\|_2$ which respects these constraints. Hence, for our lower bound, we assume a uniform prior on the direction $w/\|w\|_2 \in S^{d-1}$, and assume that $\gamma = \|w\|_2 > 0$ is distributed independently of $w$ according to a probability density $q_\gamma$ which satisfies the following assumption.

**Assumption J.1.** *For some $B > \sqrt{2}$ and $C > 0$, the function $q_\gamma : \mathbb{R} \to [0, \infty)$ satisfies that $q_\gamma(t)t^{-d+1}$ is non-increasing for $t \in [1, B]$, and $\int_{\sqrt{2}}^{B} q_\gamma(t)dt \geq C$.*

We now state our lower bound, restating Theorem F.2 for convenience.

**Theorem J.2.** *Consider $d \geq 2$ samples $\{(x_i, y_i = |\langle x_i, w\rangle|)\}_{i=1\dots d}$, in which the $x_i$'s are drawn i.i.d. from $N(0, I_d)$, and $w$ is drawn from two independent variables: $w/\|w\|$ uniformly distributed in $S^{d-1}$ and $\|w\|$ distributed with density satisfying Assumption J.1. Let $\mathcal{A}$ be any estimation procedure (deterministic or randomized) that takes as input $\{(x_i, y_i)\}_{i=1,\dots,d}$ and outputs $w' \in \mathbb{R}^d$. Then with probability $\omega(d^{-2})$ it holds $w' \notin \{-w, w\}$.*

*Proof.* The key idea of the proof will be to establish that with probability $\omega(d^{-2})$ over the draws of the data $\{x_i\}_{i=1,\dots,d}$ and the hidden vector $w$, the following event occurs: There exist a pair of antipodal solutions $\{-w', w'\}$ different from $\pm w$, such that the posterior probability measure $p(\tilde{w} \mid \{(x_i, y_i)\}_{i=1,\dots,d})$ over any possible hidden vector $\tilde{w} \in \mathbb{R}^d$ satisfies $p(\{-w', w'\} \mid \{(x_i, y_i)\}) \geq p(\{-w, w\}| \mid \{(x_i, y_i)\})$. In this event, the MAP estimator will thus fail to exactly recover $\{-w, w\}$ at least with probability $1/2$ (over the randomness of the algorithm). Finally, using the optimality of the Maximum-a-Posteriori Bayes estimator in minimizing the probability of error, the result follows.

Let $X = (x_i)_{i=1\dots d} \in \mathbb{R}^{d\times d}$, be the matrix where for $i = 1, \dots, d$ with $i$-th *row* equal to $x_i^\top$, and $X^{-1}$ its inverse (which exists with probability 1 since the determinant of a squared matrix with i.i.d. Gaussian entries is non-zero almost surely [A8]). Furthermore, let $y = (y_i)_{i=1\dots d} \in \mathbb{R}^d$ the vector of the labels. Let us introduce binary variables $\varepsilon \in \{-1, 1\}^d$, and the associated matrix

$$A_\varepsilon := X^{-1}\mathrm{diag}(\varepsilon)X, .$$

where by $\mathrm{diag}(\varepsilon)$ we refer to the $d \times d$ diagonal matrix with the vector $\varepsilon$ on the diagonal.

We say that a $w' \in \mathbb{R}^d$ is a feasible solution if for all $i = 1, \ldots, d$ it holds that $|\langle x_i, w' \rangle| = y_i$. Notice that if $w'$ is a feasible solution, then for any $\varepsilon \in \{-1, 1\}^d$, $A_\varepsilon w'$ is also a feasible solution. This follows since for each $i = 1, \ldots, d$ it holds by definition $x_i^\top X^{-1} = e_i^\top$, where $e_i$ is the $i$-th standard basis vector, and therefore $x_i^\top A_\epsilon = \varepsilon_i x_i^\top$. Hence we have

$$|x_i^\top A_\varepsilon w'| = |\varepsilon_i x_i^\top w'| = y_i .$$

On the other hand, if $w'$ is a feasible solution, then there exists $\varepsilon \in \{-1, 1\}^d$, for which for all $i = 1, \ldots, d$, it holds $\langle x_i, w' \rangle = \varepsilon_i y_i$. Therefore, using the definition of $y_i$ and the already established properties of $A_\varepsilon$,

$$\langle x_i, w' \rangle = \varepsilon_i y_i = x_i^\top \varepsilon_i w = x_i^\top A_\varepsilon w .$$

Hence, $X(w' - A_\varepsilon w) = 0$. As $X$ is invertible almost surely, we conclude that $w' = A_\varepsilon w$. Combining the above, we conclude that the set of feasible solutions is almost surely the set

$$\mathcal{B}_w = \{A_\varepsilon w | \varepsilon \in \{-1, 1\}^d\}.$$

Of course, this set includes $w$ when $\varepsilon = \mathbf{1}$ is the all-one vector, and $-w$ when $\varepsilon = -\mathbf{1}$ is the all-minus-one vector. Furthermore, from the almost sure linear independence of all $x_i, i = 1, \ldots, d + 1$, and that $w$ is drawn independent of $X$, we conclude that for all $\varepsilon \notin \{-\mathbf{1}, \mathbf{1}\}$ it holds almost surely that $A_\varepsilon w \notin \{-w, w\}$.

Now consider the joint density of the setup in this notation (where we recall that $\tilde{w} \in \mathbb{R}^d$ denotes the generic vector to be recovered, while $w$ is the actual draw of the prior), which decomposes as

$$p(X, \tilde{w}, y) = p_X(X) \cdot p_{\tilde{w}}(\tilde{w}) \cdot p(y \mid X, \tilde{w}) , X \in \mathbb{R}^{d \times d}, \tilde{w} \in \mathbb{R}^d, y \in \mathbb{R}^d .$$

Notice that since we work under the noiseless assumption it holds $p(y \mid X, \tilde{w}) = \delta(y - |X\tilde{w}|)$, where by a slight abuse of notation for a vector $v \in \mathbb{R}^d$ we denote by $|v| \in \mathbb{R}^d$ the vector with elements $|v_i|, i = 1, \ldots, d$. Further recall that in this notation we sample a hidden $w \sim p_{\tilde{w}}$ and independently a matrix $X \sim p_X$. We observe the vector of labels $y = |Xw|$ and $X$. The posterior probability $p(\tilde{w} \mid X, y)$ is therefore

$$p(\tilde{w} \mid X, y) = \frac{p(X, \tilde{w}, y)}{p(X, y)} \propto p_{\tilde{w}}(\tilde{w}) \cdot p(y \mid X, \tilde{w}) . \tag{42}$$

From our previous argument, we know that this posterior distribution is necessarily supported in the set $\mathcal{B}_w$ of $2^d$ points of the form $(X^{-1} \cdot \text{diag}(\varepsilon))y$ for any $\varepsilon \in \{-1, 1\}^d$, which include $w$. Denoting by $\delta(\tilde{w})$ the Dirac unit mass at $\tilde{w}$, we have

$$p(\tilde{w} \mid X, y) = \frac{1}{Z} \sum_{w' \in \mathcal{B}_w} \alpha_{X,y}(w') \delta(\tilde{w} - w') , \tag{43}$$

for some normalizing constant $Z$ and some coefficients $\alpha_{X,y}(\varepsilon)$ that we now determine. We evaluate the posterior distribution over $\tilde{w}$ from (42) using the coarea formula [A27]: Given an arbitrary test function $\phi \in C_c^\infty(\mathbb{R}^d)$, and $F : \mathbb{R}^d \to \mathbb{R}^d$ defined as $F(u) := |Xu|$, we have

$$Z \int_{\mathbb{R}^d} p(\tilde{w} \mid X, y) \phi(\tilde{w}) d\tilde{w} \; = \; \int_{\mathbb{R}^d} p_{\tilde{w}}(\tilde{w}) \delta(y - F(\tilde{w})) \phi(\tilde{w}) d\tilde{w} \tag{44}$$

$$= \; \int_{\mathbb{R}^d} \left( \int_{F^{-1}(z)} \delta(y - z) p_{\tilde{w}}(u) \phi(u) |DF(u)|^{-1} d\mathcal{H}_0(u) \right) dz \tag{45}$$

$$= \; \int_{\mathbb{R}^d} \delta(y - z) \left( \int_{\mathcal{B}_z} p_{\tilde{w}}(u) \phi(u) |DF(u)|^{-1} d\mathcal{H}_0(u) \right) dz \tag{46}$$

$$= \; \sum_{w' \in \mathcal{B}_w} p_{\tilde{w}}(w') \phi(w') |\det(X)|^{-1} , \tag{47}$$

where $d\mathcal{H}_0$ is the 0-th dimensional Hausdorff measure. From (43) we also have that

$$\int_{\mathbb{R}^d} p(\tilde{w} \mid X, y) \phi(\tilde{w}) d\tilde{w} = \sum_{w' \in \mathcal{B}_w} \alpha_{X,y}(w') \phi(w') ,$$

hence we deduce that the weights in (43) satisfy

$$\forall \varepsilon , \ \alpha_{X,y}(X^{-1} \cdot \mathrm{diag}(\varepsilon)y) = p_{\tilde{w}}(X^{-1} \cdot \mathrm{diag}(\varepsilon)y)|\det(X)|^{-1} .$$

By plugging $y = |Xw| = \mathrm{diag}(\varepsilon^*)Xw$ for the sign coefficients $\varepsilon_i^* = \mathrm{sign}(\langle x_i, w \rangle)$, and recalling the definition of $A_\varepsilon$, we conclude that the posterior distribution over the hidden vector $\tilde{w}$ satisfies almost surely

$$p(\tilde{w} \mid X, y) = \begin{cases} \frac{1}{Z} p_{\tilde{w}}(\tilde{w}) & \tilde{w} \in \mathcal{B}_w \\ 0 & \tilde{w} \notin \mathcal{B}_w \end{cases}$$

where $Z := \sum_{\tilde{w} \in \mathcal{B}_w} p_{\tilde{w}}(\tilde{w})$.

Now to prove the desired result, based on the folklore optimality of the Maximum-A-Posteriori (MAP) estimator in minimizing probability of failure of exact recovery (see Lemma J.4 for completeness) it suffices to prove that with probability $\omega(d^{-2})$ there exists $w' \in \mathcal{B}_w \setminus \{-w, w\}$ such that

$$p_{\tilde{w}}(w') \geq p_{\tilde{w}}(w) . \tag{48}$$

Indeed, recall that since $p_{\tilde{w}}$ is rotationally invariant, we have $p_{\tilde{w}}(\tilde{w}) = p_{\tilde{w}}(-\tilde{w})$ for any $\tilde{w}$, therefore (48) immediately implies $p_{\tilde{w}}(\pm w') \geq p_{\tilde{w}}(\pm w)$. Hence, the MAP estimator (and therefore any estimator) fails to exactly recover an element of $\{w, -w\}$ with probability $\omega(d^{-2})$, as we wanted.

Now, using a standard change of variables to spherical coordinates, for all $\tilde{w} \in \mathbb{R}^d$ the density of the prior equal to $p_{\tilde{w}}(\tilde{w}) = q_\gamma(\|\tilde{w}\|_2)\|\tilde{w}\|_2^{-d+1}$. In particular, based on Assumption 36 it suffices to prove that with probability $\omega(d^{-2})$ there exists a $w' \in \mathcal{B}_w \setminus \{-w, w\}$ such that $1 \leq \|w'\|_2 < \|w\|_2$, or equivalently there exists $\varepsilon \in \{-1, 1\}^d \setminus \{-\mathbf{1}, \mathbf{1}\}$ such that

$$1 \leq \|A_\varepsilon w\|_2 < \|w\|_2 . \tag{49}$$

We establish (49) by actually studying only one such $\varepsilon$, potentially the simplest choice, which we call $\varepsilon^{(1)}$ where $\varepsilon_1^{(1)} = -1$ and $\varepsilon_j^{(1)} = +1$ for $j = 2, \ldots, d$. This is accomplished by the following key lemma:

**Lemma J.3.** *Suppose $X \in \mathbb{R}^{d \times d}$ has i.i.d. $N(0, I_d)$ entries, and $w$ is drawn independently of $X$, such that $w/\|w\|_2$ is drawn from the uniform measure of $S^{d-1}$ and its norm $\|w\|_2$ is independent of $w/\|w\|_2$ and distributed according to a density $q_\gamma$ satisfying Assumption (36). Set also $A_{\varepsilon^{(1)}} = X^{-1}\mathrm{diag}(\varepsilon^{(1)})X$. Then with probability greater than $\omega(d^{-2})$, it holds*

$$1 \leq \|A_{\varepsilon^{(1)}} w\|_2 < \|w\|_2 . \tag{50}$$

This lemma thus proves (49) and the failure of the MAP estimator with probability $\omega(d^{-2})$.

We conclude the proof by formally stating and using the optimality of the MAP estimator in terms of minimizing the error probability, by relating it to a standard error correcting setup. From our previous argument, we can reduce ourselves to decoders that operate in the discrete set $\mathcal{B}_w$, since any $\tilde{w}$ outside this set will be different from $\pm w$ almost surely.

**Lemma J.4.** *Suppose $\mathcal{X}$ is a discrete set, and let $x^* \in \mathcal{X}$ be an element to be recovered, with posterior distribution $p(x|y)$, $x \in \mathcal{X}$, after having observed the output $y = g(x^*)$. Then, any estimator producing $\hat{x} = \hat{x}(y)$ will incur in an error probability $\mathbb{P}(\hat{x} \neq x^*)$ at least $1 - \max_x p(x|y)$, with equality if $\hat{x}$ is the Maximum-A-Posterior (MAP) estimator which outputs $\arg\max_x p(x|y)$.*

We apply the Lemma J.4 for $\mathcal{X}$ containing all the pairs of antipodal elements of $\mathcal{B}_w$, that is $\mathcal{X} = \{\{w', -w'\} : w' \in \mathcal{B}_w\}$ and $x^* = \{w, -w\}$. As we have established that the MAP estimator fails to exactly recover $x^*$ with probability $\omega(d^{-2})$ this completes the proof.

$\square$

## J.1 Proof of Lemma J.3

*Proof.* If $e_1$ denotes the first standard basis vector, observe that by elementary algebra,

$$A_{\varepsilon^{(1)}} = X^{-1} \left( I_d - 2 e_1 e_1^\top \right) X = I_d - 2 \tilde{x}_1 x_1 , \tag{51}$$

where $x_1^\top$ is the first row of $X$ and $\tilde{x}_1$ is the first column of $X^{-1}$.

We need a spectral decomposition of matrices of the form $A = I_d - 2 u v^\top$, which is provided in the following lemma:

**Lemma J.5.** *Let $\eta \in \mathbb{R}$ and $A = I_d - 2\eta u v^\top \in \mathbb{R}^{d \times d}$, with $\|u\|_2 = \|v\|_2 = 1$, and $\alpha = \langle u, v \rangle$. Then $A^\top A$ has the eigenvalue 1 with multiplicity $d - 2$, and two additional eigenvalues $\lambda_1, \lambda_2$ with multiplicity 1 given by*

$$\lambda_1 = 1 + 2\eta \left( \eta - \alpha - \sqrt{\eta^2 + 1 - 2\eta\alpha} \right) , \quad \lambda_2 = 1 + 2\eta \left( \eta - \alpha + \sqrt{\eta^2 + 1 - 2\eta\alpha} \right) . \tag{52}$$

*In particular, $\lambda_{\min}(A^\top A) = \lambda_1 < 1$ and $\lambda_{\max}(A^\top A) = \lambda_2 > 1$ whenever $\eta > 0$ and $|\alpha| < 1$.*

From (51), we now apply Lemma J.5. By noting that $\langle x_1, \tilde{x}_1 \rangle = 1$ since $X X^{-1} = I_d$, note that the lemma applies for $A_{\varepsilon^{(1)}}$ with parameters

$$\alpha = \left\langle \frac{x_1}{\|x_1\|_2}, \frac{\tilde{x}_1}{\|\tilde{x}_1\|_2} \right\rangle = \frac{1}{\|x_1\|_2 \cdot \|\tilde{x}_1\|_2} , \text{ and } \eta = \|x_1\|_2 \cdot \|\tilde{x}_1\|_2 .$$

Since $|\alpha| \in (0, 1]$ by Cauchy-Schwarz and and $\alpha\eta = 1$, it follows that $\eta \geq 1$ and the eigenvalues of $A_{\varepsilon^{(1)}}^\top A_{\varepsilon^{(1)}}$ are $\left( \lambda_{\min}(A_{\varepsilon^{(1)}}^\top A_{\varepsilon^{(1)}}), 1, \ldots, 1, \lambda_{\max}(A_{\varepsilon^{(1)}}^\top A_{\varepsilon^{(1)}}) \right)$, with

$$\lambda_{\min}(A_{\varepsilon^{(1)}}^\top A_{\varepsilon^{(1)}}) = 1 + 2\eta \left( \eta - \alpha - \sqrt{\eta^2 - 1} \right) = -1 + 2\eta^2 - 2\eta\sqrt{\eta^2 - 1} \tag{53}$$

$$\lambda_{\max}(A_{\varepsilon^{(1)}}^\top A_{\varepsilon^{(1)}}) = 1 + 2\eta \left( \eta - \alpha + \sqrt{\eta^2 - 1} \right) = -1 + 2\eta^2 + 2\eta\sqrt{\eta^2 - 1} . \tag{54}$$

In fact, we claim that $|\alpha| < 1$ with probability 1, which by Lemma J.5 implies that

$$\lambda_{\min}(A_{\varepsilon^{(1)}}^\top A_{\varepsilon^{(1)}}) < 1 < \lambda_{\max}(A_{\varepsilon^{(1)}}^\top A_{\varepsilon^{(1)}}) . \tag{55}$$

Indeed, recalling from Lemma J.5 that by definition $\alpha = \langle \frac{x_1}{\|x_1\|}, \frac{\tilde{x}_1}{\|\tilde{x}_1\|} \rangle$ with $\tilde{x}_1 = (X^\top X)^{-1} x_1$, first observe that $|\alpha| < 1$ almost surely. Indeed, $|\alpha| = 1$ iff $\tilde{x}_1$ and $x_1$ are colinear, that is for some scalar $\lambda$ it holds $(X^\top X)^{-1} x_1 = \lambda x_1$, which in particular implies that $x_1$ is an eigenvector of $(X^\top X)^{-1}$, or equivalently of $X^\top X$. Letting $y_i = x_i^\top x_1$, this means that

$$\lambda x_1 = (X^\top X) x_1 = \left( \sum_i x_i x_i^\top \right) x_1 = \sum_i x_i y_i .$$

Since $X$ has rank $d$ almost surely, $\{x_i\}_{i=1 \ldots d}$ are linearly independent almost surely, which in turn implies that $y_i = \langle x_1, x_i \rangle = 0$ for $i \neq 1$ almost surely. This is a 0-probability event since the $x_i$'s are continuously distributed and independent of each other.

In what follows to ease notation we denote $\varepsilon^{(1)}$ simply by $\varepsilon$ and in particular $A_{\varepsilon^{(1)}}$ simply by $A_\varepsilon$. In the following lemma we establish that $\eta \lesssim d^2$ with probability close to 1. The proof of this fact is given in Section J.2. More precisely, we claim the following:

**Lemma J.6.** *There exist constants $C > 0$ and $d_0 > 0$ such that for any $d \geq d_0$,*

$$\mathbb{P} \left( \eta \leq C d^2 \right) \geq 1 - 1/d .$$

We shall now establish (50) building from Lemma J.6. We first relate the spectrum of $A_\varepsilon$ with the probability that $\|A_\varepsilon w\|_2 < \|w\|_2$ or equivalently $\left\| A_\varepsilon \frac{w}{\|w\|_2} \right\|_2 < 1$. Let $\check{w} := w/\|w\|$, so $w = \gamma \check{w}$, with $\check{w} \in S^{d-1}$ uniformly distributed, and independent from $\gamma$. We claim that with respect to the randomness of $\check{w}$ but conditioning on $X$ it holds

$$\mathbb{P}_{\check{w}}(\|A_\varepsilon \check{w}\| < 1) = \frac{2}{\pi} \arcsin \left( \sqrt{\frac{1 - \lambda_{\min}(A_\varepsilon^\top A_\varepsilon)}{\lambda_{\max}(A_\varepsilon^\top A_\varepsilon) - \lambda_{\min}(A_\varepsilon^\top A_\varepsilon)}} \right) . \tag{56}$$

Indeed, assuming without loss of generality that the two eigenvectors of $A_\varepsilon^\top A_\varepsilon$ associated with the distinct eigenvalues $\lambda_{\min}(A_\varepsilon^\top A_\varepsilon)$ and $\lambda_{\max}(A_\varepsilon^\top A_\varepsilon)$ are respectively $e_1$ and $e_2$, the first two standard basis vectors, we have that

$$\|A_\varepsilon \check{w}\|_2^2 = \lambda_{\min}(A_\varepsilon^\top A_\varepsilon)\check{w}_1^2 + \lambda_{\max}(A_\varepsilon^\top A_\varepsilon)\check{w}_2^2 + \sum_{i>2} \check{w}_i^2 \,,$$

and therefore, using the uniform distribution on $S^{d-1}$ of $\check{w}$, it holds

$$
\begin{aligned}
\mathbb{P}_{\check{w}}(\|A_\varepsilon \check{w}\|_2 < 1) &= \mathbb{P}_{\check{w}}(\|A_\varepsilon \check{w}\|_2^2 \le \|\check{w}\|^2) \\
&= \mathbb{P}_{\check{w}}(\lambda_{\min}(A_\varepsilon^\top A_\varepsilon)\check{w}_1^2 + \lambda_{\max}(A_\varepsilon^\top A_\varepsilon)\check{w}_2^2 \le \check{w}_1^2 + \check{w}_2^2) \\
&= \mathbb{P}_{\check{w}}\left(\lambda_{\min}(A_\varepsilon^\top A_\varepsilon)\frac{\check{w}_1^2}{\check{w}_1^2 + \check{w}_2^2} + \lambda_{\max}(A_\varepsilon^\top A_\varepsilon)\frac{\check{w}_2^2}{\check{w}_1^2 + \check{w}_2^2} \le 1\right) \\
&= \mathbb{P}_{\theta \sim U[0,2\pi]}\left(\lambda_{\min}(A_\varepsilon^\top A_\varepsilon)\cos(\theta)^2 + \lambda_{\max}(A_\varepsilon^\top A_\varepsilon)\sin(\theta)^2 \le 1\right) \,, \quad (57)
\end{aligned}
$$

where the last equality follows since the marginal of $\check{w}$ corresponding to the first two coordinates is also rotationally invariant.

From the last identity of (57) and (55), we verify that

$$
\begin{aligned}
\mathbb{P}_{\theta \sim U[0,2\pi]}&\left(\lambda_{\min}(A_\varepsilon^\top A_\varepsilon)\cos(\theta)^2 + \lambda_{\max}(A_\varepsilon^\top A_\varepsilon)\sin(\theta)^2 \le 1\right) \\
&= \frac{1}{2\pi}\int_0^{2\pi} \mathbb{1}\left[\lambda_{\min}(A_\varepsilon^\top A_\varepsilon)\cos(\theta)^2 + \lambda_{\max}(A_\varepsilon^\top A_\varepsilon)\sin(\theta)^2 \le 1\right] d\theta \\
&= \frac{2}{\pi}\int_0^{\pi/2} \mathbb{1}\left[\lambda_{\min}(A_\varepsilon^\top A_\varepsilon)\cos(\theta)^2 + \lambda_{\max}(A_\varepsilon^\top A_\varepsilon)\sin(\theta)^2 \le 1\right] d\theta \\
&= \frac{2}{\pi}\theta^* \,,
\end{aligned}
$$

where $\theta^*$ is the only solution in $(0, \pi/2)$ of

$$\lambda_{\min}(A_\varepsilon^\top A_\varepsilon)\cos(\theta)^2 + \lambda_{\max}(A_\varepsilon^\top A_\varepsilon)\sin(\theta)^2 = 1 \,. \tag{58}$$

From (58) we obtain directly (56), as claimed.

Now, the quantity $\rho := \frac{1-\lambda_{\min}(A_\varepsilon^\top A_\varepsilon)}{\lambda_{\max}(A_\varepsilon^\top A_\varepsilon)-\lambda_{\min}(A_\varepsilon^\top A_\varepsilon)}$, expressed in terms of $\alpha = 1/\eta$ and $\eta$ becomes

$$\rho = \frac{1 - \lambda_{\min}(A_\varepsilon^\top A_\varepsilon)}{\lambda_{\max}(A_\varepsilon^\top A_\varepsilon) - \lambda_{\min}(A_\varepsilon^\top A_\varepsilon)} = \frac{-\eta^2 + \eta\sqrt{\eta^2-1}+1}{2\eta\sqrt{\eta^2-1}} \,,$$

and satisfies $0 \le \rho = \rho(\eta) < 1$ almost surely. Denoting

$$f(\eta) := \arcsin\left(\sqrt{\rho}\right) \,,$$

we verify that $f'(\eta) < 0$ for $\eta \ge 1$. In order to leverage Lemma J.6, we consider the event that $\eta \le C_2 d^2$. We can lower bound $f(\eta)$ as follows. First, observe that $t \mapsto \arcsin(\sqrt{t})$ is non-decreasing in $t \in (0, 1)$, thus

$$f(\eta) \ge \arcsin\left(\sqrt{\frac{\eta(\sqrt{\eta^2-1}-\sqrt{\eta^2})+1}{2\eta^2}}\right) \,,$$

since

$$\frac{-\eta^2 + \eta\sqrt{\eta^2-1}+1}{2\eta\sqrt{\eta^2-1}} \ge \frac{-\eta^2 + \eta\sqrt{\eta^2-1}+1}{2\eta^2} = \frac{\eta(\sqrt{\eta^2-1}-\sqrt{\eta^2})+1}{2\eta^2} \,.$$

Moreover, since $\sqrt{t+1} - \sqrt{t} = \frac{1}{2\sqrt{t}} + O(t^{-3/2})$, we have that

$$\frac{\eta(\sqrt{\eta^2-1}-\sqrt{\eta^2})+1}{2\eta^2} = \frac{3}{4}\eta^{-2} + O(\eta^{-4}) \,,$$

which, combined with the fact that $\arcsin(t) = t + O(t^3)$ for $|t| \leq 1$, leads to

$$f(\eta) \geq \frac{3}{4}\eta^{-1} + O(\eta^{-2}) \ .$$

Finally, using Lemma J.6 and the definition of $f(\eta)$, we obtain that

$$\mathbb{P}_{\check{w}}(\|A_\varepsilon \check{w}\| \leq 1) \geq \frac{6}{4\pi C_2} d^{-2} + O(d^{-4})$$

with probability (over $X$) greater than $1/2$. Since $X$ and $w$ are independent, we conclude that

$$\mathbb{P}_{X,\check{w}}(\|A_\varepsilon \check{w}\| \leq 1) \geq \frac{1}{2}\left(\frac{6}{4\pi C_2}d^{-2} + O(d^{-4})\right) = C_4 d^{-2} + O(d^{-4}) \ , \tag{59}$$

where $C_4$ is a constant.

Now we show that

$$\mathbb{P}_{\check{w}}\left(\|A_\varepsilon \check{w}\|_2^2 \geq 1 - 1/\sqrt{d}\right) \geq 1 - \exp\left(-\Omega(\sqrt{d})\right) \ .$$

Recall that $\check{w}$ is distributed uniformly on the sphere $S^{d-1}$, and that all eigenvalues of $A_\varepsilon^\top A_\varepsilon$ are all greater or equal to 1, except for $\lambda_{\min}$. Assuming without loss of generality that $e_1$ is the eigenvector corresponding to $\lambda_{\min}$, we have for any $\check{w} \in S^{d-1}$,

$$\|A_\varepsilon \check{w}\|_2^2 \geq 1 - \check{w}_1^2 \ .$$

Let $H$ be the hemisphere $H = \{\check{w}_1 \leq 0 \mid \check{w} \in S^{d-1}\}$. By the classic isoperimetric inequality for the unit sphere $S^{d-1}$ [A25, Chapter 1], the measure of the $r$-neighborhood of $H$, which we denote by $H_r = \{u \in S^{d-1} \mid \text{dist}(u, H) \leq r\}$, satisfies

$$\mathbb{P}_{\check{w}}(H_r) = \mathbb{P}_{\check{w}}(\check{w}_1 \leq r) \geq 1 - \exp(-(d-1)r^2/2) \ .$$

An analogous inequality holds for the event $\{\check{w}_1 \geq -r\}$ by the sign symmetry of the distribution of $\check{w}$. Plugging in $r = d^{-1/4}$, It follows that

$$\begin{aligned}
\mathbb{P}_{\check{w}}\left(\|A_\varepsilon \check{w}\|^2 \geq 1 - 1/\sqrt{d}\right) &\geq \mathbb{P}_{\check{w}}\left(1 - \check{w}_1^2 \geq 1 - 1/\sqrt{d}\right) \\
&= \mathbb{P}_{\check{w}}\left(|\check{w}_1| \leq 1/d^{1/4}\right) \\
&\geq 1 - \exp\left(-\Omega(\sqrt{d})\right) \ .
\end{aligned}$$

Therefore, combining the above with (59) using the union bound, we obtain

$$\mathbb{P}_{X,\check{w}}\left(\sqrt{1 - d^{-1/2}} \leq \|A_\varepsilon \check{w}\| \leq 1\right) \geq C_4 d^{-2} + O(d^{-4}) - \exp(-\Omega(\sqrt{d})) = C_4 d^{-2} + O(d^{-4}) \ . \tag{60}$$

Finally, since $B > \sqrt{2}$ and $\sqrt{1 - d^{-1/2}} \geq 1/\sqrt{2}$, we have

$$\begin{aligned}
\mathbb{P}_{\tilde{w}}(\gamma\sqrt{1 - d^{-1/2}} \geq 1) &= & \mathbb{P}_{\tilde{w}}\left(\gamma \geq \frac{1}{\sqrt{1 - d^{-1/2}}}\right) \\
&= & \int_{\frac{1}{\sqrt{1-d^{-1/2}}}}^{B} q_\gamma(v)dv := Q_s \tag{61}
\end{aligned}$$

Since $w = \gamma\check{w}$, where $\check{w}$ is uniformly distributed in $\mathcal{S}^{d-1}$ and $\gamma$ is independent of $\check{w}$, we conclude by assembling (60) and (61) that

$$\mathbb{P}_{X,w}\left(1 \leq \|A_\varepsilon w\| \leq \|w\|\right) \geq (C_4 d^{-2} + O(d^{-4}))Q_s = C_5 d^{-2} + O(d^{-4}) \ ,$$

since $Q_s \geq Q_{1/\sqrt{2}} \geq C$ for $d \geq 2$ thanks to Assumption 36. This concludes the proof of (50). $\quad\square$

## J.2 Auxiliary Lemmas

*Proof of Lemma J.4.* Observe that
$$\mathbb{P}(\hat{x} \neq x^*) = 1 - \mathbb{P}(\hat{x} = x^*) = 1 - p(\hat{x}|y) \geq 1 - \max_x p(x|y) \, ,$$

with equality if $\hat{x}$ is the Maximum-a-Posteriori estimator. $\qquad\square$

*Proof of Lemma J.5.* First notice that we can reduce to a two-by-two matrix, since the directions orthogonal to both $u$ and $v$ clearly belong to an eigenspace of eigenvalue 1. The result follows directly by computing the characteristic equation $\det[A^\top A - \lambda I] = 0$. $\qquad\square$

*Proof of Lemma J.6.* First, observe that since the law of $X$ is rotationally invariant, we can assume without loss of generality that $x_1$ is proportional to $e_1^\top$, the first standard basis vector. Using the Schur complement, we have

$$X = \begin{pmatrix} \|x_1\|_2 & 0 \\ v & \bar{X} \end{pmatrix} \, , \text{ and } X^{-1} = \begin{pmatrix} \|x_1\|_2^{-1} & 0 \\ b & \bar{X}^{-1} \end{pmatrix} \, , \tag{62}$$

where $v$ is the $(d-1)$-dimensional vector given by $v_i = \|x_1\|_2^{-1}\langle x_1, x_{i+1}\rangle = x_{i+1,1} \sim N(0,1)$, $\bar{X}$ is a $(d-1) \times (d-1)$ matrix whose entries are drawn i.i.d. from $N(0,1)$, and $b = -\|x_1\|_2^{-1}\bar{X}^{-1}v$. Observe that $\bar{X}$ and $v$ are independent, since the choice of basis depends only on $x_1$. The coordinates of $v$ are independent as well for the same reason. It follows that

$$\begin{aligned}
\|\tilde{x}_1\|_2^2 &= \|x_1\|_2^{-2}\left(1 + \|\bar{X}^{-1}v\|_2^2\right) \\
&\leq \|x_1\|_2^{-2}\left(1 + \|\bar{X}^{-1}\|^2 \cdot \|v\|_2^2\right) \, ,
\end{aligned} \tag{63}$$

where $\|\bar{X}^{-1}\| = \max_{u \in S^{d-1}}\|\bar{X}^{-1}u\|_2$ is the operator norm of $\bar{X}^{-1}$. Now let $\alpha$ be a fixed constant, which will be specified later. Additionally, assume that $d$ is sufficiently large so that $\alpha d^4 \geq 2$. From Eq. (63), we have that

$$\begin{aligned}
\mathbb{P}\{\eta^2 \geq \alpha d^4\} &\leq \mathbb{P}\left\{\|x_1\|_2^2\left(\|x_1\|_2^{-2}\left(1 + \|\bar{X}^{-1}\|^2 \cdot \|v\|_2^2\right)\right) \geq \alpha d^4\right\} \\
&= \mathbb{P}\left\{1 + \|\bar{X}^{-1}\|^2 \cdot \|v\|_2^2 \geq \alpha d^4\right\} \\
&\leq \mathbb{P}\left\{\|\bar{X}^{-1}\|^2 \cdot \|v\|_2^2 \geq \alpha d^4/2\right\} \\
&= \mathbb{P}\left\{\|\bar{X}^{-1}\| \cdot \|v\|_2 \geq \sqrt{\alpha/2} \cdot d^2\right\} \, .
\end{aligned} \tag{64}$$

To upper bound Eq. (64), we use the fact that $\bar{X}^{-1}$ and $v$ are independent, and split the event into two cases: $\{\|v\|_2 \geq \sqrt{d}/2\}$ and $\{\|v\|_2 < \sqrt{d}/2\}$. By [A42, Theorem 3.1.1], we know that there exists a constant $C_1 > 0$ such that

$$\mathbb{P}\left\{\|v\|_2 < \sqrt{d}/2\right\} \leq \exp(-C_1 \cdot d) \, .$$

Moreover, by [A41, Theorem 1.2], we have that for sufficiently large $d$, there exists a universal constant $C_2 > 0$ such that for any $t > 0$,

$$\mathbb{P}\left\{\|\bar{X}^{-1}\| \geq t\sqrt{d}\right\} \leq C_2/t \, .$$

By setting $\alpha = 2C_2^2$ and $d$ sufficiently large so that $\exp(-C_1 d) \leq 1/(2d)$, we have

$$\begin{aligned}
\mathbb{P}\Big\{\|\bar{X}^{-1}\| \cdot \|v\|_2 \geq \sqrt{\alpha/2} \cdot d^2\Big\} & \\
&\leq \mathbb{P}\left\{\|\bar{X}^{-1}\| \cdot \sqrt{d}/2 \geq \sqrt{\alpha/2} \cdot d^2\right\} \cdot \mathbb{P}\left\{\|v\|_2 > \sqrt{d}/2\right\} + \mathbb{P}\left\{\|v\|_2 \leq \sqrt{d}/2\right\} \\
&\leq \mathbb{P}\left\{\|\bar{X}^{-1}\| \cdot \sqrt{d}/2 \geq \sqrt{\alpha/2} \cdot d^2\right\} + \exp(-C_1 d) \\
&= \mathbb{P}\left\{\|\bar{X}^{-1}\| \geq \sqrt{2\alpha} \cdot d^{3/2}\right\} + \exp(-C_1 d) \\
&\leq C_2/(\sqrt{2\alpha} \cdot d) + \exp(-C_1 d) \\
&\leq 1/d \, .
\end{aligned}$$

Therefore,

$$\mathbb{P}\{\eta \geq \sqrt{2}C_2 \cdot d^2\} = \mathbb{P}\{\eta^2 \geq 2C_2^2 \cdot d^4\} \leq \mathbb{P}\left\{\|\bar{X}^{-1}\| \cdot \|v\|_2 \geq C_2 \cdot d^2\right\} \leq 1/d \, .$$

$\qquad\square$

# K  Auxiliary Results

## K.1  The Periodic Gaussian

**Definition K.1.** *Let $\Psi_s(z) : [-1/2, 1/2) \to \mathbb{R}_+$ be the periodic Gaussian density function defined by*

$$\Psi_s(z) := \sum_{k=-\infty}^{\infty} \frac{1}{s\sqrt{2\pi}} \exp\left( -\frac{1}{2}\left(\frac{z-k}{s}\right)^2 \right) .$$

*We refer to the parameter $s$, the standard deviation of the Gaussian before periodicization, as the "width" of the periodic Gaussian $\Psi_s$.*

**Remark K.2.** *For intuition, we can consider two extreme settings of the width $s$. If $s \ll 1$, then $\Psi_s$ is close in total variation distance to the Gaussian of standard deviation $s$ since the tails outside $[-1/2, 1/2)$ will be very light. On the other hand, if $s \gg 1$, then $\Psi_s$ is close in total variation distance to the uniform distribution on $[0, 1)$. This intuition is formalized in Claim K.6.*

The Gaussian distribution on $\mathbb{R}$ satisfies the following tail bound called Mill's inequality.

**Lemma K.3** (Mill's inequality [A42, Proposition 2.1.2]). *Let $z \sim N(0, 1)$. Then for all $t > 0$, we have*

$$\mathbb{P}(|z| \geq t) = \sqrt{\frac{2}{\pi}} \int_t^\infty e^{-x^2/2} dx \leq \frac{1}{t} \cdot \sqrt{\frac{2}{\pi}} e^{-t^2/2} .$$

The Poisson summation formula, stated in Lemma K.5 below, will be useful in our calculations. We first define the dual of a lattice $\Lambda$ to make the formula easier to state.

**Definition K.4.** *The dual lattice of a lattice $\Lambda$, denoted by $\Lambda^*$, is defined as*

$$\Lambda^* = \{ y \in \mathbb{R}^d \mid \langle x, y \rangle \in \mathbb{Z} \text{ for all } x \in \Lambda \} .$$

A key property of the dual lattice is that if $B$ is a basis of $\Lambda$ then $(B^T)^{-1}$ is a basis of $\Lambda^*$; in particular, $\det(\Lambda^*) = \det(\Lambda)^{-1}$, where $\det(\Lambda)$ is defined as $\det(\Lambda) = \det(B)$ (the determinant of a lattice is basis-independent) [A31, Chapter 1].

For "nice" functions defined any lattice, the following formula holds [A10, Theorem 2.3].

**Lemma K.5** (Poisson summation formula). *For any lattice $\Lambda \subset \mathbb{R}^d$ and any function $f : \mathbb{R}^d \to \mathbb{C}$ satisfying some "niceness" assumptions[4]*

$$\sum_{x \in \Lambda} f(x) = \det(\Lambda^*) \cdot \sum_{y \in \Lambda^*} \widehat{f}(y) ,$$

*where $\widehat{f}(y) = \int_{\mathbb{R}^d} f(x) e^{-2\pi i \langle y, x \rangle} dx$, and $\Lambda^*$ is the dual lattice of $\Lambda$.*

Note that by the properties of the Fourier transform, for a fixed $c \in \mathbb{R}^d$

$$\sum_{x \in \Lambda + c} f(x) = \sum_{x \in \Lambda} f(x + c) = \det(\Lambda^*) \sum_{y \in \Lambda^*} \exp(-2\pi i \langle c, y \rangle) \cdot \widehat{f}(y) .$$

**Claim K.6** (Adapted from [A40, Claim 2.8.1]). *For any $s > 0$ and any $z \in [-1/2, 1/2)$ the periodic Gaussian density function $\Psi_s(z)$ satisfies*

$$\Psi_s(z) \leq \frac{1}{s\sqrt{2\pi}} \left( 1 + 2(1 + s^2) e^{-1/(2s^2)} \right) .$$

*and*

$$|\Psi_s(z) - 1| \leq 2(1 + 1/(4\pi s)^2) e^{-2\pi^2 s^2} .$$

---

[4]For our purposes, it suffices to know that the Gaussian function of any variance $s > 0$ satisfies this niceness assumption. Precise conditions can be found in [A10, Theorem 2.3].

*Proof.* We first derive an expression for $\Psi_s(0)$ using the Poisson summation formula. Note that the Fourier transform of $f(y) = \exp(-y^2/2)$ is given by $\widehat{f}(u) = \sqrt{2\pi} \cdot \exp(-2\pi^2 u^2)$. Moreover, viewing $\mathbb{Z}$ as a one-dimensional lattice, the determinant of the dual lattice $((1/s)\mathbb{Z})^* = s\mathbb{Z}$ is $s$. Hence,

$$
\begin{aligned}
\Psi_s(0) &= \frac{1}{s\sqrt{2\pi}} \sum_{y \in (1/s)\mathbb{Z}} \exp(-y^2/2) \\
&= \frac{\det(s\mathbb{Z})\sqrt{2\pi}}{s\sqrt{2\pi}} \cdot \sum_{u \in s\mathbb{Z}} \exp(-2\pi^2 u^2) \\
&= \sum_{u \in s\mathbb{Z}} \exp(-2\pi^2 u^2) \, .
\end{aligned}
\tag{65}
$$

We now observe that $\Psi_s(z) \leq \Psi_s(0)$ for any $z \in [-1/2, 1/2)$. This can again be shown using the Poisson summation formula as follows.

$$
\begin{aligned}
\Psi_s(z) &= \frac{1}{s\sqrt{2\pi}} \sum_{y \in (1/s)\mathbb{Z}+z/s} \exp(-y^2/2) \\
&= \sum_{u \in s\mathbb{Z}} \exp(-2\pi i u z/s) \cdot \exp(-2\pi^2 u^2) \\
&\leq \sum_{u \in s\mathbb{Z}} |\exp(-2\pi i u z/s)| \cdot \exp(-2\pi^2 u^2) \\
&\leq \sum_{u \in s\mathbb{Z}} \exp(-2\pi^2 u^2) \\
&= \Psi_s(0) \, .
\end{aligned}
$$

Hence, it suffices to upper bound $\Psi_s(0)$ and show a lower bound for $\Psi_s(z)$ for all $z \in [-1/2, 1/2)$. For the first upper bound, we use Mill's inequality (Lemma K.3) to obtain

$$
\begin{aligned}
\Psi_s(0) &= \frac{1}{s\sqrt{2\pi}} \sum_{y \in (1/s)\mathbb{Z}} \exp(-y^2/2) \\
&\leq \frac{1}{s\sqrt{2\pi}} \left( 1 + 2\exp(-1/(2s^2)) + 2 \int_1^\infty \exp(-x^2/(2s^2)) dx \right) \\
&\leq \frac{1}{s\sqrt{2\pi}} \left( 1 + 2(1+s^2)\exp(-1/(2s^2)) \right) \, .
\end{aligned}
$$

For the second upper bound, we use Eq. (65) and Mill's inequality to obtain

$$
\begin{aligned}
\Psi_s(0) &= \sum_{u \in s\mathbb{Z}} \exp(-2\pi^2 u^2) \\
&= 1 + \sum_{u \in s\mathbb{Z} \setminus \{0\}} \exp(-2\pi^2 u^2) \\
&= 1 + 2 \sum_{k=1}^\infty \exp(-2\pi^2 s^2 k^2) \\
&\leq 1 + 2\exp(-2\pi^2 s^2) + 2 \int_1^\infty \exp(-2\pi^2 s^2 x^2) dx \\
&\leq 1 + 2(1 + 1/(4\pi s)^2)\exp(-2\pi^2 s^2) \, .
\end{aligned}
$$

For the lower bound on $\Psi_s(z)$, we use the Poisson summation formula and Mill's inequality again to obtain

$$\Psi_s(z) = \sum_{u \in s\mathbb{Z}} \exp(-2\pi i z u / s) \cdot \exp(-2\pi^2 u^2)$$

$$= 1 + \sum_{u \in s\mathbb{Z} \setminus \{0\}} \exp(-2\pi i z u / s) \cdot \exp(-2\pi^2 u^2)$$

$$\geq 1 - 2 \sum_{k=1}^{\infty} |\exp(-2\pi i z k)| \cdot \exp(-2\pi^2 s^2 k^2)$$

$$\geq 1 - 2 \left( \exp(-2\pi^2 s^2) + \int_1^{\infty} \exp(-2\pi^2 s^2 x^2) dx \right)$$

$$\geq 1 - 2(1 + 1/(4\pi s)^2) \exp(-2\pi^2 s^2) .$$

$\square$

## K.2 Auxiliary Lemmas for the Constant Noise Regime

**Lemma K.7.** *Fix some $\tau \in (0, 1]$. Then, for $\arccos : [-1, 1] \to [0, \pi]$ it holds that*

$$\sup_{x, y \in [-1,1], |x-y| \leq \tau} |\arccos(x) - \arccos(y)| \leq \arccos(1 - \tau).$$

*Proof.* Let us fix some arbitrary $\xi \in [0, \tau]$ and consider the function $G(x) = \arccos(x) - \arccos(x + \xi)$. Given the fact that $\arccos$ is decreasing, it suffices to show that $|G(x)| \leq \arccos(1 - \tau)$ for all $x \in [-1, 1 - \xi]$. By direct computation it holds

$$G'(x) = -\frac{1}{\sqrt{1 - x^2}} + \frac{1}{\sqrt{1 - (x + \xi)^2}}$$

$$= \frac{\xi(2x + \xi)}{\sqrt{1 - x^2} \sqrt{1 - (x + \xi)^2} (\sqrt{1 - x^2} + \sqrt{1 - (x + \xi)^2})}.$$

Hence, the function $G$ decreases until $x = -\xi/2$ and increases beyond this point. Consequently, $G$ obtains its global maximum at one the endpoints of $[-1, 1 - \xi]$. But since $\cos(\pi - a) = -\cos(a)$ for all $a \in \mathbb{R}$ it also holds for all $b \in [-1, 1]$ $\arccos(-b) + \arccos(b) = \pi$. Hence,

$$G(-1) = \pi - \arccos(-1 + \xi) = \arccos(1 - \xi) = G(1 - \xi).$$

Therefore,

$$G(x) \leq \arccos(1 - \xi) \leq \arccos(1 - \tau).$$

The proof is complete. $\square$

## K.3 Auxiliary Lemmas for the Exponentially Small Noise Regime

**Lemma K.8.** *[Restated Lemma E.7] Suppose $n \leq C_0 d$ for some constant $C_0 > 0$ and $s \in \mathbb{R}^n$ satisfies for some $m \in \mathbb{Z}^n$ that $|\langle m, s \rangle| = \exp(-\Omega((d \log d)^3))$. Then for some sufficiently large constant $C > 0$, if $N = \lceil d^3 (\log d)^2 \rceil$ there is an $m' \in \mathbb{Z}^{n+1}$ which is equal with $m$ in the first $n$ coordinates, satisfies $\|m'\|_2 \leq C d^{\frac{1}{2}} \|m\|_2$ and is an integer relation for the $(s_1)_N, \ldots, (s_n)_N, 2^{-N}$.*

*Proof.* We start with noticing that since $N = o((d \log d)^3)$ we have

$$|\langle m, s \rangle| \leq \exp(-\Omega((d \log d)^3)) = O(2^{-N}) .$$

Hence, since for any real number $x$ we have $|x - (x)_N| \leq 2^{-N}$, it holds

$$\sum_{i=1}^{n} m_i(s_i)_N = \sum_{i=1}^{n} m_i s_i + O(\sum_{i=1}^{n} m_i 2^{-N})$$

$$= O(2^{-N}) + O(\sum_{i=1}^{n} |m_i| 2^{-N})$$

$$= O(\sum_{i=1}^{n} |m_i| 2^{-N}).$$

Now observe that the number $\sum_{i=1}^{n} m_i(s_i)_N$ is a rational number of the form $a/2^N, a \in \mathbb{Z}$. Hence using the last displayed equation we can choose some integer $m'_{n+1}$ with

$$\sum_{i=1}^{n} m_i(s_i)_N = m'_{n+1} 2^{-N}.$$

for which using Cauchy-Schwartz and $n = O(d)$ it holds

$$|m'_{n+1}| = O(\|m\|_1) = O(\sqrt{n}\|m\|_2) = O(\sqrt{d}\|m\|_2).$$

Hence $m' = (m_1, \ldots, m_n, -m'_{n+1})$ is an integer relation for $(s_1)_N, \ldots (s_n)_N, 2^{-N}$. On top of that

$$\|m'\|_2^2 \leq \|m\|_2^2 + O(d\|m\|_2^2) = O(d\|m\|_2^2).$$

This completes the proof. $\qquad\square$

**Lemma K.9** (Restated Lemma E.8). *Suppose that $\gamma \leq d^Q$ for some $Q > 0$. For some hidden direction $w \in S^{d-1}$ we observe $d + 1$ samples of the form $(x_i, z_i), i = 1, \ldots, d+1$ where for each $i$, $x_i$ is a sample from $N(0, I_d)$ samples, and*

$$z_i = \cos(2\pi(\gamma\langle w, x_i \rangle)) + \xi_i,$$

*for some unknown and arbitrary $\xi_i \in \mathbb{R}$ satisfying $|\xi_i| \leq \exp(-(d \log d)^3)$. Denote by $X \in \mathbb{R}^{d \times d}$ the random matrix with columns given by the $d$ vectors $x_2, \ldots, x_{d+1}$. With probability $1 - \exp(-\Omega(d))$ the following properties hold.*

*(1)*

$$\max_{i=1,\ldots,d+1} \|x_i\|_2 \leq 10\sqrt{d}.$$

*(2)*

$$\min_{i=1,\ldots,d+1} |\sin(2\pi\gamma\langle x_i, w \rangle)| \geq 2^{-d}.$$

*(3) For all $i = 1, \ldots, d+1$ it holds $z_i \in [-1, 1]$ and*

$$z_i = \cos(2\pi(\gamma\langle x_i, w \rangle + \xi'_i)),$$

*for some $\xi'_i \in \mathbb{R}$ with $|\xi'_i| = \exp(-\Omega((d \log d)^3))$.*

*(4) The matrix $X$ is invertible. Furthermore,*

$$\|X^{-1}x_1\|_\infty = O(2^{\frac{d}{2}}\sqrt{d}).$$

*(5)*

$$0 < |\det(X)| = O(\exp(d \log d)).$$

*Proof.* For the first part, notice that for each $i = 1, 2, \ldots, d+1$, the quantity $\|x_i\|_2^2$ is distributed like a $\chi^2(d)$ distribution with $d$ degrees of freedom. Using standard results on the tail of the $\chi^2$ distribution (see e.g. [A43, Chapter 2]) we have for each $i$,

$$\mathbb{P}\left(\|x_1\|_2 \geq 10\sqrt{d}\right) = \exp(-\Omega(d)).$$

Hence,

$$\mathbb{P}\left(\bigcup_{i=1}^{d+1} \|x_i\|_2 \geq 10\sqrt{d}\right) \leq (d+1)\mathbb{P}\left(\|x_1\|_2 \geq 10\sqrt{d}\right) = O(d\exp^{-\Omega(d)}) = \exp(-\Omega(d)),$$

For the second part, first notice that for large $d$ the following holds: if for some $\alpha \in \mathbb{R}$ we have $|\sin(\alpha)| \leq 2^{-d}$ then for some integer $k$ it holds $|\alpha - k\pi| \leq 2^{-d+1}$. Indeed, by substracting an appropriate integer multiple of $\pi$ we have $\alpha - k\pi \in [-\pi/2, \pi/2]$. Now by applying the mean value theorem for the branch of arcsin defined with range $[-\pi/2, \pi/2]$ we have that

$$|\alpha - k\pi| = |\arcsin(\sin\alpha) - \arcsin(0)| \leq \frac{1}{\sqrt{1-\xi^2}}|\sin\alpha| \leq \frac{1}{1-\xi^2}2^{-d}$$

for some $\xi$ with $|\xi| \leq |\sin\alpha| \leq 2^{-d}$. Hence, using the bound on $\xi$ we have

$$|\alpha - k\pi| \leq \frac{1}{1-2^{-2d}}2^{-d} \leq 2^{-d+1}\ .$$

Using the above observation, we have that if for some $i$ it holds $|\sin(2\pi\gamma\langle x_i, w\rangle)| \leq 2^{-d}$ then for some integer $k \in \mathbb{Z}$ it holds $|\langle x_i, w\rangle - \frac{k}{2\gamma}| \leq \frac{1}{\gamma}2^{-d}$. Furthermore, since by Cauchy-Schwartz and the first part with probability $1 - \exp(-\Omega(d))$ we have

$$|\langle x_i, w\rangle| \leq \|x_i\| \leq 10\sqrt{d},$$

it suffices to consider only the integers $k$ satisfying $|k| \leq 10\gamma\sqrt{d}$, with probability $1 - \exp(-\Omega(d))$. Hence,

$$
\begin{aligned}
\mathbb{P}\left(\bigcup_{i=1}^{d+1} |\sin(2\pi\gamma\langle x_i, w\rangle)| \leq 2^{-d}\right) &\leq \mathbb{P}\left(\bigcup_{i=1}^{d+1}\bigcup_{k:|k|\leq 10\gamma\sqrt{d}} |\langle x_i, w\rangle - \frac{k}{2\gamma}| \leq \frac{1}{\gamma}2^{-d}\right) \\
&\leq 20d\sqrt{d}\gamma \sup_{k\in\mathbb{Z}} \mathbb{P}\left(|\langle x_1, w\rangle - k/2\gamma| \leq \frac{1}{\gamma}2^{-d}\right) \\
&\leq 40d\sqrt{d}2^{-d} \\
&= \exp(-\Omega(d)),
\end{aligned}
$$

where we used the fact that $\langle x_1, w\rangle$ is distributed as a standard Gaussian, and that for a standard Gaussian $Z$ and for any interval $I$ of any interval of length $t$ it holds $\mathbb{P}(Z \in I) \leq \frac{1}{\sqrt{2\pi}}t \leq t$.

For the third part, notice that from the second part for all $i = 1, \ldots, d+1$ it holds

$$1 - \cos^2(2\pi\gamma\langle x_i, w\rangle) = \sin^2(2\pi\gamma\langle x_i, w\rangle) = \Omega(2^{-2d})$$

with probability $1 - \exp(-\Omega(d))$. Hence, since $\|\xi\|_\infty \leq \exp(-(d\log d)^3)$ we have that for all $i = 1, \ldots, d+1$ it holds

$$z_i = \cos(2\pi\gamma\langle x_i, w\rangle)) + \xi_i \in [-1, 1],$$

with probability $1 - \exp(-\Omega(d))$. Hence, the existence of $\xi_i'$ follows by the fact that image of the cosine is the interval $[-1, 1]$. Now by mean value theorem we have

$$\xi_i = \cos(2\pi(\gamma\langle x_i, w\rangle + \xi_i')) - \cos(2\pi\gamma\langle x_i, w\rangle)) = 2\pi\gamma\xi_i'\sin(2\pi\gamma t)$$

for some $t \in (\langle x_i, w\rangle - |\xi_i|, \langle x_i, w\rangle + |\xi_i|)$. By the 1-Lipschitzness of the sine function, the second part and the exponential upper bound on the noise we can immediately conclude

$$|\sin(2\pi\gamma t)| \geq \sin(2\pi\gamma\langle x_i, w\rangle) - |\xi_i| = \Omega(2^{-d}),$$

with probability $1 - \exp(-\Omega(d))$. Hence it holds $|\xi_i'|\Omega(2^{-d}) \leq |\xi_i|$ and therefore

$$|\xi_i'| \leq 2^d|\xi_i| = \exp(-\Omega((d\log d)^3))$$

with probability $1 - \exp(-\Omega(d))$.

For the fourth part, for the fact that $X$ is invertible, consider its determinant, that is the random variable $\det(X)$. The determinant is non-zero almost surely, i.e. $\det(X) \neq 0$ almost surely. This follows from the fact that the determinant is a non-zero polynomial of the entries of $X$, e.g. for $X = I_d$ it equals one, hence, using folklore results as all entries of $X$ are i.i.d. standard Gaussian it is almost surely non-zero [A8]. Now, using standard results on the extreme singular values of $X$, such as [A37, Equation (3.2)], we have that $\sigma_{\max}(X^{-1}) = 1/\sigma_{\min}(X) \leq 2^d$, with probability $1 - \exp(-\Omega(d))$. In particular, using also the first part, it holds

$$\|X^{-1}x_1\|_\infty \leq \|X^{-1}x_1\|_2 \leq \sqrt{\sigma_{\max}(X^{-1})}\|x_1\|_2 \leq 2^{\frac{d}{2}}\sqrt{d},$$

with probability $1 - \exp(-\Omega(d))$.

For the fifth part, notice that the determinant is non-zero from the fourth part.

For the upper bound on the determinant, we apply Hadamard's inequality [A19] and part 1 of the Lemma to get that

$$|\det(x_2, \ldots, x_{d+1})| \leq \prod_{i=2}^{d+1} \|x_i\|_2 \leq (10\sqrt{d})^d = O(\exp(d \log d)),$$

with probability $1 - \exp(-\Omega(d))$.

$\square$

### K.4   Auxiliary Lemmas for the Population Loss

Fix some hidden direction $w \in S^{d-1}$. Recall that for any $w' \in S^{d-1}$, we denote by

$$L(w') = \mathbb{E}_{x \sim N(0, I_d)}[(\cos(2\pi\gamma\langle w, x\rangle) - \cos(2\pi\gamma\langle w', x\rangle))^2] .$$

**Lemma K.10.** *Let us consider the (probabilist's) normalized Hermite polynomials on the real line $\{h_k\}_{k \in \mathbb{Z}_{\geq 0}}$. The following identities hold for $Z \sim N(0, 1)$.*

*(1) For all $k, \ell \in \mathbb{Z}_{\geq 0}$*

$$\mathbb{E}[h_k(Z)h_\ell(Z)] = \mathbb{1}[k = \ell] .$$

*(2) Let $Z_\rho$ be a standard Gaussian which is $\rho$-correlated with $Z$. Then, for all $\gamma > 0, k \in \mathbb{Z}_{\geq 0}$,*

$$\mathbb{E}[h_k(Z)\cos(2\pi\gamma Z_\rho)] = (-1)^{k/2}\rho^k \frac{(2\pi\gamma)^k}{\sqrt{k!}} \exp(-2\pi^2\gamma^2) \cdot \mathbb{1}[k \in 2\mathbb{Z}_{\geq 0}] .$$

*(3) The performance of the trivial estimator, which always predicts 0, equals*

$$\mathrm{Var}(\cos(2\pi\gamma Z)) = \sum_{k \in 2\mathbb{Z}_{\geq 0}\setminus\{0\}} \frac{(2\pi\gamma)^{2k}}{k!} \exp(-4\pi^2\gamma^2) = \frac{1}{2} + O(\exp(-\Omega(\gamma^2))) .$$

*Proof.* The first part follows from the standard property that the family of normalized Hermite polynomials form a complete orthonormal basis of $L^2(N(0, 1))$ [A23, Proposition B.2].

For the second part, recall the basic fact that we can set $Z_\rho = \rho Z + \sqrt{1 - \rho^2} W$ for some $W$ standard Gaussian independent from $Z$. Using [A23, Proposition 2.10], we get

$$
\begin{aligned}
\mathbb{E}[h_k(Z)\cos(2\pi\gamma Z_\rho)] &= \mathbb{E}[h_k(Z)\cos(2\pi\gamma(\rho Z + \sqrt{1-\rho^2}W))] \\
&= \frac{1}{\sqrt{k!}}\mathbb{E}\left[\frac{d^k}{dZ^k}\cos(2\pi\gamma(\rho Z + \sqrt{1-\rho^2}W))\right] \\
&= (-1)^{k/2}(2\pi\rho\gamma)^k \frac{1}{\sqrt{k!}}\mathbb{E}[\cos(2\pi\gamma(\rho Z + \sqrt{1-\rho^2}W))] \cdot \mathbb{1}(k \in 2\mathbb{Z}_{\geq 0}) \\
&\quad + (-1)^{(k+1)/2}(2\pi\rho\gamma)^k \frac{1}{\sqrt{k!}}\mathbb{E}[\sin(2\pi\gamma(\rho Z + \sqrt{1-\rho^2}W))] \cdot \mathbb{1}(k \notin 2\mathbb{Z}_{\geq 0}) \\
&= (-1)^{k/2}(2\pi\rho\gamma)^k \frac{1}{\sqrt{k!}}\mathbb{E}[\cos(2\pi\gamma(\rho Z + \sqrt{1-\rho^2}W))] \cdot \mathbb{1}(k \in 2\mathbb{Z}_{\geq 0}) \\
&= (-1)^{k/2}(2\pi\rho\gamma)^k \frac{1}{\sqrt{k!}}\mathbb{E}[\cos(2\pi\gamma Z)] \cdot \mathbb{1}(k \in 2\mathbb{Z}_{\geq 0}) \\
&= (-1)^{k/2}(2\pi\rho\gamma)^k \frac{1}{\sqrt{k!}}\exp(-2\pi^2\gamma^2) \cdot \mathbb{1}(k \in 2\mathbb{Z}_{\geq 0}) \,,
\end{aligned}
$$

where (a) in the third to last line we used that the $\sin$ is an odd function and therefore when $k$ is odd the corresponding term is zero, (b) in the second to last line we used that $Z_\rho$ follows the same standard Gaussian law as $Z$ and, (c) in the last line we used the characteristic function of the standard Gaussian to conclude that for any $t > 0$,

$$
\mathbb{E}[\cos(tZ)] = \mathrm{Re}[\mathbb{E}[e^{itZ}]] = e^{-t^2/2} \,.
$$

For the third part, notice that by applying the result from part (1) and the result from part (2) (for $\rho = 1$) it holds,

$$
\begin{aligned}
\mathrm{Var}(\cos(2\pi\gamma Z)) &= \sum_{k \in \mathbb{Z}_{\geq 0} \setminus \{0\}} \mathbb{E}[\cos(2\pi\gamma Z)h_k(Z)]^2 \\
&= \sum_{k \in 2\mathbb{Z}_{\geq 0} \setminus \{0\}} \frac{(2\pi\gamma)^{2k}}{k!} \exp(-4\pi^2\gamma^2) \\
&= \sum_{k \in 2\mathbb{Z}_{\geq 0}} \frac{(2\pi\gamma)^{2k}}{k!} \exp(-4\pi^2\gamma^2) - \exp(-4\pi^2\gamma^2) \\
&= \sum_{k \geq 0} \frac{1}{2} \cdot \frac{(2\pi\gamma)^{2k}}{k!} \exp(-4\pi^2\gamma^2)(1 + (-1)^k) - \exp(-4\pi^2\gamma^2) \\
&= \frac{1}{2}\left(\sum_{k \geq 0} \frac{(4\pi^2\gamma^2)^k}{k!}\exp(-4\pi^2\gamma^2) + \sum_{k \geq 0}\frac{(-4\pi^2\gamma^2)^k}{k!}\exp(-4\pi^2\gamma^2)\right) - \exp(-4\pi^2\gamma^2) \\
&= \frac{1}{2} + \frac{1}{2}\exp(-8\pi^2\gamma^2) - \exp(-4\pi^2\gamma^2) \\
&= \frac{1}{2} + O(\exp(-\Omega(\gamma^2))) \,.
\end{aligned}
$$

$\square$