# OpenReview forum: "On the Cryptographic Hardness of Learning Single Periodic Neurons"
_NeurIPS.cc/2021/Conference — NeurIPS 2021 Poster_

### Official Review · Reviewer_sRgS · 2021-07-11

**Rating:** 7
**Confidence:** 3

**Summary:**

The authors consider learning a single periodic neuron over isotropic Gaussian distributions in the presence of noise. The authors connect single periodic neurons with lattice basis reduction algorithms and CLWE, and discuss noise level regions where estimation is impossible, crypto-hard or easy.

**Limitations And Societal Impact:**

No discussions on societal impact are available.

**Main Review:**

1. Originality

The task is not new but the method is new. The connection between this problem with CLWE for establishing cryptographic hardness under moderate noise is interesting. Related work is adequately cited and well discussed.

2. Quality

The submission is technically sound and claims are well supported. Weaknesses are not addressed in depth, and some questions are raised in (Section 3. Clarity). Overall the paper is well-written and easy to follow. Some sentences seem to appear several times throughout the paper, but it does not affect the flow of reading.

3. Clarity

The paper is clearly written and well organized. The history and motivations of discussing this problem is also given.

Some minor questions regarding the work:

(a) Considering the gaps between impossible, crypto hard and easy regions, is it possible to strengthen the result to reduce the gaps? For example, the information theoretic bound of 1 versus the crypto hard bound of omega(1) could be quite large. It would be interesting to see if the techniques can be modified or strengthened to reduce this gap. The same goes for the gap between the crypto hard region and easy region: the gap between poly(d) and exp((dlog d)^3) is quite large and it would be interesting to see if there is any discussion about these gaps.

(b) It would be nice to see some more discussion about practical implications of such a result for applications, particularly for readers not familiar with hardness of learning. Readers not within the community may find it hard to justify the importance or interestingness of the result, since intuitively speaking it is quite natural to have regions of the noise level where it is easy or hard to estimate the underlying vector w. I think it would be beneficial to add some more discussion relating practical implications of this work.

4. Significance:

The results are interesting and benefit the learning community. The methodologies can bring impact to a range of other problems.

Update after reviewing comments: Increased score to 7.


**Time Spent Reviewing:**

3 hours

---

> ### Author Response · Authors · 2021-08-09
> **Author Response**
>
> We thank the reviewer for bringing up great comments and questions that would certainly make our work accessible and convincing to a wider audience.
>
> ***"Considering the gaps between impossible, crypto hard and easy regions, is it possible to strengthen the result to reduce the gaps? For example, the information theoretic bound of 1 versus the crypto hard bound of omega(1) could be quite large. It would be interesting to see if the techniques can be modified or strengthened to reduce this gap."***
>
> We thank the reviewer for bringing this point up. Our Figure 1 was slightly misleading because of the gap we mistakenly introduced between the "impossible" and "crypto-hard" region: the crypto-hard region also contains the case where 1/\beta=O(1), increasing the noise-level can only “increase” the algorithmic hardness. We will update our Figure 1 to fix this mistake.
>
> ***"The same goes for the gap between the crypto hard region and easy region: the gap between poly(d) and exp((dlog d)^3) is quite large and it would be interesting to see if there is any discussion about these gaps."***
>
> This is certainly an interesting open question. We recall to the reviewer that if such a noise regime were learnable by some algorithm, that algorithm would have to exploit individual samples in a highly non-trivial way as in the LLL algorithm in order to circumvent the superpolynomial SQ lower bound [Son+17]. Although a mild, but technically challenging, improvement could be possible, we do not think our current LLL-based algorithm would be able to provably obtain noise tolerance which is exp(o(d)).
>
> To strengthen this point, we conjecture that this “subexponential-small noise” regime is still hard for polynomial-time algorithms. Given CLWE's analogy to the LWE problem [Reg05], it is plausible that the quantum reduction for CLWE (and hence cosine learning) also applies for the sub-exponentially small noise regime, as in the quantum reduction from SVP to LWE. If this is true, a polynomial-time algorithm for our cosine learning problem with sub-exponentially small noise would yield a “breakthrough” quantum algorithm for approximate SVP, since no polynomial-time algorithms are known to achieve subexponential approximation factors. We will add this part in the revised version of our work.
>
> ***"It would be nice to see some more discussion about practical implications of such a result for applications, particularly for readers not familiar with hardness of learning."***
>
> We view our work as having *indirect* implications since we attempt to quantitatively understand the inherent difficulty of a deceptively simple-looking high dimensional learning task in our paper. As we scale-up our community’s learning models, e.g. the case of BERT and GPT-3, the ML community has observed almost monotonic improvements in the performance of these massive models, with some empirical papers even showing that the progress in performance tends to follow certain "scaling-laws" [Kap+20, Figure 1]. Extrapolating from this trend, it is plausible for anyone interested in the practice of machine learning to ask the following question:
>
> "Is learning always possible up to the "statistical limit", once our model is large enough to model the target function, and we have massive quantities of high-quality data samples, corrupted by very little noise?"
>
> Our message in this direction is that there are deceptively simple learning tasks that are possible statistically and yet potentially lie outside the reach of billions of years of computation time on even the world's fastest computing hardware. In fact, we argue that the hard learning task that demonstrates this hardness result is rather surprising due to its apparent simplicity (it certainly was surprising for us); the target function is nothing but a cosine applied to a linear projection of the data, the input is distributed as the standard d-dimensional Gaussian, and the amount of label noise added is very small.
>
> Of course, we acknowledge that the learning task we introduce in our paper is far-fetched from the learning problems encountered in practice, which is why we do not claim *direct* practical impact. Yet, we believe that the theoretical existence of this "needle" in the "haystack" of learning tasks is valuable in itself.
>
> Practical considerations aside, we believe our work, and more generally the study of hardness of learning, has some intrinsic appeal for our innate curiosity because they provide partial, yet precise answers to the following natural question:
>
> "How hard can robust learning be if I assume that the label is generated by a "simple" target function, and the input data distribution is as "nice" as can be?"
>
> ***"... intuitively speaking it is quite natural to have regions of the noise level where it is easy or hard to estimate the underlying vector w."***
>
> We would like to respectfully push back a little on this point. Some high-dimensional inference problems do not exhibit computational-to-statistical gaps (e.g. regular PCA [BBP05]). It’s important to distinguish whether the apparent intractability of some settings is an information-theoretic one (i.e., no algorithm can solve the problem even when given infinite time and computational budget) or a computational one. Such an understanding can guide us to efficiently allocate our (potentially very valuable) computational resources where they are truly necessary. While our work focuses on the study of a comp-to-stats gap relevant to neural networks (NN), let us practically motivate their study with another timely example.
>
> A case of particular recent interest is the model of group or pooled testing, where one seeks to “infer” a subset of infected individuals out of a large total population, by testing groups/subsets of them at once. While the “classic” information-theoretic/statistical inference question is how many test kits are needed for the infected individuals to be recovered by some algorithm, the actual practically relevant question is how many kits are necessary for a computationally efficient inference algorithm to succeed. The natural discrepancy between these two questions is the reason that led to the rigorous study of the comp-to-stats gap of group testing (e.g., [IZ21] and references therein).
>
> Similar to group testing, we hope that our study of the comp-to-stats gap of cosine linear models will shed some light on the signal-to-noise ratios that allow computationally-efficient inference in NN-relevant high dimensional models, a fundamentally important question in the machine learning community. We will add this discussion to the revised version of our work.
>
> ### References
> * [Son+17] Le Song, Santosh Vempala, John Wilmes, and Bo Xie. *On the Complexity of Learning Neural Networks*. NeurIPS 2017.
> * [Reg05] Oded Regev. *On lattices, learning with errors, random linear codes, and cryptography*. STOC 2005.
> * [Kap+20] Jared Kaplan et al. *Scaling laws for neural language models*. arxiv, 2020.
> * [IZ21] Fotis Iliopoulos, Ilias Zadik. *Group testing and local search: is there a computational-statistical gap?* COLT 2021.
> * [BBP05] Jinho Baik, Gerard Ben Arous, and Sandrine Peche. *Phase transition of the largest eigenvalue for nonnull complex sample covariance matrices*. The Annals of Probability 33(5), 2005.

---

> > ### Comment · Reviewer_sRgS · 2021-08-25
> > **Response to comments**
> >
> > Thanks for the detailed responses---I am increasing my score to 7. The responses are nicely written and address my concerns well. I believe including some of these insights into the main paper in future revisions would be a nice addition!

---

### Official Review · Reviewer_Dpoz · 2021-07-16

**Rating:** 8
**Confidence:** 4

**Summary:**

This paper considers a learning problem of the following form: given noisy labeled examples $(x, z)$ where $x \sim N(0, I)$ and $z = \cos(2\pi \gamma \langle w, x \rangle) + \xi$ (here $|\xi| \leq \beta$ is adversarial, additive, bounded noise), learn the hidden parameter $w$ (or at least weakly learn the mapping in terms of square loss). Thus $z$ is the (noisy) output of a single neuron with a periodic activation under Gaussian inputs. The main results are:
- In the "moderate noise" $\beta = 1/\text{poly}(d)$ regime, a reduction showing that a learner for this problem could solve continuous LWE in a regime believed to be hard.
- In the "small noise" $\beta = \exp(-d)$ regime, a novel efficient learning algorithm using LLL-based integer relation detection.

As one consequence, the hardness profile of this problem with respect to the noise rate $\beta$ exhibits interesting computational-to-statistical phase traditions / gaps.

**Limitations And Societal Impact:**

The authors are clear about the scope and limitations of this result. In the main review, I've mentioned a couple of points I would be interested to see some further discussion of, space permitting. I am not aware of any potential negative societal impact of this work.

Nit: In the statement of the cryptographic lower bound in some places in the text and e.g. Cor 4.4, I think it confuses matters a bit to speak of quantum attacks. The reduction shows that a classical poly-time learner of periodic neurons would yield a classical poly-time algorithm for CLWE. As one cares primarily about classical learners of such neurons, hardness for such learners holds under the assumption that CLWE is classically hard, which is of course a _weaker_ assumption than that it is quantum hard. That is, the headline lower bound of this paper really holds under a weaker assumption than some places in the text make it sound. If one did care about quantum learners of periodic neurons, then quantum attacks against CLWE would be of interest, but then one gets a little bit into the weeds of the representation of the data the quantum learner is given, etc. I understand the intent is probably to provide evidence for the weaker assumption (classical hardness of CLWE) by saying people actually believe the stronger one (quantum hardness of CLWE), but the wording could perhaps be tweaked in some places.

**Main Review:**

There has been a lot of work on learning deep learning primitives such as single neurons in various models recently, especially distribution-specific SQ and gradient-based methods. While such methods are quite general, it has been often noted that they technically exclude approaches such as SGD. This paper finds a new and interesting place in this landscape by showing cryptographic lower bounds that hold against arbitrary poly-time learners under a benign input distribution, namely the Gaussian. The proof of the cryptographic lower bound is clean and elegant, and the use of continuous LWE for the reduction is novel and may prove to be a technique of further interest in this area.

Regarding the class of periodic neurons, while it is true that they have been considered in prior works such as Song et al 2017 and have interesting theoretical properties, I still find them a little artificial from a practical point of view (at the least, one tends to want an increasing activation).  But this is a minor shortcoming considering this is still a new and strong result, and one would hope that results such as this will lead to better understanding of the landscape for more usual neurons. The authors do note that the cosine neuron is well-approximated by one-layer ReLU networks. The cryptographic lower bounds that one would obtain as a corollary for such one-layer networks (with additive noise) are against poly-time learners, and hence nominally weaker than (although technically incomparable with) known exp(n) SQ lower bounds in the realizable setting (e.g., Diakonikolas et al, 2020). I would be interested to see a short discussion of such a quantitative comparison, especially as someone unfamiliar with the hard regime for CLWE.

The LLL-based learning algorithm is also novel and an important technical contribution. It would appear that the periodicity makes some such technique essential. If space permits, I would be interested to see a brief comparison with existing approaches for non-periodic activations and GLMs at least at a high level.

Overall I think this is a significant and original contribution to the literature. The paper is well-written and clearly organized. Some of the proofs (esp the supporting lemmas in Appendix D) are somewhat technical and I did not verify them in full detail, but they appear correct to me.

**Time Spent Reviewing:**

4

---

> ### Author Response · Authors · 2021-08-09
> **Author Response**
>
> ***“The proof of the cryptographic lower bound is clean and elegant, and the use of continuous LWE for the reduction is novel and may prove to be a technique of further interest in this area.”***
>
> We thank the reviewer for a thorough, thoughtful, and positive review. We especially appreciate the questions the reviewer brought up regarding how our result compares to previous work using different activations (e.g., non-periodic activations), and model of computation (e.g., SQ), as these comparisons clarify where our result stands in the hardness of learning landscape shaped by previous works.
>
> ***“The cryptographic lower bounds ... are against poly-time learners, and hence nominally weaker than (although technically incomparable with) known exp(n) SQ lower bounds in the realizable setting (e.g., Diakonikolas et al, 2020). I would be interested to see a short discussion of such a quantitative comparison, especially as someone unfamiliar with the hard regime for CLWE.”***
>
> This is a great question which gives us the opportunity to clarify further some aspects of CLWE hardness and hardness of cosine learning. Unlike SQ lower bounds which give unconditional quantitative lower bounds, the quantum reduction which shows the average-case hardness of CLWE does not necessarily yield quantitative unconditional lower bounds. To be more precise, whether or not the reduction leads to a quantitative lower bound is conditional on an underlying computational assumption. For CLWE (and hence, cosine learning), this computational assumption is the following _worst-case_ hardness assumption:
>
> “There are no polynomial-time quantum algorithms that approximate the Shortest Vector Problem (SVP) up to polynomial factors.”
>
> This is akin to how an NP-hard reduction does not by itself give a quantitative lower bound. In fact, this is the reason researchers rely on stronger quantitative conjectures, such as the Exponential-Time Hypothesis (ETH) or Strong Exponential-Time Hypothesis (SETH), to prove concrete (conditional) lower bounds for computational problems [Wil17]. We emphasize the phrase “worst-case” as we believe that average-case hardness based on worst-case problems is _qualitatively_ significant.
>
> Of course, one could assume quantitative lower bounds, say exp(d) or exp(\sqrt{d}), on solving poly-factor approximate-SVP or CLWE directly to get concrete quantitative conditional lower bounds for cosine learning. However, we refrain from making such stronger assumptions as we do not believe that there is a strong consensus on a concrete quantitative lower bound for approximate SVP, other than the belief that it is unlikely to be solvable in poly-time [MR09, Conjecture 1.2]. We will add this discussion in the revised version of our work.
>
> ***“If space permits, I would be interested to see a brief comparison with existing approaches for non-periodic activations and GLMs at least at a high level.”***
>
> Several previous works have established strong positive learning results for monotonic Lipschitz activations. For instance, the GLM-tron algorithm (reminiscent of the classical perceptron algorithm) by [Kak+11] for learning GLMs with such activations, and polynomial-time approximation schemes (PTAS) for ReLU regression over log-concave distributions by [Dia+20a]. At a high-level, these works use the fact that one can design convex surrogate losses with nice properties for monotonic Lipschitz activations [Kan18], and use projected gradient descent on this surrogate loss function to achieve positive learning results.
>
> In general, we do not think these existing approaches can apply to our problem. The main technical reason is that the cosine is not monotonic, so we cannot design convex surrogate loss functions with the desired properties. Moreover, the pre-existing algorithms are captured by the SQ framework [Dia+20b], and hence are subject to the superpolynomial SQ lower bounds of [Son+17] for our cosine learning problem, which our proposed LLL-based method is able to bypass.
>
> On the lower bound side, an interesting future direction would be to study Hermite decomposition of activation functions (over Gaussian marginals) and establish lower bounds on the performance of low-degree methods [KWB19] or SGD [AGJ21], or local search methods [GZ19] for activation functions whose low-degree Hermite coefficients are exponentially small.
>
> We will add all these comments when we revise the paper.
>
> ***“The reduction shows that a classical poly-time learner of periodic neurons would yield a classical poly-time algorithm for CLWE. As one cares primarily about classical learners of such neurons, hardness for such learners holds under the assumption that CLWE is classically hard, which is of course a weaker assumption than that it is quantum hard … I understand the intent is probably to provide evidence for the weaker assumption (classical hardness of CLWE) by saying people actually believe the stronger one (quantum hardness of CLWE), but the wording could perhaps be tweaked in some places.”***
>
> This is certainly a valid point, and we will certainly revise parts of our text to reflect this and make it clear. Indeed, the reduction from CLWE to cosine learning is entirely classical and the quantum part only steps in once we connect CLWE to SVP.
>
> The reason we insisted on connecting cosine learning directly to SVP, through the quantum reduction, instead of early stopping at CLWE, is because SVP is a *worst-case* problem whose computational hardness has been studied extensively and believed by experts in lattice-based cryptography, whereas CLWE is an *average-case* problem which was introduced relatively recently by [Bru+21]. As CLWE is a relatively recent development, we chose to emphasize the connection between cosine learning and SVP so that our main result would be accessible and convincing to readers familiar with either lattice-based cryptography or hardness of learning. Of course, this exposition choice could have been different if the standing of CLWE hardness within the research community were similar to that of its well-established discrete analog, LWE.
>
> ### References
>
> * [Wil17] Virginia Vassilevska Williams. _On some fine-grained questions in algorithms and complexity_. ICM, 2018.
> * [MR09] Daniele Micciancio and Oded Regev. *Lattice-based Cryptography*. Post-Quantum Cryptography, Springer, 2009.
> * [Bru+21] Joan Bruna, Oded Regev, Min Jae Song, and Yi Tang. _Continuous LWE_. STOC, 2021.
> * [Kak+11] Sham Kakade, Varun Kanade, Ohad Shamir, and Adam Kalai. _Efficient learning of generalized linear and single index models with isotonic regression_. NeurIPS, 2011.
> * [Kan18] Varun Kanade. *Learning real-valued functions*. Lecture notes, 2018.
> * [Dia+20a] Ilias Diakonikolas, Surbhi Goel, Sushrut Karmalkar, Adam R. Klivans, and MahdiSoltanolkotabi. _Approximation schemes for relu regression_. COLT, 2020.
> * [Dia+20b] Ilias Diakonikolas, Daniel Kane, and Nikos Zarifis. _Near-Optimal SQ Lower Bounds for Agnostically Learning Halfspaces and ReLUs under Gaussian Marginals_. NeurIPS, 2020.
> * [Son+17] Le Song, Santosh Vempala, John Wilmes, and Bo Xie. *On the Complexity of Learning Neural Networks*. NeurIPS 2017.
> * [KWB19] Dmitriy Kunisky, Alexander S. Wein, and Afonso S. Bandeira. _Notes on Computational Hardness of Hypothesis Testing: Predictions using the Low-Degree Likelihood Ratio_. arXiv 2019.
> * [AGJ21] Gerard Ben-Arous, Reza Gheissari, and Aukosh Jagannath. _Online stochastic gradient descent on non-convex losses from high-dimensional inference_. JMLR 2021.
> * [GZ19] David Gamarnik and Ilias Zadik. _The Landscape of the Planted Clique Model: Dense Subgraphs and the Overlap Gap Property_. arXiv 2019.

---

> > ### Comment · Reviewer_Dpoz · 2021-08-25
> > **Review update**
> >
> > I appreciate the detailed responses from the authors. They address my questions and it would certainly be nice to see some of these remarks incorporated into the final version.

---

### Official Review · Reviewer_Pqda · 2021-07-16

**Rating:** 7
**Confidence:** 3

**Summary:**

The first line of the abstract nicely summarises the work and I cannot summarise this any better.
"We show a simple reduction which demonstrates the cryptographic hardness of learning a single periodic neuron over isotropic Gaussian distributions in the presence of noise".

The goal is to learn w using samples (x, cos(2 \pi \gamma <w, x>)+noise) where x~N(0, I_d). The noise considered is adversarial. Learning under large noise is not possible information theoretically. The authors give a discussion around their main result that includes an LLL based learning algorithm for the case when the noise is small. The main cryptographic reduction is for medium noise case and is through the CLWE (Continuous Learning with Errors) problem.

**Ethical Concerns:**

This work is theoretical in nature. There are no serious ethical concerns.

**Limitations And Societal Impact:**

This work is theoretical in nature.

**Main Review:**

The paper adds to the understanding of learning simple neural networks. The paper is well written in my opinion and uses allowed space to provide discussion of all relevant issues.  The cryptographic reduction give strong guarantees against "any" polynomial time learning algorithm.

- This additional comment is to acknowledge reading author response. There is no other changes.

**Time Spent Reviewing:**

4

---

> ### Author Response · Authors · 2021-08-09
> **Author Response**
>
> **“The paper is well written in my opinion and uses allowed space to provide discussion of all relevant issues. The cryptographic reduction give strong guarantees against "any" polynomial time learning algorithm.”**
>
> We thank the reviewer for the positive review on our work. We would be more than happy to provide any clarifications that could convince the reviewer to increase their score.

---

### Official Review · Reviewer_CDi6 · 2021-07-19

**Rating:** 7
**Confidence:** 4

**Summary:**


The paper studies the problem of weakly learning single cosine functions (one periodic neuron) when samples come from a gaussian distribution, and in the presence of inverse polynomial amounts of noise. The main result shows that there is no polynomial time algorithm for this learning task under the cryptographic assumption that the Shortest Vector Problem is hard for quantum algorithms. In contrast, if the amount of noise is inverse exponentially small, the paper provides an efficient algorithm for learning.


**Limitations And Societal Impact:**

Yes

**Main Review:**

The paper’s context is lower bounds for neural networks and computational to statistical gaps. The particular kind of functions studied here were previously proved to be hard for SQ algorithms. While the authors emphasize that they prove hardness for all algorithms rather than just the restricted class of SQ algorithms, I would say that the two kinds of results are not quite directly comparable, as the SQ lower bounds are information theoretic, while here we have hardness under crypto assumptions.

More specifically, the problem studies learning a periodic cosine functions with a hidden d-dimensional direction, from a number of samples distributed according to an isotropic Gaussian distribution, in the squared loss setting.

The results essentially characterize the complexity of the problem (in fact the computational to statistical gaps) as a function of the signal to noise ratio.

The first algorithm is a simple learning algorithm that runs in exponential time and learns from a constant amount of noise. The idea is to construct an eps-net of the d-dimensional sphere, sample enough examples, and output the hypothesis in the net that is closest to the empirical value (where rounding is performed to the nearest integer).

The 2nd result is the reduction from the problem of Continuous Learning with Errors (CLWE) to the current learning task with polynomial amount of noise. In fact, this step is also based on a clean observation that one can immediately transform samples from the first problem into samples of the 2nd problem.

Finally, they show an efficient algorithm that only uses linearly many samples (in the dimension) and learns in the presence of inverse exponentially amounts of noise. The problem here becomes that of solving a linear system of equations (which brings up the non-SQ aspect) , and it is solved using the LLL-latice basis reduction algorithm. (To me it also appears similar, at least at a high-level, to the BKW algorithm for solving noisy parities). The result initially is shown for the no-noise case, and hence computations reduce to solving systems of equations over integers. Then the authors observe that the granulation of the equations is coarse enough to be able to tolerate an exponential amounts of noise.

The paper is significant in the fact that it studies the entire SNR range for a natural learning problem. I don’t believe any of the results are particularly difficult to prove – in fact I would say that their approach is the first approach someone familiar with the literature would try.  However, the paper is exceptionally well-written, and the results are elegant and clean.


**Time Spent Reviewing:**

1

---

> ### Author Response · Authors · 2021-08-09
> **Author Response**
>
> ***“The paper is significant in the fact that it studies the entire SNR range for a natural learning problem … (this part is addressed below) ... the paper is exceptionally well-written, and the results are elegant and clean.”***
>
> We thank the reviewer for complimenting our exposition, and providing a very clear and concise summary of our paper.
>
> **_“While the authors emphasize that they prove hardness for all algorithms rather than just the restricted class of SQ algorithms, I would say that the two kinds of results are not quite directly comparable, as the SQ lower bounds are information theoretic, while here we have hardness under crypto assumptions.”_**
>
> This is certainly a valid point, and we will revise parts of our exposition to prevent any possible confusion. Indeed, the key appeal of SQ lower bounds is that they are unconditional (in addition to being quantitative and comprehensive); if an algorithm can be simulated as an SQ algorithm, then it is immediately subject to SQ lower bounds without any unproven assumptions. We will add more emphasis to the fact that our results are conditional on a (worst-case) computational hardness assumption, albeit a well-founded one in the cryptographic community, and that our conditional hardness result and SQ lower bounds are not directly comparable.
>
> **_“I don’t believe any of the results are particularly difficult to prove – in fact I would say that their approach is the first approach someone familiar with the literature would try. ”_**
>
> We agree that our simple reduction showing the hardness of cosine learning is relatively straightforward from CLWE hardness once one is, of course, aware of the CLWE hardness, and notices the key correspondences shown in Section 4, Eq.(6) and (7). We also agree that our covering argument for the constant regime is more or less standard.
>
> Yet, we respectfully disagree that our proposed LLL-based algorithm in the exponentially-small regime, as well as its involved proof of correctness, is the “first-approach” someone in learning theory would try. First, as we explained in our work, the use of lattice-based methods for inference tasks is rather rare in the learning theory literature. On this, let us highlight that, as the reviewer has already pointed out, our suggested LLL-based method is not an SQ-method and in particular bypassed the SQ hardness for the setting as established in [Son+17]. We believe that this makes a case that in the algorithmic learning theory literature it can be beneficial to think beyond the classical SQ framework, potentially borrowing techniques from other computer science fields, such as we do here with algorithmic tools mainly used in cryptography. Furthermore, the algorithm adapts to our cosine learning setting a careful combination of algorithm design ideas, dating back to the breakthrough work of Lagarias, Odlyzko [LO83], and Frieze [Fri86] for solving the random subset sum problems. We believe that at a “first sight” the cosine learning set-up appears significantly different from the random subset sum setup, and in our case, we employed several non-trivial steps to connect the two. Finally, the analysis uses multiple careful probabilistic arguments including the anti-concentration properties of low-degree polynomials [CW01] which is, to the best of our knowledge, an analysis tool that has not been employed in the analysis of any LLL-based algorithms in the past.
>
> ### References
>
> * [LO83] Jeffrey C. Lagarias and Andrew. M. Odlyzko. _Solving low density subset sum problems_. FOCS 1983.
> * [Fri86] Alan M. Frieze. _On the Lagarias-Odlyzko Algorithm for the Subset Sum Problem_. SIAM J. Comput. 15, 1986.
> * [CW01] Anthony Carbery and James Wright. _Distributional and Lq norm inequalities for polynomials over convex bodies in Rn_. Mathematical Research Letters 8, 2001.
> * [Son+17] Le Song, Santosh Vempala, John Wilmes, and Bo Xie. *On the Complexity of Learning Neural Networks*. NeurIPS 2017.

---

> > ### Comment · Reviewer_CDi6 · 2021-08-17
> > **response**
> >
> > I read the author response and I appreciate the clarification about the difficulties in providing the LLL-based algorithm. I particularly like the point that more algebraic methods, such as crypto tools, may provide insights to bypass the limitations of SQ algorithms.

---

### Decision · Program_Chairs · 2021-09-27

**Decision:**

Accept (Poster)

**Comment:**

This work establishes computational hardness for the problem of learning periodic functions with a small amount of adversarial additive noise added to each example. This problem was recently considered in a work by (Song et al., 2017) who established hardness in the Statistical Query model. The current work provides a reduction from the computationally hard problem of Continuous Learning with Errors (CLWE) -- a continuous analogue of LWE that was introduced in a recent work (where its computational hardness was established). A second contribution of the paper is an efficient algebraic algorithm solving the underlying learning problem when the additive noise is very small.
The reviewers uniformly agreed that this is an interesting contribution that should appear in NeurIPS.